# Functional Virtual Adversarial Training for Semi-Supervised Time Series Classification

**Qingyi Pan**
Department of Statistics and Data Science
Tsinghua University
Beijing, China
pqy22@mails.tsinghua.edu.cn

**Yicheng Li**[*]
Department of Statistics and Data Science
Tsinghua University
Beijing, China
liyc22@mails.tsinghua.edu.cn

## Abstract

Real-world time series analysis, such as healthcare, autonomous driving, and solar energy, faces unique challenges arising from the scarcity of labeled data, highlighting the need for effective semi-supervised learning methods. While the Virtual Adversarial Training (VAT) method has shown promising performance in leveraging unlabeled data for smoother predictive distributions, straightforward extensions of VAT often fall short on time series tasks as they neglect the temporal structure of the data in the adversarial perturbation. In this paper, we propose the framework of functional Virtual Adversarial Training (f-VAT) that can incorporate the functional structure of the data into perturbations. By theoretically establishing a duality between the perturbation norm and the functional model sensitivity, we propose to use an appropriate Sobolev ($H^{-s}$) norm to generate structured functional adversarial perturbations for semi-supervised time series classification. Our proposed f-VAT method outperforms recent methods and achieves superior performance in extensive semi-supervised time series classification tasks (e.g., up to $\approx 9\%$ performance improvement). We also provide additional visualization studies to offer further insights into the superiority of f-VAT.

## 1 Introduction

Time series analysis has attracted considerable attention from both academia and industry, due to its relevance to critical domains such as electrocardiogram (ECG) interpretation in medical diagnosis [14] and photovoltaic module power calibration in solar energy systems [35]. However, capturing intrinsic temporal structural properties of time series data, such as noisy fluctuations, long-term trends, and periodic patterns, is still challenging [22], particularly with scarce labeled data [41]. Because manual annotation is labor-intensive and costly with massive unlabeled data, semi-supervised time series classification methods have become a promising research direction.

In this regard, existing methods attempt to leverage massive unlabeled data and limited labeled data to alleviate overfitting. Some studies generate high-confidence pseudo-labels for unlabeled samples and then train a deep model on the expanded dataset [6, 36]. Although such methods can iteratively improve predictive performance, they require careful selection of confidence thresholds and remain susceptible to "confirmation bias": earlier errors in pseudo-labels are likely to be reinforced during training, making it difficult for deep models to correct early biases or capture global trend information [10, 2]. Some recent studies enhance pseudo-labeling with strong data augmentation and self-supervised representation learning to better utilize unlabeled data. For instance, TS-TCC [18] generates the pseudo-label on a weakly augmented view, while encouraging consistent representations

---

[*]Corresponding Author

39th Conference on Neural Information Processing Systems (NeurIPS 2025).

under strong augmentation. However, naive data augmentation methods (e.g., local cropping, random shifting) [42] can disrupt key temporal information, thus affecting generalization.

Recently, Virtual Adversarial Training (VAT) based consistency regularization [31] constructs adversarial perturbations that maximally change the predictions of the model and then penalize the difference between the original and perturbed outputs, thus enforcing local smoothness and effectively tightening the decision boundary in the data manifold. While VAT and its variants have shown promising performance in the fields of computer vision [31], and natural language processing [28], most of these methods are based on the classical Euclidean norm to bound perturbations. Unfortunately, a straightforward extension of VAT to semi-supervised time series classification often proves less effective because bounding virtual perturbations via the Euclidean norm ignores the time series nature of the input data, producing jagged and spiky anomalous patterns that disrupt low-frequency trend information. Consequently, adversarial perturbations generated by standard VAT are not truly worst-case for time series data, making VAT inefficient in improving smoothness of predictive distributions and predictive performance.

To address these challenges, we propose *functional Virtual Adversarial Training (f-VAT)* for semi-supervised time series classification. F-VAT constructs adversarial perturbations in various (possibly) infinite-dimensional spaces, such as the Sobolev spaces [1], which can better capture the underlying structure of the data. For time series, this approach enables us to generate adversarial perturbations that can simultaneously preserve low-frequency trend information and flexibly explore input space, facilitating smoother predictive distribution and alleviating severe overfitting to limited labeled data.

In this paper, we theoretically establish the duality between the perturbation and the smoothness of (non-)linear functional models. Based on theoretical analysis, for time series data, we use an appropriate Sobolev norm to generate functional adversarial perturbations that preserve low-frequency trend information while avoiding "jagged" anomalous patterns. We conduct extensive experiments on real-world datasets to verify that f-VAT significantly outperforms other competitive baselines (e.g., up to 9.42% on CricketX and 8.30% on SelfReg). Further visualization indicates that, compared to the original VAT, our proposed functional adversarial perturbations lead to more stable convergence and better final performance.

In summary, our main contributions are as follows.

- We propose the framework of functional Virtual Adversarial Training (f-VAT) that allows us to construct perturbations in various function spaces. For linear and non-linear models, we theoretically establish the duality between the perturbation norm and gradient sensitivity (Theorem 3.3 and Theorem 3.4), showing how to generate structured adversarial perturbations.

- For the semi-supervised time series classification problem, we propose to use an appropriate Sobolev norm ($H^{-s}$ norm in Section 3.3) to generate structured adversarial perturbations. The Sobolev norm allows us to properly control the model's sensitivity to high-frequency noises while capturing low-frequency trend information.

- Extensive experimental results (Section 4.2) on semi-supervised time series classification are provided to demonstrate the superiority of f-VAT over existing methods. Additional visualization results (Section 4.3) indicate that functional adversarial perturbations can significantly smooth the loss landscape to achieve stable convergence and better performance.

## 2 Preliminaries on Virtual Adversarial Training

Let us briefly introduce the procedure of Virtual Adversarial Training (VAT) [31]. We consider the semi-supervised classification problem with a small labeled dataset and a large unlabeled dataset. Let $\{X_1, X_2, \ldots, X_n\} \in \mathcal{X}$ denote the entire input data, where each $X_i$ can be associated with a label $y_i \in \mathcal{Y} = \{1, 2, \ldots, K\}$ if it is labeled, or $y_i$ is unknown if it is unlabeled. We denote by $\mathcal{D}^l$ the labeled dataset, $\mathcal{D}^u$ the unlabeled dataset, and $\mathcal{D} = \mathcal{D}^l \cup \mathcal{D}^u$ the entire dataset. Our goal is to learn the conditional probability (vector) $p(\cdot \mid X_i; \theta) = f_\theta(X) \in \mathbb{R}^K$.

The idea of VAT is to generate adversarial perturbations $r_i$ for each sample $X_i$ and then penalize the difference between the original and perturbed outputs, facilitating the local smoothness of predictive distribution. Formally, the Local Distributional Smoothness $\text{LDS}(X_i, r_i; f_\theta)$ is used to measure such

difference for the model outputs. In this paper, we mainly focus on the squared loss

$$\text{LDS}(X_i, r_i; f_\theta) = \|f_\theta(X_i + r_i) - f_\theta(X_i)\|_2^2, \tag{1}$$

while another common choice is the Kullback-Leibler (KL) divergence [31]. We adopt the squared error, because its symmetry yields better numerical stability in the early stages of training [13]. The adversarial perturbation $r_i$ aims to maximize the LDS loss, characterizing the worst-case local perturbation for the model $f_\theta$ as

$$r_i^* = \underset{r_i : \|r_i\|_2 \le \epsilon}{\arg\max} \, \text{LDS}(X_i, r_i; f_\theta), \tag{2}$$

where $\epsilon > 0$ is a small hyperparameter controlling the perturbation norm in Euclidean space. Then, the VAT loss over $X_i$ is defined as

$$\mathcal{L}^{\text{VA}}(X_i; f_\theta) = \text{LDS}(X_i, r_i^*; f_\theta) = \underset{r_i : \|r_i\|_2 \le \epsilon}{\max} \, \text{LDS}(X_i, r_i; f_\theta). \tag{3}$$

In practice, since the analytic expression of $r_i$ is intractable, an interactive gradient ascent [31] is used to approximate the optimal perturbation $r_i^*$. The overall objective function combines the supervised loss $\mathcal{L}_0$ on the labeled set $\mathcal{D}^l$ and the VAT loss $\mathcal{L}^{\text{VA}}$ on the entire dataset $\mathcal{D}$ is

$$\mathcal{L}(\mathcal{D}; f_\theta) = \mathcal{L}_0(\mathcal{D}^l; f_\theta) + \mathcal{L}^{\text{VA}}(\mathcal{D}; f_\theta), \tag{4}$$

where $\mathcal{L}^{\text{VA}}(\mathcal{D}; f_\theta) = \frac{1}{n} \sum_{i=1}^n \mathcal{L}^{\text{VA}}(X_i; f_\theta)$ is the VAT loss over the entire dataset $\mathcal{D}$.

## 3 Functional Virtual Adversarial Training

One key aspect of VAT is the choice of the norm used to bound the local perturbation $\|r\| \le \epsilon$ in Eq. (2). In the literature [31], it seems to be straightforward to use the Euclidean norm $\|\cdot\|_2$ as the input is typically represented vector in $\mathbb{R}^d$. However, when the input space $\mathcal{X}$ consists of functions, which is the case for time series data, the choice of the norm can become more intricate. As we will see in the following theoretical analysis, the choice of the norm can have a significant impact on the VAT loss and its interpretation. In this section, for simplicity, we consider one-dimensional output $f : \mathcal{X} \to \mathbb{R}$ and focus on some fixed sample $x_0$, while our results can be easily extended to the case where $f$ is a vector-valued function.

### 3.1 The Duality Perspective of VAT

To start with, let us first consider the setting where the input $x$ is a vector in $\mathbb{R}^d$, but we consider a more general setting where the perturbation is bounded by a general norm. Let $\Sigma$ be a positive definite matrix, we denote the $\Sigma$-norm by $\|v\|_\Sigma := \sqrt{v^\top \Sigma v}$. Replacing the Euclidean norm in Eq. (2) with the $\Sigma$-norm, we introduce the following VAT loss:

$$\mathcal{L}_\Sigma^{\text{VA}}(x_0, \epsilon; f) = \underset{r : \|r\|_\Sigma \le \epsilon}{\sup} \, \text{LDS}(x_0, r; f). \tag{5}$$

To illustrate the impact of the $\Sigma$-norm on the VAT loss, let us first consider the case where $f(x)$ is a linear function. We have the following result.

**Proposition 3.1.** *Let $f(x) = \langle \beta, x \rangle$ be a linear model. Then, the VAT loss Eq. (5) is equivalent to*

$$\mathcal{L}_\Sigma^{VA}(x_0, \epsilon; f) = \epsilon^2 \|\beta\|_{\Sigma^{-1}}^2. \tag{6}$$

*Proof.* Since $f(x)$ is linear, we have

$$\mathcal{L}_\Sigma^{\text{VA}}(x_0, \epsilon; f) = \underset{r : \|r\|_\Sigma \le \epsilon}{\sup} |\langle \beta, x_0 \rangle - \langle \beta, x_0 + r \rangle|^2 = \underset{r : \|r\|_\Sigma \le \epsilon}{\sup} |\langle \beta, r \rangle|^2.$$

This supremum is achieved when $r$ is in the direction of $\Sigma^{-1}\beta$, i.e.,

$$r^* = \epsilon \Sigma^{-1}\beta / \left\|\Sigma^{-1}\beta\right\|_\Sigma = \epsilon \Sigma^{-1}\beta / \|\beta\|_{\Sigma^{-1}},$$

and thus

$$\mathcal{L}_\Sigma^{\text{VA}}(x_0, \epsilon; f) = \underset{r : \|r\|_\Sigma \le \epsilon}{\sup} |\langle \beta, r \rangle|^2 = \langle \beta, r^* \rangle^2 = \epsilon^2 \|\beta\|_{\Sigma^{-1}}^2.$$

$\square$

Moreover, the result can be extended to the case where $f(x)$ is a continuously differentiable function as follows. The proof is deferred to Appendix A.3.

**Proposition 3.2.** *Let $f(x)$ be continuously differentiable. Then, we have*

$$\lim_{\epsilon \to 0^+} \epsilon^{-2} \mathcal{L}_{\Sigma}^{VA}(x_0, \epsilon; f) = \|\nabla f(x_0)\|_{\Sigma^{-1}}^2. \tag{7}$$

The results from Proposition 3.1 and Proposition 3.2 reveal a simple but profound duality structure underlying the VAT loss. First, as the gradient $\nabla f(x_0)$ reflects the sensitivity of the function, the VAT loss can be interpreted as a measure of the model's sensitivity to perturbations in the input space. Moreover, the use of the $\Sigma$-norm in the perturbation constraint $\|r\|_{\Sigma} \leq \epsilon$ naturally introduces a dual norm $\|\cdot\|_{\Sigma^{-1}}$ penalizing the sensitivity of $f$ along the different directions. This is reminiscent of duality in optimization, where constraints in the primal space translate to penalties in the dual space.

Therefore, one of the crucial questions is how to choose the matrix $\Sigma$ when designing the VAT loss. If $\Sigma$ is isotropic (e.g., $\Sigma = I$), the VAT loss reduces to the Euclidean sensitivity $\|\nabla f(x_0)\|_2^2$. If $\Sigma$ is anisotropic, the VAT loss prioritizes robustness along directions where $\Sigma$ assigns lower curvature. This coincides with our intuition: allowing larger adversarial perturbations results in more constraint on the model's sensitivity in those directions. While it seems to be trivial when $\Sigma$ is a diagonal matrix that simply scales the input, the choice of $\Sigma$ can be more complex and encompasses structural information about the input space, which would lead to substantial differences in the VAT loss. This is what we will see under the time series data.

## 3.2 VAT under Functional Inputs

In the setting of the time series data [24], the input $x$ is viewed as a function of some interval $T$, which would require a more sophisticated treatment on the theoretical analysis. Let $L^2 = L^2(T)$ be the Hilbert space of square integrable functions on $T$ and denote the inner product by $\langle \cdot, \cdot \rangle_{L^2(T)}$. We suppose that the input $x \in L^2(T)$. For the perturbation, let us take another Banach space $E$ and denote by $\|\cdot\|_E$ the norm in $E$. Typically, we can choose $E$ to be a Sobolev space, which will be discussed in the next subsection. The dual space $E^*$ of $E$ is defined as the space of continuous linear functionals on $E$, which is equipped with the dual norm $\|\cdot\|_{E^*}$ defined by

$$\|g\|_{E^*} = \sup_{x \in E : \|x\|_E \leq 1} |g(x)|.$$

To avoid technicalities, we will only provide informal results here. For a formal treatment and rigorous proofs, we refer the readers to Appendix A in appendix.

The functional virtual adversarial training (f-VAT) introduces a more general form of the VAT loss, which is defined as

$$\mathcal{L}_E^{VA}(x_0, \epsilon; f) = \sup_{r \in L^2 \cap E : \|r\|_E \leq \epsilon} \text{LDS}(x_0, r; f). \tag{8}$$

Our first result parallels Proposition 3.1 in the functional setting.

**Theorem 3.3** (Informal). *Let $f(x) = \langle \beta, x \rangle_{L^2(T)}$, $\beta \in L^2(T)$ be a functional linear model. Then, the loss Eq.* (8) *is equivalent to*

$$\mathcal{L}_E^{VA}(x_0, \epsilon; f) = \epsilon^2 \|\beta\|_{E^*}^2, \tag{9}$$

*where $\|\beta\|_{E^*}$ can be infinite if $\beta \notin E^*$.*

Then, the next result is the functional version of Proposition 3.2.

**Theorem 3.4** (Informal). *Let $f : L^2(T) \to \mathbb{R}$ be differentiable at $x_0 \in L^2(T)$. Then,*

$$\lim_{\epsilon \to 0^+} \epsilon^{-2} \mathcal{L}_E^{VA}(x_0, \epsilon; f) = \|\nabla f(x_0)\|_{E^*}^2. \tag{10}$$

Theorem 3.3 and Theorem 3.4 establish the duality of VAT under general functional spaces, showing that a constraint over the $E$-norm of the perturbation results in penalizing the dual space $E^*$-norm of the model's sensitivity. These results also recover Proposition 3.1 and Proposition 3.2 as special cases, since $\|\cdot\|_{\Sigma}$ and $\|\cdot\|_{\Sigma^{-1}}$ are dual under the canonical Euclidean norm. While finite dimensional spaces under different norms are equivalent, the infinite dimensional spaces can be substantially

different. The functional setting generalizes the vector-based analysis in the preceding subsection, but its infinite-dimensional nature allows for a richer specification of perturbation constraints that align with the geometry of functional data.

These findings offer several key insights into the behavior and potential of VAT when applied to functional data, such as time series. A critical implication of these results lies in the role of the space $E$ (or equivalently, the norm $\|\cdot\|_E$), which defines the constraint of the perturbation $r$. It is known that the stronger norm of $E$ (thus a "smaller" space) leads to a weaker norm of $E^*$ (thus a "bigger" space), which can be seen from the duality between $\|\cdot\|_\Sigma$ and $\|\cdot\|_{\Sigma^{-1}}$. We will use the Sobolev spaces to show a concrete example in the next section. Consequently, the result highlights that, if we want to impose a stronger norm on the sensitivity of the method, we should, in contrast, relax the norm of the perturbation.

Moreover, this perspective also has practical implications for designing VAT-based training procedures. The choice of $E$ and thus $E^*$ determines the directions and structure of perturbations against which the model is made robust. For time series data, where the inputs are functions over a time interval $T$, this flexibility is particularly valuable. Time series often exhibit structural properties such as smoothness, periodicity, or specific temporal dependencies. By selecting a space that captures these characteristics—for instance, a Sobolev space that penalizes high-frequency oscillations through derivative-based norms—the VAT loss can be tailored to prioritize robustness under a particular sense. This could be especially beneficial in applications where high-frequency components are considered noise, which encourages the model to focus on the underlying smooth signal.

## 3.3 Structured Adversarial Perturbations via Sobolev Spaces

Sobolev spaces [1] are a powerful tool in functional analysis, particularly useful for analyzing the smoothness and regularity of functions. Using Sobolev spaces, we can define a structured norm for the f-VAT constraint that captures the smoothness properties of the perturbations. In this subsection, we will briefly introduce the Sobolev spaces, their duality, and practical ways to compute their norms with discrete points. We refer the readers to Appendix B in the appendix for more details.

For a positive integer $s$, the Sobolev space $H^s(T)$ comprises functions in $L^2(T)$ whose weak derivatives up to order $s$ are also in $L^2(T)$. The weak derivative generalizes differentiation to functions lacking classical smoothness, enabling control over their regularity. A function with classical continuous derivatives up to order $s$ is in $H^s(T)$, but the converse is not necessarily true. The norm in a Sobolev space is simply the sum of the $L^2(T)$ norms of all derivatives up to order $s$ and the inner product can be similarly defined, which make $H^s(T)$ a Hilbert space.

More broadly, $H^\alpha(T)$ can be defined for any real $\alpha \in \mathbb{R}$. Non-integer $\alpha$ captures fractional smoothness, while negative $\alpha$ encompasses distributions rougher than $L^2(T)$ functions. Moreover, it is known that for $s > 0$, $H^s$ is densely embedded into $L^2(T)$, while $L^2(T)$ is dense in $H^{-s}(T)$.

The duality of Sobolev spaces can be easily identified with the index $s$. The dual of $H^\alpha(T)$ is $H^{-\alpha}(T)$ under the $L^2(T)$ inner product. This duality enables us to frame the VAT loss using the dual norm $\|\cdot\|_{H^{-s}}$, assessing model sensitivity to perturbations constrained by $\|\cdot\|_{H^s}$ for $s > 0$.

**Computing the Norm with Discrete Observations.** In practical settings, time series are discrete vectors in $\mathbb{R}^N$. For simplicity, we assume that the time series is sampled at $N$ equidistant points, but the results can be extended to non-equidistant points with minor adjustments. Then, we can approximate the Sobolev norms using finite difference operators [9]. Define the first-order difference matrix $D_1 \in \mathbb{R}^{N \times N}$ with $(D_1 r)_i = r_{i+1} - r_i$ and higher-order differences $D_k = D_1^k$. For integer $k$, the squared $H^k$ norm of $k$ with discrete observations $r_N \in \mathbb{R}^N$ can be approximated as:

$$\|r\|_{H^k}^2 \approx r_N^\top A_{k,N} r_N, \quad \text{where} \quad A_{k,N} = I_N + D_1^\top D_1 + D_2^\top D_2 + \cdots + D_k^\top D_k,$$

where $I_N$ is the $N \times N$ identity matrix. This sums the $\ell_2$-norm of $r$ and its discrete derivatives up to order $k$.

However, we actually will use the $H^{-s}$ norm for the perturbation, so the above formulation is not directly applicable. For general $\alpha \in \mathbb{R}$, we will use the spectral method [9] instead. It is known that the Sobolev space $H^\alpha(T)$ can be represented via the fractional power of the Laplacian operator. We have $H^\alpha(\mathbb{R}^d) = \left\{ f = (I - \Delta)^{-\alpha/2} u : u \in L^2(\mathbb{R}^d) \right\}$ and $\|f\|_{H^\alpha}^2 = \left\| (I - \Delta)^{\alpha/2} f \right\|_{L^2(\mathbb{R}^d)}^2$.

Therefore, let us take the discrete negative Laplacian matrix

$$L_N = \begin{bmatrix} 2 & -1 & 0 & \cdots & 0 & 0 \\ -1 & 2 & -1 & \cdots & 0 & 0 \\ \vdots & \vdots & \ddots & \ddots & \vdots & \vdots \\ 0 & 0 & \cdots & -1 & 2 & -1 \\ 0 & 0 & \cdots & 0 & -1 & 2 \end{bmatrix}.$$

We can compute the $H^\alpha$ norm by

$$\|r\|_{H^\alpha}^2 \approx r_N^\top (I_N + L_N)^\alpha r_N, \quad \alpha \in \mathbb{R}, \tag{11}$$

where $(I_N + L_N)^\alpha$ is defined via the spectral theorem: if $I_N + L_N = Q\Lambda Q^\top$ with diagonal $\Lambda$, then $(I_N + L_N)^\alpha = Q\Lambda^\alpha Q^\top$, where $\Lambda^\alpha$ takes the exponent of each diagonal entry. This formulation is also valid for non-integer or negative $s$, where we note that $(I_N + L_N)$ is positive definite and thus the diagonal entries of $\Lambda$ are positive. This gives us practical ways to compute the Sobolev norms with discrete observations.

### 3.4 The Algorithm

Let us now summarize the f-VAT procedure. Overall, we apply a mini-batch gradient descent to minimize the total loss. In practice, since the analytic expression of optimal perturbation $r^*$ is intractable in Eq. (2), we approximate it using gradient ascent. We illustrate each step of updating $\theta$ in Algorithm 1, where we recall the labeled loss $\mathcal{L}_0(\mathcal{D}^l; f_\theta)$ and LDS in Eq. (1). According to our theory in Theorem 3.3 and the duality of Sobolev spaces, we use $H^{-s}$ for the norm of adversarial perturbation $r$ to impose a smoothness constraint on the functional model. In this paper, we mainly use $s = 2$ to penalize (weak) derivative up to the second order of the model, empirically verified by ablation studies in Fig. 1.

---

**Algorithm 1** Functional Virtual Adversarial Training Step

---

1: **Input**: Data batch $\mathcal{D}, \mathcal{D}^l$, model $f_\theta$, order of the Sobolev norm $s \geq 0$, radius $\epsilon$, adversarial iterations $L$, learning rate $\eta$.
2: **for** each sample $X_i \in \mathcal{D}$ **do** ▷ Approximate $r_i^*$
3: $\quad$ Randomly initialize perturbation vector $r_i$ over $\|r_i\|_{H^{-s}} \leq \epsilon$.
4: $\quad$ **for** $\ell = 1 \to L$ **do**
5: $\quad\quad$ Gradient ascent $r_i \leftarrow r_i + \eta\nabla_{r_i}\mathrm{LDS}(X_i, r_i; f_\theta)$
6: $\quad\quad$ Normalize $r_i \leftarrow \epsilon \frac{r_i}{\|r_i\|_{H^{-s}}}$.
7: $\quad$ **end for**
8: **end for**
9: $\theta \leftarrow \theta - \eta\nabla_\theta\mathcal{L}(\theta)$, where $\mathcal{L}(\theta) = \mathcal{L}_0(\mathcal{D}^l; f_\theta) + \frac{1}{|\mathcal{D}|}\sum_{X_i \in \mathcal{D}}\mathrm{LDS}(X_i, r_i; f_\theta)$

---

## 4 Experiments

### 4.1 Experimental Settings

We use dozens of publicly available datasets from the UCR and UEA repositories [11], including the representative univariate dataset (i.e., CricketX, UWave, and InsectWing) and the multivariate dataset (i.e., SelfReg, NATOPS, and Heartbeat) in [19]. These representative datasets are from difficult to easy, and widely-used in semi-supervised time series classification [19, 21]. Additionally, we construct more empirical results on several large-scale China Securities Index (CSI) datasets (i.e., CSI 50 and 500 futures) spanning from 2020 to 2023 for predicting directions (upward or downward) of futures prices [44, 33]. The dataset collects records spanning from 2020 to 2022. Each time step contains bid/ask prices and corresponding volumes. Following [19], each dataset is split into train (60%), valid (20%), and test set (20%). We rescale each dataset into the range $[0, 1]$ for numerical stability. We refer readers to Appendix C for more details.

## 4.2 Performance Evaluation

**Semi-Supervised Performance** Following [19], we compare f-VAT with six deep learning-based methods, including recent SemiTime [19] and TapNet [43]. The baseline SupL represents the deep model trained only on the labeled dataset. We adopt the baseline results to Table 1 from the original work [19, 25, 43]. Following [40], we average the performance of each method over five runs with different random seeds and train/valid/test splits. Table 1 reports the mean and standard error from multiple runs.

Table 1: The semi-supervised classification accuracy (%) with standard deviation across six real-world datasets. Best performance in boldface.

| Dataset | Ratio | SupL | PI | MTL | SemiTime | TapNet | VAT | f-VAT |
|---|---|---|---|---|---|---|---|---|
| CricketX | 10% | $44.88_{\pm0.51}$ | $38.87_{\pm2.26}$ | $40.94_{\pm1.97}$ | $44.88_{\pm3.13}$ | $39.42_{\pm0.82}$ | $42.85_{\pm3.97}$ | $\mathbf{49.18}_{\pm1.96}$ |
|  | 20% | $51.61_{\pm0.45}$ | $44.44_{\pm2.91}$ | $50.12_{\pm1.22}$ | $51.61_{\pm0.66}$ | $51.41_{\pm0.31}$ | $49.14_{\pm0.50}$ | $\mathbf{57.91}_{\pm3.58}$ |
|  | 40% | $58.71_{\pm0.46}$ | $53.39_{\pm2.18}$ | $55.10_{\pm1.12}$ | $58.71_{\pm2.78}$ | $58.97_{\pm0.72}$ | $58.63_{\pm0.50}$ | $\mathbf{68.39}_{\pm2.25}$ |
| UWave | 10% | $81.46_{\pm0.18}$ | $81.53_{\pm0.54}$ | $76.35_{\pm0.56}$ | $81.46_{\pm0.60}$ | $82.34_{\pm0.58}$ | $94.41_{\pm0.09}$ | $\mathbf{94.82}_{\pm0.39}$ |
|  | 20% | $84.57_{\pm0.87}$ | $81.66_{\pm0.74}$ | $81.77_{\pm0.94}$ | $84.57_{\pm0.49}$ | $86.35_{\pm0.43}$ | $95.53_{\pm0.31}$ | $\mathbf{96.45}_{\pm0.27}$ |
|  | 40% | $86.91_{\pm0.98}$ | $86.45_{\pm1.20}$ | $86.91_{\pm0.68}$ | $86.91_{\pm0.47}$ | $89.24_{\pm0.69}$ | $94.76_{\pm0.54}$ | $\mathbf{97.23}_{\pm0.43}$ |
| InsectWing | 10% | $54.96_{\pm1.25}$ | $43.16_{\pm3.20}$ | $50.45_{\pm1.01}$ | $54.96_{\pm1.61}$ | $55.53_{\pm1.18}$ | $55.49_{\pm1.28}$ | $\mathbf{58.01}_{\pm1.12}$ |
|  | 20% | $59.01_{\pm1.13}$ | $48.35_{\pm0.81}$ | $56.43_{\pm0.88}$ | $59.01_{\pm1.56}$ | $60.36_{\pm0.38}$ | $61.27_{\pm0.19}$ | $\mathbf{61.28}_{\pm1.86}$ |
|  | 40% | $62.38_{\pm1.39}$ | $55.32_{\pm2.04}$ | $60.90_{\pm0.87}$ | $62.38_{\pm0.76}$ | $63.87_{\pm1.41}$ | $63.48_{\pm0.30}$ | $\mathbf{64.81}_{\pm1.15}$ |
| SelfReg | 10% | $46.49_{\pm2.01}$ | $50.44_{\pm0.76}$ | $50.88_{\pm2.01}$ | $49.68_{\pm2.83}$ | $50.87_{\pm3.31}$ | $53.12_{\pm4.51}$ | $\mathbf{59.31}_{\pm3.06}$ |
|  | 20% | $52.44_{\pm3.15}$ | $53.94_{\pm2.63}$ | $52.19_{\pm2.01}$ | $52.63_{\pm1.31}$ | $54.39_{\pm2.74}$ | $55.76_{\pm0.35}$ | $\mathbf{61.60}_{\pm1.13}$ |
|  | 40% | $51.31_{\pm3.48}$ | $55.69_{\pm2.74}$ | $56.14_{\pm2.01}$ | $49.56_{\pm1.72}$ | $54.38_{\pm0.76}$ | $53.47_{\pm1.04}$ | $\mathbf{64.44}_{\pm3.13}$ |
| NATOPS | 10% | $68.98_{\pm2.89}$ | $75.83_{\pm4.39}$ | $73.91_{\pm3.73}$ | $68.52_{\pm0.81}$ | $70.37_{\pm7.12}$ | $82.38_{\pm0.96}$ | $\mathbf{86.04}_{\pm1.41}$ |
|  | 20% | $81.02_{\pm1.60}$ | $82.51_{\pm1.25}$ | $82.41_{\pm2.89}$ | $80.09_{\pm2.12}$ | $77.77_{\pm1.39}$ | $82.81_{\pm0.52}$ | $\mathbf{86.25}_{\pm1.38}$ |
|  | 40% | $88.89_{\pm2.78}$ | $88.27_{\pm1.19}$ | $90.27_{\pm1.39}$ | $87.49_{\pm2.41}$ | $82.87_{\pm2.12}$ | $90.15_{\pm1.60}$ | $\mathbf{93.13}_{\pm0.15}$ |
| Heartbeat | 10% | $67.08_{\pm3.57}$ | $72.13_{\pm1.99}$ | $71.61_{\pm2.47}$ | $71.61_{\pm1.71}$ | $72.84_{\pm1.23}$ | $73.86_{\pm0.59}$ | $\mathbf{76.25}_{\pm1.22}$ |
|  | 20% | $73.25_{\pm0.71}$ | $72.01_{\pm0.78}$ | $73.66_{\pm0.71}$ | $74.49_{\pm1.43}$ | $73.24_{\pm1.88}$ | $71.59_{\pm0.13}$ | $\mathbf{76.46}_{\pm1.06}$ |
|  | 40% | $67.08_{\pm1.89}$ | $73.28_{\pm1.53}$ | $73.61_{\pm3.07}$ | $72.43_{\pm3.11}$ | $73.66_{\pm0.71}$ | $75.00_{\pm0.11}$ | $\mathbf{77.28}_{\pm0.40}$ |

Table 1 shows that f-VAT consistently outperforms other competitive baselines across all real-world datasets reported in various label ratios. For instance, f-VAT achieves up to 9.42% performance improvements on CricketX and 8.30% on SelfReg with label ratio $\alpha = 0.4$, while in the label-scarce scenario $\alpha = 0.1$, f-VAT still achieves up to 6.19% performance improvements on SelfReg.

To further verify the superiority of f-VAT, we randomly sample 30 datasets from UCR/UEA datasets [16]. Then, we add a classical baseline meanTeacher [38] and a recent state-of-the-art Class-Aware Temporal and Contextual Contrasting (CA-TCC) [18] for fair comparison. Table 2 shows that f-VAT still consistently outperforms other competitive baselines in all settings. Due to the page limit, the details of more empirical results can be found in Appendix E.

Table 2: The average accuracy (%) and average rank under different label ratios.

| Method | 10% | | 20% | | 40% | |
|---|---|---|---|---|---|---|
|  | AvgAcc | AvgRank | AvgAcc | AvgRank | AvgAcc | AvgRank |
| SupL | 35.31 | 6.67 | 36.92 | 7.00 | 37.15 | 7.33 |
| PI | 53.09 | 3.93 | 55.16 | 4.40 | 63.60 | 4.47 |
| MTL | 45.19 | 5.70 | 45.72 | 5.87 | 46.11 | 6.80 |
| meanTeacher | 42.89 | 5.93 | 50.94 | 4.87 | 63.85 | 4.13 |
| SemiTime | 56.53 | 3.57 | 58.93 | 3.77 | 69.02 | 3.13 |
| TapNet | 58.67 | 3.70 | 60.41 | 3.40 | 70.28 | 3.17 |
| CA-TCC | 58.07 | 3.37 | 59.84 | 3.67 | 63.27 | 4.27 |
| f-VAT | 65.85 | 1.50 | 68.87 | 1.50 | 76.24 | 1.53 |

Additionally, we construct more empirical results on several large-scale China Securities Index (CSI) datasets (i.e., CSI 50 and 500 futures). Table 3 presents the performance of various semi-supervised methods across different label ratios, and shows that f-VAT significantly outperforms other competitive baselines on futures datasets, especially on more volatile CSI 500 futures. This is because f-VAT's adversarial perturbations incorporating key temporal structure facilitate deep models to effectively use unlabeled samples to yield smoother predictive distribution with better generalization.

Table 3: The performance comparison on domestic futures datasets like CSI 50 and 500 futures. Best performance in boldface.

| Futures | Ratio | SupL | PI | MTL | SemiTime | TapNet | CA-TCC | f-VAT |
|---|---|---|---|---|---|---|---|---|
| 50 | 10% | $40.05_{\pm1.38}$ | $42.87_{\pm1.45}$ | $53.94_{\pm0.13}$ | $55.19_{\pm0.77}$ | $54.62_{\pm0.54}$ | $55.72_{\pm1.16}$ | $\mathbf{58.64}_{\pm0.53}$ |
| | 20% | $45.08_{\pm1.45}$ | $47.26_{\pm0.73}$ | $54.97_{\pm0.61}$ | $56.69_{\pm0.50}$ | $56.93_{\pm1.28}$ | $58.21_{\pm0.44}$ | $\mathbf{62.09}_{\pm0.26}$ |
| | 40% | $50.69_{\pm3.53}$ | $52.42_{\pm1.21}$ | $56.97_{\pm0.57}$ | $57.34_{\pm0.20}$ | $59.75_{\pm1.09}$ | $59.80_{\pm1.32}$ | $\mathbf{64.78}_{\pm0.45}$ |
| 500 | 10% | $34.23_{\pm0.24}$ | $44.00_{\pm0.06}$ | $39.53_{\pm1.35}$ | $38.86_{\pm2.04}$ | $40.53_{\pm0.83}$ | $39.79_{\pm1.54}$ | $\mathbf{43.77}_{\pm0.73}$ |
| | 20% | $35.38_{\pm1.06}$ | $46.57_{\pm0.48}$ | $45.26_{\pm0.29}$ | $46.04_{\pm0.12}$ | $44.58_{\pm1.70}$ | $45.05_{\pm1.74}$ | $\mathbf{52.14}_{\pm0.25}$ |
| | 40% | $43.85_{\pm1.49}$ | $49.25_{\pm0.55}$ | $47.66_{\pm1.50}$ | $50.65_{\pm1.05}$ | $51.30_{\pm1.36}$ | $54.29_{\pm0.87}$ | $\mathbf{58.66}_{\pm0.46}$ |

**Fully-Supervised Performance**   The proposed f-VAT can be easily extended to fully supervised settings. We compare f-VAT with several competitive supervised learning methods. ED [3] is the classical one-nearest-neighbor classifier based on Euclidean distance. TapNet [43] and ShapeNet [26] are deep learning-based methods that leverage manual shapelet-based features for better representations. Additionally, we include two non-neural methods called ROCKET [17] and HiveCOTE [4]. The empirical results in Table 4 are taken from the original work [3, 43, 26]. We average the performance of each method over five runs with different random seeds, and report mean values with standard deviations.

Table 4: The performance comparison between f-VAT and other baselines in fully-supervised settings. We report the mean and standard deviation over five runs. Best performance in boldface.

| Dataset | Hive-COTE | ROCKET | ED | TapNet | ShapeNet | VAT | f-VAT |
|---|---|---|---|---|---|---|---|
| CricketX | $74.10_{\pm0.03}$ | $76.10_{\pm0.01}$ | $62.90_{\pm0.14}$ | $66.20_{\pm0.25}$ | $68.30_{\pm0.51}$ | $68.54_{\pm1.40}$ | $\mathbf{77.25}_{\pm0.94}$ |
| UWave | $92.10_{\pm0.02}$ | $93.70_{\pm0.04}$ | $88.10_{\pm0.12}$ | $89.40_{\pm0.69}$ | $90.60_{\pm0.13}$ | $92.43_{\pm0.47}$ | $\mathbf{97.75}_{\pm0.13}$ |
| InsectWing | $62.20_{\pm0.01}$ | $64.70_{\pm0.01}$ | $60.20_{\pm0.13}$ | $67.30_{\pm0.11}$ | $66.30_{\pm0.02}$ | $70.01_{\pm1.16}$ | $\mathbf{71.70}_{\pm0.55}$ |
| SelfReg | $51.60_{\pm0.67}$ | $51.40_{\pm0.59}$ | $48.30_{\pm0.12}$ | $55.10_{\pm0.26}$ | $57.80_{\pm0.03}$ | $58.75_{\pm1.25}$ | $\mathbf{60.21}_{\pm0.68}$ |
| NATOPS | $82.80_{\pm0.32}$ | $88.50_{\pm0.44}$ | $85.10_{\pm0.18}$ | $93.90_{\pm0.01}$ | $88.30_{\pm0.03}$ | $87.58_{\pm1.89}$ | $\mathbf{97.50}_{\pm0.51}$ |
| Heartbeat | $72.20_{\pm0.52}$ | $71.70_{\pm0.02}$ | $61.90_{\pm0.09}$ | $72.10_{\pm1.43}$ | $75.60_{\pm0.02}$ | $76.08_{\pm0.82}$ | $\mathbf{78.75}_{\pm0.41}$ |

Table 4 shows that f-VAT consistently outperforms other baselines with an average improvement of 2.4% in all settings, especially for datasets containing long-term trend structures (i.e., up to 4.05% improvement in UWave). Also, f-VAT exhibits lower variance than VAT, as the functional adversarial perturbation imposes stronger regularization on the model's gradient sensitivity and improves the model's stability in fully-supervised settings.

**Altering Order of Sobolev Norm**   We further explore how the order $s$ of Sobolev norm influences the performance of f-VAT. Fig. 1 illustrates that the performance of f-VAT in NATOPS with label ratio $\alpha = 0.4$ under different $s$, where the error bar reports the standard deviation over 5 runs. We observe that the model's accuracy peaks at $s = 2$. This is because under-regularization ($s \leq 1$) or over-regularization ($s \geq 3$) both would degenerate predictive performance. Based on this, we choose $s = 2$ in the rest of experiments. Additionally, we evaluate the performance on more datasets in Table 5, where $s = 0$ reduces to the original VAT. Table 5 shows that set-

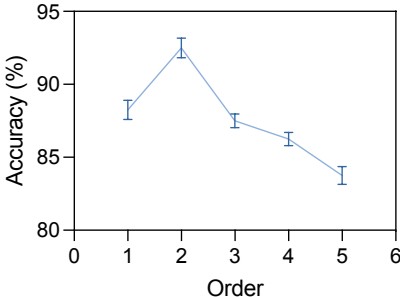

Figure 1: Various order of Sobolev norm.

ting $s = 2$ achieves the best performance in almost all settings. For high volatile time series data, setting high order $s = 3$ generates adversarial perturbations that preserve low-frequency temporal information, while for relatively stable time series data, the small order $s = 1$ allows the perturbation to flexibly explore input space. In practice, it suffices to evaluate $s \in \{1, 2, 3, 4\}$ based on the validation set and report the average performance over several runs.

Moreover, to verify the model-agnostic property of f-VAT, we compare the performance of different architectures trained by f-VAT in Appendix D. The empirical results suggest that Temporal Convolutional Network (TCN) [5] outperforms other architectures in most settings, so we adopt the eight-layer TCN as main architecture.

Table 5: The performance of f-VAT with different Sobolev norm order $s$.

| $s$ | CricketX | UWave | InsectWing | NATOPS | SelfReg |
|---|---|---|---|---|---|
| 0 | 58.63 $_{\pm 0.50}$ | 94.76 $_{\pm 0.54}$ | 63.48 $_{\pm 0.30}$ | 90.15 $_{\pm 1.60}$ | 53.47 $_{\pm 1.04}$ |
| 1 | 59.91 $_{\pm 2.32}$ | 96.54 $_{\pm 0.67}$ | 66.70 $_{\pm 0.50}$ | 89.58 $_{\pm 0.12}$ | 56.16 $_{\pm 1.73}$ |
| 2 | 61.66 $_{\pm 2.33}$ | 97.16 $_{\pm 0.28}$ | 67.08 $_{\pm 0.86}$ | 93.13 $_{\pm 0.15}$ | 58.86 $_{\pm 0.35}$ |
| 3 | 60.44 $_{\pm 0.23}$ | 96.82 $_{\pm 0.16}$ | 64.10 $_{\pm 0.86}$ | 90.10 $_{\pm 0.65}$ | 51.39 $_{\pm 1.21}$ |
| 4 | 58.22 $_{\pm 3.39}$ | 96.71 $_{\pm 0.61}$ | 66.11 $_{\pm 0.82}$ | 87.51 $_{\pm 0.52}$ | 50.93 $_{\pm 0.12}$ |

## 4.3 Further Analysis

In this subsection, we conduct additional qualitative experiments to further analyze the behaviors of the deep model trained by f-VAT in the semi-supervised time series classification.

**Perturbation Visualization** Fig. 2 visualizes various virtual perturbations on NATOPS with label ratio $\alpha = 0.4$. The color line represents the original time series data, the color line adds adversarial perturbation generated by the original VAT, and the color line adds our proposed functional adversarial perturbation by f-VAT. As shown in Fig. 2, the functional adversarial perturbation closely aligns with the original sample across all key time steps and only adds subtle fluctuations to non-critical regions, while the original VAT's perturbation introduces large spikes that disrupt underlying trend structure. These observations show that functional adversarial perturbations

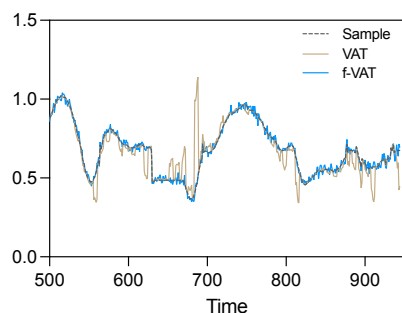

Figure 2: Visualization of perturbations.

can preserve trend information without introducing anomalous patterns, facilitating deep models to achieve smoother predictive distributions and more stable convergence.

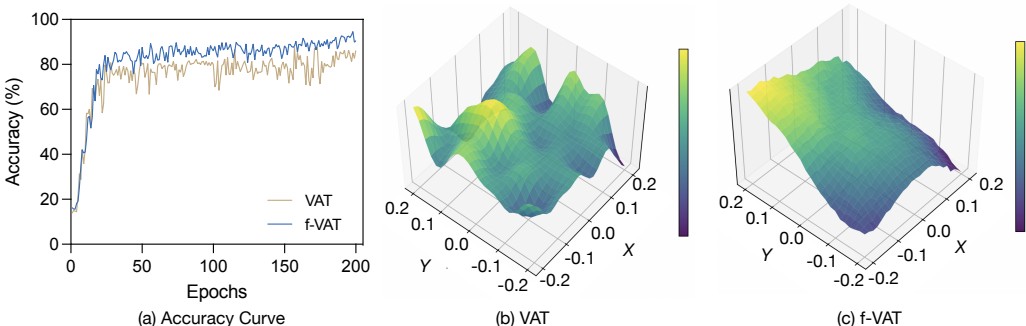

(a) Accuracy Curve  (b) VAT  (c) f-VAT

Figure 3: Visualization of performance curve (left) and the loss landscape of deep models trained by VAT (mid) and f-VAT (right) on the UWave dataset with label ratio $\alpha = 0.4$. The landscape of VAT contains many local minima, trapping the model in suboptimal solutions. The landscape of f-VAT is significantly smoother due to the adversarial perturbation in function space. Thus, f-VAT achieves faster and more stable convergence and better final results (left).

**Loss Landscapes** To validate the smoother predictive distribution offered by the functional adversarial perturbation in Sobolev space and demonstrate that f-VAT leads to easier optimization, we visualize the loss landscape of deep model trained by f-VAT and VAT using "filter normalization" [27] on NATOPS with label ratio $\alpha = 0.4$. From Fig. 3(mid), we observe that the loss landscape of VAT is highly chaotic, which causes the model to be easily trapped in suboptimal minima, leading to an unstable training curve. In contrast, deep models trained by f-VAT (as shown in Fig. 3 (right)) enjoy a significantly smoother loss landscape, where the functional adversarial perturbation in the Sobolev space preserves low-frequency trend information to provide more consistent gradient directions.

Consequently, f-VAT facilitates deep models for faster and more stable convergence with better final performance, as demonstrated in Fig. 3 (left).

**Feature Importance Analysis** To further analyze the behaviors of deep models trained by f-VAT in semi-supervised settings, Fig. 4 visualizes the gradient-based feature importance map [34] on NATOPS with different label ratios. The sample corresponds to "Fold wings" action, whose feature importance significantly changes within the time interval $[25, 35]$. These observations show that as supervision signals increase, deep model trained by f-VAT captures essential "shapelets" (i.e., "elbow's movement to the right") [23] and reduces its reliance on irrelevant features (i.e., feature 2 "shoulder rotation"), which aligns with existing knowledge that "Fold wings" action primarily relies on "elbow movement" with less reliance on "shoulder notation" [37]. This qualitative analysis validates the importance of capturing functional trend information within time series data and demonstrates that f-VAT is a highly effective and efficient method for this purpose.

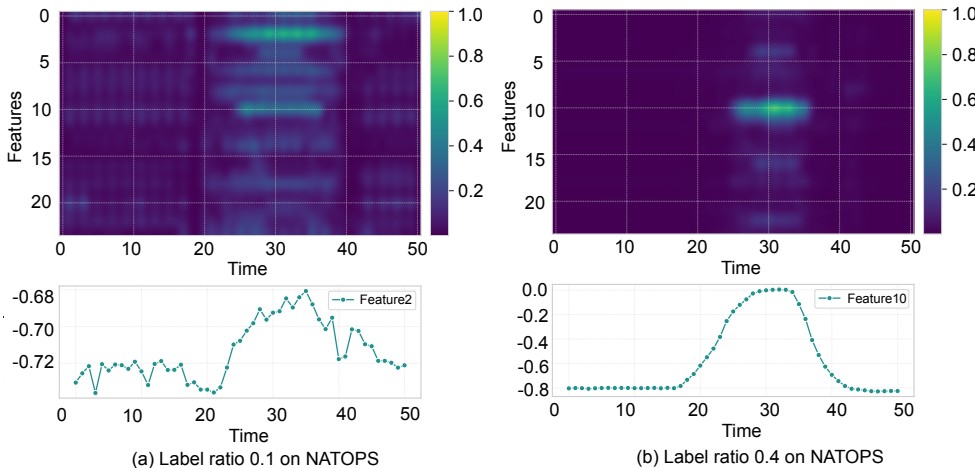

Figure 4: As the label ratio increases, the deep model trained by f-VAT ignores less important features (e.g., feature 2) and captures more critical regions (e.g., feature 10).

## 5 Limitations and Future Directions

In this paper, we propose functional Virtual Adversarial Training (f-VAT), which incorporates the functional structure of data into perturbations. Considering semi-supervised time series classification, we provide both theoretical insights and extensive empirical results showing the superiority of the proposed f-VAT method. We believe that our f-VAT method can serve as a general framework for semi-supervised learning methods with functional data.

Nevertheless, there are still some limitations in this work. We follow the implementations of representative semi-supervised time series classification settings, which primarily train backbone models from scratch, rather than fine-tuning them based on pretrained weights. We consider examining the effectiveness of pre-trained backbones as a potential future work.

Another future direction is to explore and design different adversarial perturbation norm for other functional models. For example, for data with known seasonality or periodic patterns, we can design a norm that additionally penalizes the non-periodicity of the perturbation to align with the periodic patterns, which could be done using the Fourier domain decomposition. We believe that investigating additional structure of the time series would further enhance the performance of the f-VAT method. Also, if the functional data lie on a manifold, one can introduce norm induced by the manifold structure to better align the perturbations. These further exploration would shed the light on the development of semi-supervised learning methods.

## Acknowledgments and Disclosure of Funding

We would like to thank the anonymous reviewers and area chairs for their valuable comments and suggestions.

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

# A  Duality Characterization of VAT Loss in General Spaces

Let us first introduce some preliminaries on the duality with respect to $L^2$. Let $T$ be an interval in $\mathbb{R}$ and we denote by $L^2(T)$ the space of square-integrable functions on $T$ with the inner product

$$\langle u, v \rangle_{L^2(T)} = \int_T u(x)v(x)\mathrm{d}x.$$

Let $E$ be a Banach space with norm $\|\cdot\|_E$. Let $E^*$ be the dual space of $E$, which consists of all continuous linear functionals on $E$. For $f \in E^*$ and $u \in E$, we denote by $\langle f, u \rangle_{E^*, E} = f(u)$ the duality pairing between $f$ and $u$. Suppose that $E$ is either densely embedded in $L^2(T)$ or $L^2(T)$ is densely embedded in $E$. We will discuss the relationship between the dual space $E^*$ and the space $L^2(T)$. We denote by $\|\cdot\|$ the operator norm.

**The case $E \hookrightarrow L^2$.**  Let $i_{E \to L^2} : E \to L^2$ be the embedding. Suppose that $\|u\|_{L^2} \le K\|u\|_E$ For $v \in L^2$ and $u \in E$, we define the functional

$$\phi_v(u) = \langle i_{E \to L^2}(u), v \rangle_{L^2}.$$

Then, we find that

$$|\phi_v(u)| = |\langle i_{E \to L^2}(u), v \rangle_{L^2}| \le \|i_{E \to L^2}(u)\|_{L^2}\|v\|_{L^2} \le \|i_{E \to L^2}\|\|u\|_E\|v\|_{L^2},$$

so

$$\|\phi_v\|_{E^*} = \sup_{\|u\|_E \le 1} |\phi_v(u)| \le \|i_{E \to L^2}\|\|v\|_{L^2}.$$

This shows that $\phi_v \in E^*$. Moreover, suppose $\phi_v \in E^*$ is identically zero. Then, for all $u \in E$, we have $\phi_v(u) = \langle i_{E \to L^2}(u), v \rangle_{L^2} = 0$. But since the range of $i_{E \to L^2}$ is dense in $L^2$, we must have $v = 0$. Therefore, $v \mapsto \phi_v$ defines an embedding $L^2 \hookrightarrow E^*$, which we denote by $i_{L^2 \to E^*}$. In addition, we also have

$$\langle v, i_{E \to L^2}(u) \rangle_{L^2} = \langle i_{L^2 \to E^*}(v), u \rangle_{E^*, E}, \quad \text{namely,} \quad i_{E \to L^2}^* = i_{L^2 \to E^*}.$$

If we view the embeddings as inclusions, we have $E \subset L^2 \subset E^*$.

**The case $L^2 \hookrightarrow E$.**  Let $i_{L^2 \to E} : L^2 \to E$ be the embedding. For an element $\phi \in E^*$, $\phi \circ i_{L^2 \to E}$ also defines a continuous functional on $L^2$. Consequently, from the Riesz representation theorem, there exists a unique element $v_\phi \in L^2$ such that $(\phi \circ i_{L^2 \to E})(u) = \langle u, v_\phi \rangle_{L^2}$. Moreover,

$$\|v_\phi\|_{L^2} = \sup_{\|u\|_{L^2} \le 1} |(\phi \circ i_{L^2 \to E})(u)| = \sup_{\|u\|_{L^2} \le 1} |\phi(i_{L^2 \to E}(u))| \le \|\phi\|_{E^*}\|i_{L^2 \to E}\|\|u\|_{L^2}.$$

Therefore, the mapping $\phi \mapsto v_\phi$ is a bounded linear operator from $E^*$ to $L^2$, which we denote by $i_{E^* \to L^2}$. Furthermore, if $v_\phi$ is identically zero, then $\phi$ must also be zero by the density of the range of $i_{L^2 \to E}$. Consequently, $i_{E^* \to L^2}$ is an embedding. Similarly, we have

$$\langle \phi, i_{L^2 \to E}(u) \rangle_{E^*, E} = \langle i_{E^* \to L^2}(\phi), u \rangle_{L^2}, \quad \text{namely,} \quad i_{L^2 \to E}^* = i_{E^* \to L^2}.$$

Viewing the embeddings as inclusions, we have $E^* \subset L^2 \subset E$.

## A.1  The Duality Characterization of VAT Loss

With the above notations, let us give a more rigorous statement of the theorem. When $E \hookrightarrow L^2$, we define the VAT loss as

$$\mathcal{L}^{\text{VA}}(x_0, \epsilon; f) = \sup_{r = i_{E \to L^2}(r_0) \in L^2 : \|r_0\|_E \le \epsilon} |f(x_0) - f(x_0 + r)|^2, \tag{12}$$

while when $L^2 \hookrightarrow E$, we define the VAT loss as

$$\mathcal{L}^{\text{VA}}(x_0, \epsilon; f) = \sup_{r \in L^2 : \|i_{L^2 \to E}(r)\|_E \le \epsilon} |f(x_0) - f(x_0 + r)|^2, \tag{13}$$

Let us give a more rigorous statement of the theorem.

**Theorem A.1.** *Let $f(x) = \langle \beta, x \rangle_{L^2(T)}$ be a functional linear model with $\beta \in L^2$.*

- *For the case $E \hookrightarrow L^2$, the loss Eq. (13) is equivalent to*

$$\mathcal{L}^{VA}(x_0, \epsilon; f) = \epsilon^2 \|i_{L^2 \to E^*}(\beta)\|_{E^*}^2. \tag{14}$$

- *For the case $L^2 \hookrightarrow E$, the loss Eq. (12) is equivalent to*

$$\mathcal{L}^{VA}(x_0, \epsilon; f) = \begin{cases} \epsilon^2 \|\beta_0\|_{E^*}^2, & \text{if } \beta = i_{E^* \to L^2}(\beta_0) \text{ for some } \beta_0 \in E^*, \\ \infty, & \text{otherwise.} \end{cases} \tag{15}$$

*If we view the embeddings as inclusions, we unify the two cases as*

$$\mathcal{L}^{VA}(x_0, \epsilon; f) = \epsilon^2 \|\beta\|_{E^*}^2. \tag{16}$$

*Proof.* Let $E_0$ be the set over which the supremum is taken in Eq. (13) and Eq. (12). Then, the VAT loss writes

$$\mathcal{L}^{VA}(x_0, \epsilon; f) = \sup_{r \in E_0} |\langle \beta, x_0 \rangle - \langle \beta, x_0 + r \rangle|^2 = \sup_{r \in E_0} \langle \beta, r \rangle_{L^2(T)}^2$$

Let us consider the two cases of separately.

**The case $E \hookrightarrow L^2$.** Using the relation between the dual spaces, we have

$$\mathcal{L}^{VA}(x_0, \epsilon; f) = \sup_{r = i_{E \to L^2}(r_0) \in L^2 : \|r_0\|_E \leq \epsilon} \langle \beta, r \rangle_{L^2(T)}^2$$

$$= \sup_{r_0 \in E : \|r_0\|_E \leq \epsilon} \langle \beta, i_{E \to L^2}(r_0) \rangle_{L^2(T)}^2$$

$$= \sup_{r_0 \in E : \|r_0\|_E \leq \epsilon} \langle i_{L^2 \to E^*}(\beta), r_0 \rangle_{E^*, E}^2$$

$$= \epsilon^2 \|i_{L^2 \to E^*}(\beta)\|_{E^*}^2.$$

**The case $L^2 \hookrightarrow E$.** In this case, we have

$$\mathcal{L}^{VA}(x_0, \epsilon; f) = \sup_{r \in L^2 : \|i_{L^2 \to E}(r)\|_E \leq \epsilon} \langle \beta, r \rangle_{L^2(T)}^2.$$

If $\beta = i_{E^* \to L^2}(\beta_0)$ for some $\beta_0 \in E^*$, we have

$$\mathcal{L}^{VA}(x_0, \epsilon; f) = \sup_{r \in L^2 : \|i_{L^2 \to E}(r)\|_E \leq \epsilon} \langle i_{E^* \to L^2}(\beta_0), r \rangle_{L^2(T)}^2$$

$$= \sup_{r \in L^2 : \|i_{L^2 \to E}(r)\|_E \leq \epsilon} \langle \beta_0, i_{L^2 \to E}(r) \rangle_{E^*, E}^2$$

$$= \sup_{r_1 \in E : \|r_1\|_E \leq \epsilon} \langle \beta_0, r_1 \rangle_{E^*, E}^2$$

$$= \epsilon^2 \|\beta_0\|_{E^*}^2.$$

On the other hand, if $\mathcal{L}^{VA}(x_0, \epsilon; f)$ is finite, then we can define a function on $\mathrm{Ran}\, i_{L^2 \to E}$ that

$$\phi(r_1) = \langle \beta, r \rangle_{L^2(T)}, \quad r_1 = i_{L^2 \to E}(r),$$

which is well-defined since $i_{L^2 \to E}$ is injective. It is easy to verify that $\phi$ is linear. Moreover,

$$\epsilon^2 \|\phi\|_{E^*}^2 = \sup_{r_1 \in E : \|r_1\|_E \leq \epsilon} |\phi(r_1)|^2 = \sup_{r \in L^2 : \|i_{L^2 \to E}(r)\|_E \leq \epsilon} \langle \beta, r \rangle_{L^2(T)}^2 = \mathcal{L}^{VA}(x_0, \epsilon; f) < \infty.$$

Hence, we have $\phi \in E^*$. Moreover, the definition of $\phi$ and the duality relation gives

$$\langle \beta, r \rangle_{L^2(T)} = \phi(r_1) = \langle \phi, r_1 \rangle_{E^*, E} = \langle \phi, i_{L^2 \to E}(r) \rangle_{E^*, E} = \langle i_{E^* \to L^2}(\phi), r \rangle_{L^2(T)},$$

which implies that $\beta = i_{E^* \to L^2}(\phi)$. Consequently, $\mathcal{L}^{VA}(x_0, \epsilon; f) < \infty$ if and only if $\beta \in \mathrm{Ran}\, i_{E^* \to L^2}$ and we conclude the theorem.

$\square$

## A.2 Proof of Theorem 3.4

Let us first recall that Fréchet differentiability of $f$ at $x_0$ means that there exists a continuous linear functional $\nabla f(x_0) \in L^2(T)$ such that

$$f(x_0 + r) = f(x_0) + \langle \nabla f(x_0), r \rangle_{L^2(T)} + o(\|r\|_{L^2(T)}) \quad \text{as } \|r\|_{L^2(T)} \to 0.$$

Moreover, if $L^2 \to E$, we say that the Fréchet differentiability holds under the $E$-norm if $\nabla f(x_0) = i_{E^* \to L^2}(\beta_0)$ for some $\beta_0 \in E^*$ and for any $r \in L^2$, we have

$$f(x_0 + r) = f(x_0) + \langle \nabla f(x_0), r \rangle_{L^2(T)} + o(\|i_{L^2 \to E}(r)\|_E) \quad \text{as } \|r\|_E \to 0.$$

First, we give a rigorous version of Theorem 3.4 in the following.

**Theorem A.2** (Theorem 3.4 restated). *Let $f : L^2(T) \to \mathbb{R}$ be Fréchet differentiable at $x_0 \in L^2(T)$ with gradient $\nabla f(x_0) \in L^2(T)$.*

- *If $E \hookrightarrow L^2(T)$, then*

$$\lim_{\epsilon \to 0^+} \epsilon^{-2} \mathcal{L}^{VA}(x_0, \epsilon; f) = \|i_{L^2 \to E^*}(\nabla f(x_0))\|^2_{E^*}.$$

- *If $L^2(T) \hookrightarrow E$, assume further that $\nabla f(x_0) = i_{E^* \to L^2}(\beta_0)$ for some $\beta_0 \in E^*$ and the Fréchet differentiability holds under under $E$-norm, we have*

$$\lim_{\epsilon \to 0^+} \epsilon^{-2} \mathcal{L}^{VA}(x_0, \epsilon; f) = \|\beta_0\|^2_{E^*}.$$

*Proof.* We consider the two cases separately.

**The case $E \hookrightarrow L^2(T)$** In this setting, the dual embedding $i_{L^2 \to E^*} : L^2(T) \to E^*$ satisfies

$$\langle i_{L^2 \to E^*}(v), u \rangle_{E^*, E} = \langle v, u \rangle_{L^2(T)} \quad \text{for all } v \in L^2(T), \, u \in E.$$

Combining it with the Fréchet differentiability of $f$ at $x_0$, If $r = i_{E \to L^2}(r_0)$, we have

$$\begin{aligned}
f(x_0 + r) &= f(x_0) + \langle \nabla f(x_0), r \rangle_{L^2(T)} + o(\|r\|_{L^2(T)}) \\
&= f(x_0) + \langle i_{L^2 \to E^*}(\nabla f(x_0)), r_0 \rangle_{E^*, E} + o(\|i_{E \to L^2}(r_0)\|_{L^2(T)}) \\
&= f(x_0) + \langle i_{L^2 \to E^*}(\nabla f(x_0)), r_0 \rangle_{E^*, E} + o(\|r_0\|_E),
\end{aligned}$$

Therefore, recalling that the VAT loss is given by Eq. (12), we have

$$\begin{aligned}
\mathcal{L}^{VA}(x_0, \epsilon; f) &= \sup_{r_0 \in E : \|r_0\|_E \leq \epsilon} \left( \langle i_{L^2 \to E^*}(\nabla f(x_0)), r_0 \rangle_{E^*, E} + o(\|r_0\|_E) \right)^2 \\
&= \sup_{r_0 \in E : \|r_0\|_E \leq \epsilon} \left( \langle i_{L^2 \to E^*}(\nabla f(x_0)), r_0 \rangle_{E^*, E} + o(\epsilon) \right)^2
\end{aligned}$$

Consequently,

$$\begin{aligned}
\epsilon^{-2} \mathcal{L}^{VA}(x_0, \epsilon; f) &= \sup_{r_0 \in E : \|r_0\|_E \leq \epsilon} \left( \epsilon^{-1} \langle i_{L^2 \to E^*}(\nabla f(x_0)), r_0 \rangle_{E^*, E} + o(1) \right)^2 \\
&= \sup_{s_0 \in E : \|s_0\|_E \leq 1} \left( \langle i_{L^2 \to E^*}(\nabla f(x_0)), s_0 \rangle_{E^*, E} + o(1) \right)^2 \\
&= \sup_{s_0 \in E : \|s_0\|_E \leq 1} \left( \langle i_{L^2 \to E^*}(\nabla f(x_0)), s_0 \rangle_{E^*, E} \right)^2 + o(1) \\
&= \|i_{L^2 \to E^*}(\nabla f(x_0))\|^2_{E^*} + o(1).
\end{aligned}$$

**The case** $L^2(T) \hookrightarrow E$    In this case, let us recall that the dual embedding $i_{E^* \to L^2} : E^* \to L^2(T)$ satisfies

$$\langle i_{E^* \to L^2}(\phi), u \rangle_{L^2(T)} = \langle \phi, u \rangle_{E^*, E} \quad \text{for all } \phi \in E^*, u \in L^2(T).$$

Thus, since $\nabla f(x_0) = i_{E^* \to L^2}(\beta_0)$ for some $\beta_0 \in E^*$, using the Fréchet differentiability of $f$ at $x_0$ under the $E$-norm, we have

$$
\begin{aligned}
f(x_0 + r) &= f(x_0) + \langle \nabla f(x_0), r \rangle_{L^2(T)} + o(\|i_{L^2 \to E}(r)\|_E) \\
&= f(x_0) + \langle i_{E^* \to L^2}(\beta_0), r \rangle_{L^2(T)} + o(\|i_{L^2 \to E}(r)\|_E) \\
&= f(x_0) + \langle \beta_0, i_{L^2 \to E}(r) \rangle_{E^*, E} + o(\|i_{L^2 \to E}(r)\|_E).
\end{aligned}
$$

Recall the VAT loss is given by Eq. (13), we have

$$
\begin{aligned}
\mathcal{L}^{\mathrm{VA}}(x_0, \epsilon; f) &= \sup_{r \in L^2 : \|i_{L^2 \to E}(r)\|_E \leq \epsilon} \left| \langle \beta_0, i_{L^2 \to E}(r) \rangle_{E^*, E} + o(\|i_{L^2 \to E}(r)\|_E) \right|^2 \\
&= \sup_{r_1 \in E, \|r_1\|_E \leq \epsilon} \left| \langle \beta_0, r_1 \rangle_{E^*, E} + o(\|r_1\|_E) \right|^2,
\end{aligned}
$$

where we use the density of $i_{L^2 \to E}$. Therefore,

$$
\begin{aligned}
\epsilon^{-2} \mathcal{L}^{\mathrm{VA}}(x_0, \epsilon; f) &= \sup_{r_1 \in E : \|r_1\|_E \leq \epsilon} \left( \epsilon^{-1} \langle \beta_0, r_1 \rangle_{E^*, E} + o(1) \right)^2 \\
&= \sup_{s_1 \in E : \|s_1\|_E \leq 1} \left( \langle \beta_0, s_1 \rangle_{E^*, E} + o(1) \right)^2 \\
&= \|\beta_0\|_{E^*}^2 + o(1).
\end{aligned}
$$

$\square$

### A.3   Proof of Proposition 3.2

Since $f(x)$ is continuously differentiable, we have

$$f(x_0) - f(x_0 + r) = \langle \nabla f(x_0), r \rangle + o(\|r\|_2).$$

Noticing that $\|r\|_\Sigma \leq \epsilon$ implies $\|r\|_2 \leq \epsilon \left\| \Sigma^{-\frac{1}{2}} \right\|_\infty$, we have

$$
\begin{aligned}
\mathcal{L}^{\mathrm{VA}}(x_0, \epsilon; f) &= \sup_{r : \|r\|_\Sigma \leq \epsilon} |\langle \beta, x_0 \rangle - \langle \beta, x_0 + r \rangle|^2 \\
&= \sup_{r : \|r\|_\Sigma \leq \epsilon} (\langle \nabla f(x_0), r \rangle + o(\|r\|_2))^2 \\
&= \sup_{r : \|r\|_\Sigma \leq \epsilon} (\langle \nabla f(x_0), r \rangle + o(\epsilon))^2.
\end{aligned}
$$

Therefore, when $\epsilon \to 0^+$, we obtain that

$$
\begin{aligned}
\epsilon^{-2} \mathcal{L}^{\mathrm{VA}}(x_0, \epsilon; f) &= \sup_{r : \|r\|_\Sigma \leq \epsilon} \left( \epsilon^{-2} \langle \nabla f(x_0), r \rangle^2 + o(1) \right) \\
&\to \sup_{r : \|r\|_\Sigma \leq \epsilon} \epsilon^{-2} \langle \nabla f(x_0), r \rangle^2 \\
&= \|\nabla f(x_0)\|_{\Sigma^{-1}}^2.
\end{aligned}
$$

# B   Sobolev Spaces

Sobolev spaces are a cornerstone of functional analysis, providing a framework to quantify the smoothness of functions and distributions. This section outlines their construction, duality properties, and practical computation in discrete settings, which are particularly relevant for introducing dual loss in the VAT method. We refer to books [1, 9] for details.

## B.1   Construction of Sobolev Spaces via Fourier Transforms

Sobolev spaces $H^s(\mathbb{R}^d)$ for $s \in \mathbb{R}$ are defined using Fourier transforms, offering a unified approach across all real smoothness indices. Let $\mathcal{S}(\mathbb{R}^d)$ denote the Schwartz space of smooth, rapidly decaying functions on $\mathbb{R}^d$, and $\mathcal{S}'(\mathbb{R}^d)$ its dual, the space of tempered distributions. Moreover, we can define the Fourier transformation $\hat{u}(\xi) = \mathcal{F}u(\xi) = \int_{\mathbb{R}^d} u(x)e^{-i\xi \cdot x}\,dx$ of $u \in \mathcal{S}(\mathbb{R}^d)$ and extend the Fourier transformation to $\mathcal{S}'(\mathbb{R}^d)$.

For $s \in \mathbb{R}$, the Sobolev space $H^s(\mathbb{R}^d)$ is defined as the space of tempered distributions $u$ such that

$$(1 + |\xi|^2)^{s/2}|\hat{u}(\xi)| \in L^2(\mathbb{R}^d) \tag{17}$$

with the norm and inner product

$$\|u\|_{H^s} = \left(\int_{\mathbb{R}^d} (1 + |\xi|^2)^s |\hat{u}(\xi)|^2\,d\xi\right)^{1/2}, \quad \langle u, v\rangle_{H^s} = \int_{\mathbb{R}^d} (1 + |\xi|^2)^s \hat{u}(\xi)\hat{v}(\xi)\,d\xi. \tag{18}$$

Moreover, $H^s(\mathbb{R}^d)$ is also equivalent to the completion of $\mathcal{S}(\mathbb{R}^d)$ with respect to the $H^s$ norm. The weight $(1 + |\xi|^2)^s$ modulates the contribution of different frequencies: for $s > 0$, it penalizes high-frequency components, enforcing smoothness; for $s < 0$, it emphasizes them, allowing rougher distributions; and for $s = 0$, it reduces to the $L^2(\mathbb{R}^d)$ norm, since $H^0(\mathbb{R}^d) = L^2(\mathbb{R}^d)$. It is well known that the Sobolev space $H^s(\mathbb{R}^d)$ is a Hilbert space under the inner-product $\langle \cdot, \cdot\rangle_{H^s}$.

For positive integer $s$, $H^s(\mathbb{R}^d)$ coincides with the space of $L^2$ functions whose weak derivatives up to order $s$ are also in $L^2(\mathbb{R}^d)$. For negative $s$, $H^s(\mathbb{R}^d)$ includes distributions that lack the integrability of $L^2$ functions but are constrained by the decay of their Fourier coefficients.

In applications like time series, we often consider functions on a bounded domain, such as $T = [0, 1]$. For domains $\Omega \subseteq \mathbb{R}^d$, $H^s(\Omega)$ can be defined as the restriction of functions from $H^s(\mathbb{R}^d)$ to $\Omega$, with the norm being the infimum of the $H^s(\mathbb{R}^d)$ norm over all extensions, or as the completion of $C^\infty(\Omega)$ with respect to the appropriate norm.

## B.2   Duality and Embeddings of Sobolev Spaces

Sobolev spaces exhibit a rich duality structure, which is essential for understanding their properties and applications. For $s \in \mathbb{R}$, the dual space of $H^s(\mathbb{R}^d)$ with respect to the $L^2(\mathbb{R}^d)$ inner product is $H^{-s}(\mathbb{R}^d)$. That is, any continuous linear functional $\ell \in (H^s(\mathbb{R}^d))^*$ can be represented as:

$$\ell(u) = \langle v, u\rangle_{L^2} = \int_{\mathbb{R}^d} v(x)\overline{u(x)}\,dx, \tag{19}$$

for some $v \in H^{-s}(\mathbb{R}^d)$, with the norm equivalence:

$$\|\ell\|_{(H^s)^*} = \|v\|_{H^{-s}}. \tag{20}$$

This duality arises because the pairing $\langle v, u\rangle_{L^2}$ is well-defined when $u \in H^s$ and $v \in H^{-s}$, as the Fourier transform ensures that $(1 + |\xi|^2)^{-s/2}\hat{v}(\xi)$ and $(1 + |\xi|^2)^{s/2}\hat{u}(\xi)$ yield a product in $L^2(\mathbb{R}^d)$.

Sobolev spaces also satisfy embedding theorems, which relate them to other function spaces based on smoothness. A key result is the Sobolev embedding theorem: for $s > d/2$, $H^s(\mathbb{R}^d)$ embeds continuously into $C_b(\mathbb{R}^d)$, the space of bounded continuous functions, with:

$$\|u\|_{C_b} \le C\|u\|_{H^s}, \tag{21}$$

for some constant $C$. This embedding implies that functions in $H^s$ with sufficiently large $s$ are not only continuous but also bounded, a property useful for ensuring regularity in optimization problems.

More generally, for $s_1 > s_2$, $H^{s_1}(\mathbb{R}^d) \hookrightarrow H^{s_2}(\mathbb{R}^d)$, reflecting that higher $s$ corresponds to greater smoothness.

These properties—duality and embeddings—are critical in applications like Virtual Adversarial Training (VAT), where the dual norm $\|\nabla f(x_0)\|_{H^{-s}}$ quantifies model sensitivity to perturbations $r$ constrained by $\|r\|_{H^s} \leq \epsilon$, and embeddings ensure perturbations maintain desirable regularity.

## B.3 Fractional Powers of the Laplacian and Sobolev Norms

The fractional power of the Laplacian provides an operator-theoretic perspective on Sobolev spaces, unifying the definition of $H^s(\mathbb{R}^d)$ across all $s \in \mathbb{R}$. The Laplacian $-\Delta$ is a positive, self-adjoint operator on $L^2(\mathbb{R}^d)$, and its fractional power $(-\Delta)^\alpha$ for $\alpha \in \mathbb{R}$ is defined via the Fourier transform:

$$\mathcal{F}[(-\Delta)^\alpha u](\xi) = |\xi|^{2\alpha}\hat{u}(\xi),$$

where $\hat{u}(\xi) = \mathcal{F}u(\xi)$ is the Fourier transform of $u$. However, the Sobolev norm incorporates a shifted operator, $I - \Delta$, to ensure positivity and handle low frequencies effectively. Specifically, the fractional power $(I - \Delta)^{s/2}$ satisfies:

$$\mathcal{F}[(I - \Delta)^{s/2}u](\xi) = (1 + |\xi|^2)^{s/2}\hat{u}(\xi),$$

so that:

$$\|u\|_{H^s} = \|(I - \Delta)^{s/2}u\|_{L^2}.$$

This equivalence follows from:

$$\|(I-\Delta)^{s/2}u\|_{L^2}^2 = \int_{\mathbb{R}^d} |(1+|\xi|^2)^{s/2}\hat{u}(\xi)|^2 \, d\xi = \int_{\mathbb{R}^d} (1+|\xi|^2)^s |\hat{u}(\xi)|^2 \, d\xi = \|u\|_{H^s}^2.$$

For $s > 0$, $(I - \Delta)^{s/2}$ acts as a differential operator of order $s$, penalizing high-frequency oscillations and enforcing smoothness. For example, when $s = 2$, $(I - \Delta)u = u - \Delta u$, and the norm $\|u - \Delta u\|_{L^2}$ measures both the function and its second derivatives. For $s < 0$, $(I - \Delta)^{s/2}$ is a smoothing operator, and $H^s$ includes distributions whose images under $(I - \Delta)^{-s/2}$ are in $L^2$.

The dual norm in $H^{-s}(\mathbb{R}^d)$ relates to the inverse fractional power. For $v \in H^{-s}$, we have:

$$\|v\|_{H^{-s}} = \|(I - \Delta)^{-s/2}v\|_{L^2},$$

since:

$$\|v\|_{H^{-s}} = \sup_{\|u\|_{H^s} \leq 1} |\langle v, u\rangle_{L^2}| = \sup_{\|(I-\Delta)^{s/2}w\|_{L^2} \leq 1} |\langle v, (I - \Delta)^{s/2}w\rangle_{L^2}|,$$

and setting $w = (I - \Delta)^{-s/2}u$ yields the result via the self-adjointness of $I - \Delta$. This operator formulation is particularly useful in discrete settings, as it translates directly to matrix powers, as discussed previously.

The fractional Laplacian $(-\Delta)^s$ itself (without the identity shift) is also of interest, with norm:

$$\|(-\Delta)^{s/2}u\|_{L^2}^2 = \int_{\mathbb{R}^d} |\xi|^{2s}|\hat{u}(\xi)|^2 \, d\xi.$$

While this norm emphasizes derivative behavior alone, $I - \Delta$ ensures a baseline $L^2$ contribution, making $H^s$ norms more robust for small $s$ or low frequencies. Choosing between $(I - \Delta)^{s/2}$, $(-\Delta)^{s/2}$ or $(\alpha I - \Delta)^{s/2}$ for perturbation constraints can tailor the robustness profile: the former balances function magnitude and smoothness, while the latter focuses purely on derivative control.

## B.4 Discrete Computation via Spectral Methods

In practical applications such as time series analysis, data are represented as discrete vectors in $\mathbb{R}^N$. To compute Sobolev norms in this discrete setting, we employ spectral methods that approximate the continuous operators while preserving their spectral properties, ensuring consistency with the operator perspective introduced in the discussion on fractional powers and Sobolev norms.

We recall that in the continuous case, the Sobolev norm of a function $u \in H^s(\mathbb{R}^d)$ is defined using the fractional power of the operator $I - \Delta$, where $\Delta$ is the Laplacian:

$$\|u\|_{H^s} = \|(I - \Delta)^{s/2} u\|_{L^2}.$$

This can be expressed in the Fourier domain as:

$$\|u\|_{H^s}^2 = \int_{\mathbb{R}^d} (1 + |\xi|^2)^s |\hat{u}(\xi)|^2 \, d\xi,$$

where $\hat{u}$ is the Fourier transform of $u$, and $1 + |\xi|^2$ arises from the eigenvalues of $I - \Delta$. In the discrete setting, we aim to approximate this norm for a vector $r \in \mathbb{R}^N$, representing a time series sampled at $N$ equally spaced points. For simplicity, we assume periodic boundary conditions, which are common in time series analysis and allow for efficient computation via the Fast Fourier Transform (FFT).

### B.4.1 Discrete Laplacian and Sobolev Norm

The discrete Laplacian $L_N \in \mathbb{R}^{n \times n}$ is constructed using second-order finite differences. For a uniform grid with $n$ points, $L_N$ is a tridiagonal matrix defined as:

$$L_N = \begin{bmatrix} 2 & -1 & 0 & \cdots & 0 & 0 \\ -1 & 2 & -1 & \cdots & 0 & 0 \\ 0 & 0 & \cdots & -1 & 2 & -1 \\ 0 & 0 & \cdots & 0 & -1 & 2 \end{bmatrix}.$$

This matrix $L_N$ approximates the negative Laplacian $-\Delta$ via twice difference, which is a positive semi-definite matrix. Thus, $I_N + L_N$, where $I_N$ is the $N \times N$ identity matrix, approximates the operator $I - \Delta$. Consequently, the squared $H^s$ norm of the discrete vector $r_N \in \mathbb{R}^n$ is then approximated as:

$$\|r\|_{H^s}^2 \approx r_N^\top (I_N + L_N)^s r_N.$$

### B.4.2 Speeding up the Computation under Periodic Boundary Conditions

For many applications of time series, we assume periodic boundary conditions, which allow us to use the discrete Fourier transform (DFT) to compute the Sobolev norm efficiently. Under periodic boundary conditions, we take the discrete Laplacian as

$$\bar{L}_N = \begin{bmatrix} 2 & -1 & 0 & \cdots & 0 & -1 \\ -1 & 2 & -1 & \cdots & 0 & 0 \\ 0 & 0 & \cdots & -1 & 2 & -1 \\ -1 & 0 & \cdots & 0 & -1 & 2 \end{bmatrix},$$

where we note the off-diagonal entries $-1$ at the corners (e.g., $L_{1,N} = -1$, $L_{N,1} = -1$) enforce periodicity. Then, this matrix $\bar{L}_N$ is a circulant matrix, which can be diagonalized using the discrete Fourier transform (DFT). The matrix can be diagonalized by the discrete Fourier transform matrix $F$, defined by:

$$F = (F_{jk})_{j,k=0}^{N-1}, \quad F_{jk} = \frac{1}{\sqrt{N}} e^{-i2\pi jk/N}, \quad j, k = 0, 1, \ldots, N-1,$$

which is unitary (i.e., $F^* F = I_N$, with $F^*$ being the conjugate transpose). Then, the spectral decomposition of $\bar{L}_N$ is given by:

$$\bar{L}_N = F \Lambda F^*,$$

where $\Lambda = \text{diag}(\lambda_0, \lambda_1, \ldots, \lambda_{N-1})$ is a diagonal matrix of eigenvalues. For the discrete Laplacian defined above, the eigenvalues are:

$$\lambda_k = 4 \sin^2\left(\frac{\pi k}{n}\right), \quad k = 0, 1, \ldots, n-1.$$

These eigenvalues are non-negative ($\lambda_k \geq 0$), with $\lambda_0 = 0$ corresponding to the constant mode. Consequently, $I_N + \bar{L}_N$ is decomposed as:

$$I_N + \bar{L}_N = F(I_N + \Lambda)F^*,$$

and the fractional power of the operator is then:

$$(I_N + L_N)^s = F(I_N + \Lambda)^s F^*.$$

Thus, the norm can be computed as:

$$\|r_N\|_{H^s}^2 \approx r_N^\top (I_N + L_N)^s r_N = r_N^\top F(I_N + \Lambda)^s F^* r_N = \sum_{k=0}^{n-1} (1 + \lambda_k)^s |\hat{r}_{n,k}|^2,$$

where $\hat{r}_n = F^* r_N$ is the discrete Fourier transform of $r_N$. Since the fast Fourier transform (FFT) can compute $\hat{r}_n$ in $O(N \log N)$ time and other operations are $O(N)$, the overall complexity of computing the Sobolev norm is $O(N \log N)$, improving efficiency compared to direct matrix-vector multiplication, which would be $O(N^2)$.

### B.4.3 Discrete Laplacian for Non-Uniform Points

In many practical applications, such as time series analysis with irregularly sampled data, the points $t_1 < t_2 < \cdots < t_N$ are not uniformly spaced. In this case, we have to adjust the discrete Laplacian to account for the non-uniform spacing between points.

For a non-uniform grid, let us define the forward and backward step sizes at each point:

$$h_i^+ = t_{i+1} - t_i, \quad i = 1, \ldots, N-1,$$
$$h_i^- = t_i - t_{i-1}, \quad i = 2, \ldots, N,$$

and define $h_1^- = h_1^+$ and $h_N^+ = h_N^-$. We introduce the coefficients

$$a_i = -\frac{2}{h_i^-(h_i^+ + h_i^-)}, \quad c_i = -\frac{2}{h_i^+(h_i^+ + h_i^-)}, \quad b_i = \frac{2}{h_i^+ h_i^-}, \quad i = 2, \ldots, N-1.$$

The entries of the discrete Laplacian $L_N \in \mathbb{R}^{N \times N}$ are given by

$$L_N = \begin{bmatrix} b_1 & c_1 & 0 & \cdots & 0 & 0 \\ a_2 & b_2 & c_2 & \cdots & 0 & 0 \\ 0 & 0 & \cdots & a_{N-1} & b_{N-1} & c_{N-1} \\ 0 & 0 & \cdots & 0 & a_N & b_N \end{bmatrix}.$$

With the adjusted discrete Laplacian, we can compute the Sobolev norm in a similar manner as before.

## C   Experiment Details

### C.1   Dataset Details

We choose several representative datasets as benchmarks, ranging from easy to difficult. These datasets include different properties like sampling rate, allowing for a more comprehensive evaluation of various semi-supervised time series classification methods. Table 6 summarizes the statistics of various datasets. The details of other 30 UCR/UEA datasets can be found in [16].

- **CricketX** [32]. The dataset contains gesture data with position of the X axis collected from accelerometers in 3D space. CricketX includes 12 classes: "Cancel Call", "Dead Ball", "Four", "Last Hour Leg Bye", "No Ball", "One Short", "Out", "Penalty Runs", "Six", "TV Replay". Both the training and test sets contain 390 samples.

- **UWaveGestureLibraryAll (UWave)** [30]. The dataset from the gesture recognition system used by Nokia's search engine collects user-phone interaction motions. It contains 4,478 samples. The training and test sets contain 896 and 3,582 samples, respectively.

- **InsectWingbeatSound (InsectWing)** [12]. The dataset is released by the Computational Entomology group at the University of California Riverside for insect classification. It includes wingbeat audio signals from male and female mosquitoes, different species of flies, and other insects. The training set contains 220 samples and the test set contains 1,980 samples.

- **SelfRegulationSCP2 (SelfReg)** [7]. The University of Tuebingen releases SelfRegulation-SCP2 [7], which contains EEG data with seven columns and 1,152 rows. These sensors record signals of slow cortical potentials from auditory and visual feedback. The training set contains 200 samples, and the test set contains 180 samples.

- **NATOPS** [20]. The AALTD competition releases the NATOPS dataset [20]. These sensors collect data from hands, elbows, wrists, and thumbs. The dataset consists of position coordinates. The six categories represent different actions: "I have to command", "all clear", "not clear", "spread wings", "fold wings", and "lock wings". Both training and test sets contain 180 samples.

- **Heartbeat** [29]. The Heartbeat dataset released by the PhysioNet Challenge 2016 primarily includes heart sound signals from volunteers in clinical or non-clinical environments. The signals are categorized into two classes: "normal" (113 samples) and "abnormal" (296 samples). The sensors are sequentially positioned at the aortic, pulmonic, tricuspid and mitral auscultation sites in patients spanning a broad age range.

Table 6: The statistics of univariate and multivariate datasets in UEA & UCR archive, including three univariate datasets and three multivariate datasets for evaluation.

| Dataset | Samples | Length | Dim | Class |
|---|---|---|---|---|
| CricketX | 780 | 300 | 1 | 12 |
| UWave | 4478 | 948 | 1 | 8 |
| InsectWing | 2200 | 256 | 1 | 11 |
| SelfReg | 380 | 1152 | 7 | 2 |
| NATOPS | 360 | 51 | 24 | 6 |
| Heartbeat | 409 | 405 | 61 | 5 |

### C.2   Hyperparameter Setting

We use stochastic gradient descent with a learning rate of $10^{-3}$. The batch size is set to 64 with a maximum of 300 epochs. Due to the model-agnostic properties of f-VAT, we use an eight-layers Temporal Convolutional Network (TCN) [5] as the backbone architecture to compare with other competitive baselines. We run our experiments on eight NVIDIA A10 GPUs (each with 24 GB memory).

# D Ablation Study

## D.1 Various Architectures

To verify the model-agnostic property of our proposed f-VAT, we select several representative deep architectures with comparable parameter size, including Gated Recurrent Units (GRU), Self-Attention encoder and Temporal Convolutional Network (TCN).

- **Gated Recurrent Unit** (GRU) integrates the output and memory gates to address short-term memory challenges [15]. GRUs are combined with attention mechanisms to better capture trend information within time series data.
- **Self-Attention encoder** (SA) [39]. SA encoder is composed of four self-attention layers and a positional encoding layer, which can effectively extract trend information from modeling time series data.
- **Temporal Convolutional Network** (TCN) is a widely used for sequence modeling. Its causal convolutional operations effectively capture both short-term fluctuations and long-term temporal dependencies [5]. TCN achieves superior performance in various sequence modeling tasks.

We report the mean performance of each architecture over five runs with different random seeds. As shown in Fig. 5, TCN significantly outperforms other deep architectures on NATOPS with various label ratios. These observations show that causal convolution of TCN can effectively capture trend information, thus effectively utilizing unlabeled data to further improve generalization. In this paper, we adopt the 8-layer TCN as our main architecture.

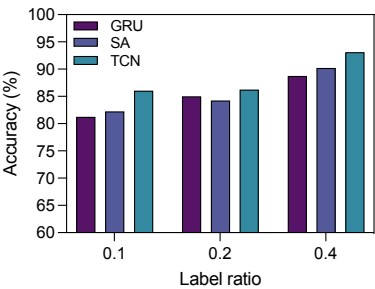

Figure 5: The performance comparison of different architectures on NATOPS with different label ratios. TCN significantly outperforms other architectures in all settings.

Table 7: Comparison of VAT, VAT-step, and f-VAT on five datasets under different label ratios.

| Methods | CricketX | | | UWave | | | InsectWing | | | SelfReg | | | NATOPS | | |
|---|---|---|---|---|---|---|---|---|---|---|---|---|---|---|---|
| | 10% | 20% | 40% | 10% | 20% | 40% | 10% | 20% | 40% | 10% | 20% | 40% | 10% | 20% | 40% |
| VAT | 42.85 ±3.97 | 49.14 ±0.50 | 58.63 ±0.50 | 94.41 ±0.09 | 95.53 ±0.31 | 94.76 ±0.54 | 55.49 ±1.28 | 61.27 ±0.19 | 63.48 ±0.30 | 53.12 ±4.51 | 55.76 ±0.35 | 53.47 ±1.04 | 82.38 ±0.96 | 82.81 ±0.52 | 90.15 ±1.60 |
| VAT-step | 40.47 ±3.84 | 47.85 ±0.33 | 59.30 ±0.36 | 93.78 ±0.67 | 96.09 ±0.32 | 95.86 ±0.55 | 57.45 ±0.21 | 60.92 ±0.48 | 63.61 ±0.58 | 48.61 ±3.27 | 53.67 ±2.79 | 52.75 ±1.80 | 81.33 ±0.96 | 83.02 ±0.12 | 86.58 ±1.18 |
| f-VAT | 49.18 ±1.96 | 57.91 ±3.58 | 68.39 ±2.25 | 94.82 ±0.39 | 96.45 ±0.27 | 97.23 ±0.43 | 58.01 ±1.12 | 61.28 ±1.86 | 64.81 ±1.15 | 59.31 ±3.06 | 61.60 ±1.13 | 64.44 ±3.13 | 86.04 ±1.41 | 86.25 ±1.38 | 93.13 ±0.15 |

To verify the straightforward improvement to VAT may be unsuitable for time series, we design a variant called *VAT-step*, which constrains the magnitude of perturbations at each time step, and conduct extensive experiments across multiple datasets with different label ratios. Table 7 shows that VAT-step only provides marginal performance gains compared to the original VAT in most settings, and even degrades performance on more challenging multivariate datasets (e.g., SelfReg and NATOPS) containing complex temporal structure. This is because clipping the perturbation at each time step struggles to adaptively scale the magnitude on critical regions for prediction (like "shapelets" in Fig. 4), unless the per-step magnitude hyperparameters are carefully tuned. Additionally, VAT-step can easily converge to the "permutation-invariant" [8] local optimum, where arbitrarily reordering time steps can still generate identical perturbations satisfying each step-wise constraint. By contrast, adversarial perturbations generated by *f-VAT*, which incorporate the temporal structure of time series, more efficiently utilize sequential information to improve the smoothness of predictive distributions and the final predictive performance.

# E    More Empirical Results

We randomly sample 30 datasets from UCR/UEA datasets and add a state-of-the-art Class-Aware Temporal and Contextual Contrasting (CA-TCC) [18] within available resources. Tables 8 to 10 show that tVAT consistently outperforms other baselines in all settings.

Table 8: The accuracy and rank of each method across 30 datasets with label ratio $\alpha = 0.1$.

| dataset | SupL | PI | MTL | meanTeacher | SemiTime | TapNet | CA-TCC | f-VAT | SupL_rank | PI_rank | MTL_rank | meanTeacher_rank | SemiTime_rank | TapNet_rank | CA-TCC_rank | f-VAT_rank |
|---|---|---|---|---|---|---|---|---|---|---|---|---|---|---|---|---|
| ACSF1 | 13.75 ±3.75 | 16.40 ±8.60 | 13.28 ±2.34 | 35.94 ±2.82 | 42.97 ±2.51 | 32.82 ±3.12 | 43.75 ±2.64 | 57.29 ±2.61 | 7 | 6 | 8 | 4 | 3 | 5 | 2 | 1 |
| Adiac | 5.98 ±4.02 | 7.54 ±1.74 | 6.06 ±0.26 | 16.73 ±2.68 | 11.19 ±6.19 | 29.04 ±1.11 | 29.25 ±2.88 | 33.94 ±4.20 | 8 | 6 | 7 | 4 | 5 | 3 | 2 | 1 |
| AllGestureWiimoteX | 10.12 ±1.35 | 36.61 ±4.46 | 26.90 ±0.78 | 23.66 ±2.68 | 46.20 ±2.91 | 24.33 ±4.69 | 42.71 ±1.68 | 44.86 ±7.36 | 8 | 4 | 5 | 7 | 1 | 6 | 3 | 2 |
| AllGestureWiimoteY | 12.68 ±1.21 | 33.93 ±7.59 | 20.42 ±0.34 | 20.76 ±2.90 | 51.49 ±2.24 | 32.36 ±3.79 | 44.19 ±1.56 | 55.36 ±1.78 | 8 | 4 | 7 | 6 | 2 | 5 | 3 | 1 |
| AllGestureWiimoteZ | 12.68 ±1.21 | 25.67 ±6.47 | 18.02 ±3.74 | 19.20 ±2.92 | 38.17 ±3.79 | 23.66 ±2.66 | 34.38 ±5.84 | 33.54 ±4.91 | 8 | 4 | 7 | 6 | 1 | 5 | 2 | 3 |
| ArrowHead | 38.10 ±0.10 | 52.34 ±4.22 | 39.61 ±0.96 | 39.84 ±2.90 | 60.78 ±2.34 | 64.84 ±1.56 | 61.67 ±3.06 | 71.25 ±1.41 | 8 | 5 | 7 | 6 | 4 | 2 | 3 | 1 |
| BME | 45.83 ±2.67 | 75.00 ±2.57 | 50.78 ±2.34 | 45.32 ±2.82 | 64.06 ±2.78 | 90.24 ±1.17 | 74.48 ±2.70 | 79.69 ±2.69 | 7 | 3 | 6 | 8 | 5 | 1 | 4 | 2 |
| Beef | 20.00 ±0.10 | 20.00 ±0.10 | 15.00 ±5.00 | 20.00 ±0.10 | 25.00 ±5.00 | 20.00 ±2.75 | 20.00 ±0.10 | 28.00 ±0.10 | 3 | 3 | 8 | 3 | 2 | 3 | 3 | 1 |
| BeetleFly | 50.00 ±0.10 | 62.50 ±2.86 | 50.00 ±0.10 | 50.00 ±0.10 | 50.00 ±0.10 | 43.75 ±6.25 | 41.67 ±2.78 | 68.75 ±5.75 | 3 | 2 | 3 | 3 | 3 | 7 | 8 | 1 |
| BirdChicken | 50.00 ±0.10 | 56.25 ±6.25 | 50.00 ±0.10 | 50.00 ±0.10 | 62.50 ±2.86 | 56.25 ±2.86 | 45.83 ±5.40 | 62.50 ±5.15 | 5 | 3 | 5 | 5 | 1 | 3 | 8 | 1 |
| CBF | 24.22 ±0.10 | 49.46 ±0.53 | 98.16 ±0.24 | 75.10 ±2.90 | 99.74 ±0.26 | 99.20 ±0.80 | 98.84 ±0.74 | 99.92 ±0.58 | 8 | 3 | 6 | 7 | 2 | 4 | 5 | 1 |
| Car | 25.00 ±0.10 | 41.66 ±2.47 | 25.00 ±0.10 | 25.00 ±0.10 | 37.50 ±2.86 | 52.08 ±2.09 | 25.00 ±0.10 | 58.34 ±4.16 | 5 | 3 | 5 | 5 | 4 | 2 | 5 | 1 |
| Chinatown | 71.83 ±0.10 | 94.49 ±1.34 | 88.58 ±0.41 | 79.69 ±0.52 | 91.56 ±0.10 | 90.56 ±2.42 | 96.35 ±3.71 | 95.88 ±0.52 | 8 | 3 | 6 | 7 | 4 | 5 | 1 | 2 |
| ChlorineConcentration | 55.36 ±0.10 | 56.06 ±1.96 | 54.96 ±0.09 | 53.70 ±0.10 | 55.67 ±0.23 | 52.18 ±1.14 | 69.43 ±3.43 | 66.97 ±0.08 | 5 | 3 | 6 | 7 | 4 | 8 | 1 | 2 |
| CinCECGTorso | 18.49 ±0.10 | 71.94 ±2.83 | 38.72 ±1.96 | 29.84 ±2.41 | 62.32 ±2.75 | 86.58 ±0.37 | 79.68 ±4.01 | 91.96 ±2.09 | 8 | 4 | 6 | 7 | 5 | 2 | 3 | 1 |
| Coffee | 50.00 ±0.10 | 50.00 ±0.10 | 50.00 ±0.10 | 50.00 ±0.10 | 75.00 ±2.86 | 70.00 ±2.40 | 50.00 ±0.10 | 82.00 ±0.10 | 4 | 4 | 4 | 4 | 2 | 3 | 4 | 1 |
| Computers | 50.00 ±0.10 | 56.64 ±5.08 | 56.25 ±6.25 | 39.06 ±0.10 | 49.22 ±2.78 | 59.38 ±3.12 | 57.82 ±2.75 | 67.97 ±3.12 | 6 | 4 | 5 | 8 | 7 | 2 | 3 | 1 |
| CricketX | 4.69 ±0.39 | 28.40 ±1.78 | 15.62 ±1.34 | 10.94 ±1.56 | 30.98 ±1.97 | 32.82 ±2.42 | 36.07 ±3.37 | 34.49 ±4.87 | 8 | 5 | 6 | 7 | 4 | 3 | 1 | 2 |
| CricketY | 5.08 ±0.10 | 18.12 ±3.30 | 10.60 ±2.92 | 11.25 ±3.13 | 27.32 ±0.18 | 35.22 ±2.63 | 35.09 ±4.17 | 36.34 ±4.55 | 8 | 5 | 7 | 6 | 4 | 2 | 3 | 1 |
| CricketZ | 6.48 ±1.40 | 25.31 ±0.40 | 11.45 ±3.24 | 10.94 ±1.56 | 32.10 ±2.01 | 32.10 ±1.38 | 35.51 ±3.27 | 38.58 ±0.13 | 8 | 5 | 6 | 7 | 3 | 3 | 2 | 1 |
| Crop | 15.47 ±3.67 | 55.90 ±1.07 | 52.96 ±2.46 | 34.93 ±1.76 | 54.20 ±0.13 | 61.14 ±2.42 | 64.75 ±0.38 | 63.31 ±0.35 | 8 | 4 | 6 | 7 | 5 | 3 | 1 | 2 |
| DiatomSizeReduction | 30.65 ±0.10 | 64.48 ±2.79 | 30.60 ±0.06 | 30.32 ±0.62 | 63.23 ±2.76 | 94.93 ±3.51 | 96.81 ±3.47 | 99.16 ±0.84 | 6 | 4 | 7 | 8 | 5 | 3 | 2 | 1 |
| DistalPhalanxOutlineAgeGroup | 59.81 ±0.10 | 82.49 ±0.39 | 79.87 ±1.52 | 70.03 ±3.62 | 77.91 ±1.92 | 76.67 ±3.09 | 75.90 ±2.10 | 82.92 ±4.58 | 8 | 2 | 3 | 7 | 4 | 5 | 6 | 1 |
| DistalPhalanxOutlineCorrect | 73.83 ±0.10 | 72.47 ±0.30 | 67.74 ±3.96 | 69.53 ±4.43 | 68.60 ±0.59 | 69.97 ±3.54 | 75.92 ±0.52 | 73.42 ±3.50 | 3 | 4 | 8 | 6 | 7 | 5 | 1 | 2 |
| DistalPhalanxTW | 48.11 ±0.10 | 77.97 ±1.72 | 64.22 ±7.34 | 66.40 ±2.86 | 73.04 ±2.73 | 72.66 ±2.19 | 77.35 ±2.21 | 80.08 ±0.39 | 8 | 2 | 7 | 6 | 4 | 5 | 3 | 1 |
| DodgerLoopDay | 13.79 ±0.10 | 18.96 ±2.86 | 20.69 ±2.41 | 13.79 ±0.10 | 24.14 ±3.00 | 27.58 ±6.80 | 17.24 ±2.75 | 28.58 ±3.00 | 7 | 5 | 4 | 7 | 3 | 2 | 6 | 1 |
| DodgerLoopGame | 51.61 ±0.10 | 51.61 ±0.10 | 51.61 ±0.10 | 51.61 ±0.10 | 69.35 ±2.44 | 67.74 ±0.10 | 58.84 ±2.70 | 74.19 ±2.86 | 5 | 5 | 5 | 5 | 2 | 3 | 4 | 1 |
| DodgerLoopWeekend | 70.97 ±0.10 | 70.97 ±0.10 | 88.71 ±2.86 | 70.97 ±0.10 | 77.42 ±3.23 | 88.71 ±1.61 | 82.79 ±2.92 | 95.16 ±1.61 | 6 | 6 | 2 | 6 | 5 | 2 | 4 | 1 |
| ECG200 | 67.95 ±1.28 | 76.45 ±3.24 | 68.08 ±2.34 | 80.80 ±2.68 | 82.81 ±2.75 | 81.92 ±2.75 | 83.60 ±3.91 | 86.30 ±3.91 | 8 | 6 | 7 | 5 | 3 | 4 | 2 | 1 |
| ECG5000 | 56.93 ±0.10 | 93.02 ±0.14 | 91.90 ±0.53 | 91.42 ±0.22 | 91.48 ±0.55 | 91.40 ±0.59 | 91.28 ±0.36 | 83.81 ±0.36 | 8 | 1 | 2 | 4 | 3 | 5 | 6 | 7 |
| Average | 35.31 | 53.09 | 45.19 | 42.89 | 56.53 | 58.67 | 58.07 | 65.85 | 6.67 | 3.93 | 5.70 | 5.93 | 3.57 | 3.70 | 3.37 | 1.50 |

Table 9: The accuracy and rank of each method across 30 datasets with label ratio $\alpha = 0.2$.

| Dataset | SupL | PI | MTL | meanTeacher | SemiTime | TapNet | CA-TCC | f-VAT | SupL_rank | PI_rank | MTL_rank | meanTeacher_rank | SemiTime_rank | TapNet_rank | CA-TCC_rank | f-VAT_rank |
|---|---|---|---|---|---|---|---|---|---|---|---|---|---|---|---|---|
| ACSF1 | 10.00 ±0.10 | 14.06 ±1.56 | 10.93 ±2.90 | 29.68 ±2.90 | 21.10 ±2.70 | 25.00 ±2.82 | 24.22 ±2.82 | 60.42 ±2.63 | 8 | 6 | 7 | 2 | 5 | 3 | 4 | 1 |
| Adiac | 5.98 ±2.94 | 7.45 ±2.97 | 2.54 ±0.04 | 14.08 ±2.90 | 20.08 ±1.43 | 34.39 ±0.14 | 30.44 ±2.90 | 38.85 ±2.02 | 7 | 6 | 8 | 5 | 4 | 2 | 3 | 1 |
| AllGestureWiimoteX | 12.68 ±1.21 | 52.23 ±2.84 | 23.44 ±0.22 | 50.00 ±2.84 | 45.98 ±0.45 | 40.62 ±0.90 | 46.43 ±2.83 | 55.36 ±0.45 | 8 | 2 | 7 | 3 | 5 | 6 | 4 | 1 |
| AllGestureWiimoteY | 12.68 ±1.21 | 43.08 ±2.05 | 24.38 ±2.05 | 14.74 ±1.78 | 34.60 ±2.23 | 44.64 ±1.75 | 58.19 ±1.26 | 61.50 ±2.75 | 8 | 4 | 6 | 7 | 5 | 3 | 2 | 1 |
| AllGestureWiimoteZ | 13.89 ±0.10 | 38.84 ±2.54 | 20.31 ±2.75 | 20.31 ±2.75 | 42.18 ±0.02 | 35.27 ±2.69 | 40.85 ±2.96 | 48.96 ±0.67 | 8 | 4 | 6 | 6 | 2 | 5 | 3 | 1 |
| ArrowHead | 38.10 ±0.10 | 65.16 ±2.71 | 55.78 ±0.76 | 25.00 ±0.10 | 67.50 ±1.88 | 75.16 ±2.70 | 69.06 ±2.37 | 83.44 ±2.63 | 7 | 5 | 6 | 8 | 4 | 2 | 3 | 1 |
| BME | 45.83 ±2.51 | 82.81 ±2.55 | 49.22 ±0.76 | 21.10 ±2.93 | 65.62 ±2.86 | 87.89 ±2.79 | 88.54 ±2.77 | 89.06 ±2.79 | 7 | 4 | 6 | 8 | 5 | 3 | 2 | 1 |
| Beef | 45.83 ±2.51 | 25.00 ±2.61 | 20.00 ±0.10 | 20.00 ±0.10 | 35.00 ±2.86 | 25.00 ±2.68 | 23.33 ±2.52 | 40.00 ±2.95 | 6 | 3 | 6 | 6 | 2 | 3 | 5 | 1 |
| BeetleFly | 50.00 ±0.10 | 50.00 ±0.10 | 50.00 ±0.10 | 50.00 ±0.10 | 50.00 ±0.10 | 37.50 ±2.50 | 62.50 ±2.79 | 63.50 ±2.70 | 3 | 3 | 3 | 3 | 3 | 8 | 2 | 1 |
| BirdChicken | 50.00 ±0.10 | 50.00 ±0.10 | 50.00 ±0.10 | 54.00 ±0.10 | 50.00 ±0.10 | 52.00 ±2.53 | 50.00 ±0.10 | 53.25 ±2.70 | 4 | 4 | 4 | 1 | 4 | 3 | 4 | 2 |
| CBF | 57.80 ±2.64 | 100.00 ±0.10 | 97.55 ±0.77 | 99.48 ±0.52 | 98.96 ±0.52 | 100.00 ±0.10 | 97.54 ±0.43 | 99.44 ±1.95 | 8 | 1 | 6 | 3 | 5 | 1 | 7 | 4 |
| Car | 25.00 ±0.10 | 25.00 ±0.10 | 25.00 ±0.10 | 25.00 ±0.10 | 37.50 ±2.70 | 52.08 ±2.09 | 29.17 ±2.76 | 58.34 ±2.61 | 6 | 6 | 6 | 6 | 3 | 2 | 4 | 1 |
| Chinatown | 71.83 ±0.10 | 87.43 ±2.75 | 78.52 ±2.60 | 97.92 ±2.90 | 94.70 ±0.10 | 96.35 ±0.52 | 97.40 ±0.45 | 94.96 ±0.52 | 8 | 6 | 7 | 1 | 5 | 3 | 2 | 4 |
| ChlorineConcentration | 55.36 ±0.10 | 65.07 ±0.90 | 54.12 ±0.74 | 77.22 ±1.88 | 57.29 ±0.10 | 66.90 ±0.76 | 53.70 ±2.69 | 84.80 ±0.10 | 6 | 4 | 7 | 2 | 5 | 3 | 8 | 1 |
| CinCECGTorso | 18.49 ±0.10 | 63.34 ±2.51 | 34.00 ±0.42 | 73.22 ±2.67 | 74.31 ±1.10 | 94.00 ±1.04 | 63.69 ±2.10 | 86.18 ±3.00 | 8 | 6 | 7 | 4 | 3 | 1 | 5 | 2 |
| Coffee | 50.00 ±0.10 | 50.00 ±0.10 | 50.00 ±0.10 | 50.00 ±0.10 | 50.00 ±0.10 | 80.00 ±2.64 | 50.00 ±0.10 | 58.00 ±0.10 | 3 | 3 | 3 | 3 | 3 | 1 | 3 | 2 |
| Computers | 50.00 ±0.10 | 62.89 ±2.60 | 63.28 ±2.63 | 46.09 ±1.17 | 63.28 ±1.56 | 55.47 ±2.72 | 59.38 ±2.65 | 67.58 ±2.75 | 7 | 4 | 2 | 8 | 2 | 6 | 5 | 1 |
| CricketX | 6.48 ±1.40 | 33.44 ±2.82 | 16.77 ±0.64 | 30.18 ±2.90 | 41.83 ±2.28 | 42.54 ±0.41 | 48.93 ±2.83 | 43.16 ±0.45 | 8 | 5 | 7 | 6 | 4 | 3 | 1 | 2 |
| CricketY | 6.48 ±1.40 | 18.30 ±2.77 | 17.21 ±2.62 | 15.32 ±1.56 | 39.91 ±0.09 | 43.39 ±2.73 | 41.25 ±2.51 | 46.49 ±2.55 | 8 | 5 | 6 | 7 | 4 | 2 | 3 | 1 |
| CricketZ | 6.48 ±1.40 | 30.09 ±0.09 | 16.05 ±1.90 | 36.03 ±0.93 | 46.07 ±1.10 | 39.02 ±1.70 | 44.76 ±2.94 | 41.20 ±2.52 | 8 | 6 | 7 | 5 | 1 | 4 | 2 | 3 |
| Crop | 23.44 ±2.40 | 59.36 ±2.77 | 48.42 ±0.94 | 59.82 ±2.61 | 58.88 ±1.60 | 65.79 ±1.77 | 71.77 ±0.67 | 68.95 ±0.43 | 8 | 5 | 7 | 4 | 6 | 3 | 1 | 2 |
| DiatomSizeReduction | 35.48 ±2.01 | 78.03 ±2.01 | 30.60 ±0.23 | 68.75 ±0.73 | 89.84 ±0.78 | 99.16 ±0.10 | 71.91 ±2.75 | 100.00 ±0.84 | 7 | 4 | 8 | 6 | 3 | 2 | 5 | 1 |
| DistalPhalanxOutlineAgeGroup | 59.81 ±0.10 | 79.37 ±0.43 | 70.13 ±2.55 | 79.09 ±0.04 | 74.78 ±2.90 | 72.40 ±0.10 | 78.24 ±2.29 | 81.74 ±2.55 | 8 | 2 | 7 | 3 | 5 | 6 | 4 | 1 |
| DistalPhalanxOutlineCorrect | 73.83 ±0.10 | 68.08 ±0.37 | 55.96 ±2.71 | 65.10 ±0.06 | 79.91 ±0.15 | 73.40 ±2.77 | 79.57 ±1.56 | 80.76 ±0.10 | 4 | 6 | 8 | 7 | 2 | 5 | 3 | 1 |
| DistalPhalanxTW | 48.11 ±0.10 | 75.39 ±0.08 | 67.26 ±0.08 | 75.78 ±0.10 | 74.61 ±1.17 | 00.00 ±0.10 | 75.78 ±0.10 | 79.30 ±0.39 | 7 | 4 | 6 | 2 | 5 | 8 | 2 | 1 |
| DodgerLoopDay | 13.79 ±0.10 | 18.96 ±2.79 | 20.69 ±2.90 | 17.24 ±2.92 | 29.31 ±2.78 | 34.48 ±2.92 | 21.84 ±2.96 | 26.20 ±1.73 | 8 | 6 | 5 | 7 | 2 | 1 | 4 | 3 |
| DodgerLoopGame | 51.61 ±0.10 | 61.29 ±2.90 | 53.22 ±1.02 | 58.06 ±2.79 | 59.68 ±2.94 | 75.81 ±2.78 | 59.14 ±2.77 | 77.42 ±2.81 | 8 | 3 | 7 | 6 | 4 | 2 | 5 | 1 |
| DodgerLoopWeekend | 70.97 ±0.10 | 83.87 ±2.50 | 91.94 ±1.62 | 82.26 ±2.47 | 88.71 ±2.83 | 95.16 ±1.61 | 84.95 ±2.91 | 96.77 ±0.10 | 8 | 6 | 3 | 7 | 4 | 2 | 5 | 1 |
| ECG200 | 66.67 ±0.10 | 73.22 ±0.90 | 78.02 ±2.90 | 81.59 ±2.71 | 84.38 ±2.70 | 76.34 ±2.61 | 80.36 ±2.71 | 86.31 ±2.90 | 8 | 7 | 5 | 3 | 2 | 6 | 4 | 1 |
| ECG5000 | 56.93 ±0.10 | 93.16 ±0.10 | 92.28 ±0.49 | 91.21 ±0.29 | 91.96 ±0.33 | 92.43 ±0.54 | 92.21 ±0.08 | 94.21 ±0.08 | 8 | 2 | 4 | 7 | 6 | 3 | 5 | 1 |
| Average | 36.92 | 55.16 | 45.72 | 50.94 | 58.93 | 60.41 | 59.84 | 68.87 | 7.00 | 4.40 | 5.87 | 4.87 | 3.77 | 3.40 | 3.67 | 1.50 |

Table 10: The accuracy and rank of each method across 30 datasets with label ratio $\alpha = 0.4$.

| dataset | SupL | PI | MTL | meanTeacher | SemiTime | TapNet | CA-TCC | f-VAT | SupL_rank | PI_rank | MTL_rank | meanTeacher_rank | SemiTime_rank | TapNet_rank | CA-TCC_rank | f-VAT_rank |
|---|---|---|---|---|---|---|---|---|---|---|---|---|---|---|---|---|
| ACSF1 | 11.25 ±1.25 | 33.60 ±2.34 | 8.59 ±2.34 | 29.69 ±2.70 | 35.16 ±2.66 | 43.75 ±1.56 | 44.79 ±2.88 | 46.88 ±1.56 | 7 | 5 | 8 | 6 | 4 | 3 | 2 | 1 |
| Adiac | 4.78 ±2.97 | 3.93 ±1.25 | 2.93 ±0.25 | 11.06 ±2.70 | 36.26 ±0.14 | 55.67 ±1.25 | 48.67 ±2.92 | 60.08 ±1.21 | 6 | 7 | 8 | 5 | 4 | 2 | 3 | 1 |
| AllGestureWiimoteX | 13.89 ±0.10 | 51.56 ±1.11 | 25.22 ±0.67 | 55.58 ±0.67 | 54.28 ±0.67 | 57.36 ±2.90 | 55.65 ±2.76 | 59.15 ±2.58 | 8 | 6 | 7 | 4 | 5 | 2 | 3 | 1 |
| AllGestureWiimoteY | 13.89 ±0.10 | 52.90 ±2.45 | 24.22 ±2.79 | 56.03 ±0.67 | 69.19 ±2.24 | 62.50 ±2.82 | 54.28 ±0.56 | 61.99 ±2.90 | 8 | 6 | 7 | 4 | 1 | 2 | 5 | 3 |
| AllGestureWiimoteZ | 13.89 ±0.10 | 49.11 ±2.68 | 18.52 ±0.22 | 54.30 ±1.34 | 50.22 ±2.69 | 47.99 ±0.67 | 36.16 ±2.73 | 54.61 ±2.90 | 8 | 4 | 7 | 1 | 3 | 5 | 6 | 2 |
| ArrowHead | 38.10 ±0.10 | 67.66 ±2.67 | 41.64 ±2.77 | 77.81 ±2.05 | 73.28 ±2.96 | 82.82 ±2.96 | 66.46 ±2.96 | 81.56 ±2.75 | 8 | 4 | 7 | 2 | 3 | 5 | 6 | 1 |
| BME | 45.84 ±2.97 | 88.28 ±2.79 | 54.30 ±2.71 | 76.56 ±2.41 | 93.75 ±2.90 | 95.32 ±1.56 | 89.06 ±2.70 | 100.00 ±0.10 | 8 | 5 | 7 | 6 | 3 | 2 | 4 | 1 |
| Beef | 20.00 ±0.10 | 25.00 ±2.79 | 20.00 ±0.10 | 25.00 ±2.79 | 25.00 ±2.79 | 30.00 ±2.80 | 25.00 ±2.77 | 50.00 ±2.96 | 5 | 3 | 6 | 3 | 3 | 2 | 6 | 1 |
| BeetleFly | 50.00 ±0.10 | 56.25 ±2.75 | 50.00 ±0.10 | 50.00 ±0.10 | 68.75 ±2.88 | 62.50 ±2.92 | 50.00 ±0.10 | 87.50 ±2.77 | 5 | 4 | 5 | 5 | 2 | 3 | 5 | 1 |
| BirdChicken | 50.00 ±0.10 | 62.50 ±2.55 | 50.00 ±0.10 | 56.25 ±2.55 | 81.25 ±2.77 | 75.00 ±2.77 | 50.00 ±0.10 | 62.50 ±2.69 | 5 | 2 | 5 | 4 | 1 | 5 | 5 | 2 |
| CBF | 37.89 ±2.62 | 100.00 ±0.10 | 97.92 ±1.04 | 99.48 ±0.62 | 100.00 ±0.10 | 99.71 ±0.29 | 100.00 ±0.10 | 98.96 ±0.44 | 8 | 1 | 6 | 5 | 1 | 4 | 1 | 6 |
| Car | 22.92 ±2.09 | 25.00 ±0.10 | 25.00 ±0.10 | 45.84 ±2.90 | 45.83 ±0.10 | 60.42 ±0.10 | 41.67 ±2.75 | 63.67 ±2.63 | 8 | 6 | 6 | 3 | 4 | 2 | 5 | 1 |
| Chinatown | 71.83 ±0.10 | 86.68 ±2.59 | 83.08 ±2.77 | 97.92 ±1.04 | 91.98 ±0.10 | 93.98 ±2.77 | 98.96 ±0.45 | 99.40 ±0.52 | 8 | 6 | 7 | 3 | 5 | 4 | 2 | 1 |
| ChlorineConcentration | 55.36 ±0.10 | 80.24 ±2.69 | 54.18 ±0.64 | 89.92 ±2.65 | 59.94 ±1.61 | 78.43 ±1.77 | 53.70 ±0.39 | 92.97 ±0.10 | 6 | 3 | 7 | 2 | 5 | 4 | 8 | 1 |
| CinCECGTorso | 32.94 ±2.93 | 79.74 ±2.47 | 41.37 ±2.03 | 82.10 ±2.76 | 86.61 ±1.39 | 91.57 ±2.88 | 94.70 ±0.35 | 91.57 ±2.88 | 8 | 6 | 7 | 5 | 4 | 2 | 1 | 2 |
| Coffee | 50.00 ±0.10 | 100.00 ±0.10 | 50.00 ±0.10 | 90.00 ±0.10 | 99.00 ±0.10 | 99.00 ±0.10 | 80.00 ±2.70 | 100.00 ±2.88 | 7 | 1 | 7 | 5 | 3 | 3 | 6 | 1 |
| Computers | 50.00 ±0.10 | 68.36 ±2.73 | 53.71 ±2.90 | 66.02 ±2.72 | 60.94 ±0.10 | 65.62 ±2.69 | 66.40 ±2.97 | 71.09 ±0.10 | 8 | 2 | 7 | 4 | 6 | 5 | 3 | 1 |
| CricketX | 5.08 ±0.10 | 42.77 ±2.82 | 18.84 ±2.05 | 52.68 ±1.61 | 58.76 ±0.62 | 63.57 ±2.94 | 56.49 ±2.90 | 66.78 ±0.71 | 8 | 6 | 7 | 5 | 3 | 2 | 4 | 1 |
| CricketY | 22.76 ±2.78 | 31.25 ±2.79 | 15.14 ±3.00 | 30.62 ±2.97 | 54.51 ±2.84 | 60.31 ±2.99 | 54.23 ±2.94 | 64.10 ±2.95 | 7 | 5 | 8 | 6 | 3 | 2 | 4 | 1 |
| CricketZ | 6.48 ±1.40 | 44.20 ±2.68 | 18.84 ±2.93 | 40.62 ±2.90 | 55.80 ±2.96 | 60.14 ±2.96 | 56.13 ±2.93 | 63.75 ±2.94 | 8 | 5 | 7 | 6 | 4 | 2 | 3 | 1 |
| Crop | 24.52 ±0.96 | 71.32 ±2.90 | 49.84 ±0.66 | 54.92 ±2.20 | 62.40 ±0.78 | 60.79 ±0.20 | — | 74.32 ±1.47 | 7 | 2 | 6 | 5 | 3 | 4 | — | 1 |
| DiatomSizeReduction | 31.46 ±0.90 | 92.16 ±2.75 | 45.14 ±2.63 | 82.19 ±2.45 | 95.32 ±2.61 | 95.21 ±2.62 | 59.48 ±2.90 | 96.21 ±2.08 | 8 | 4 | 7 | 6 | 2 | 3 | 5 | 1 |
| DistalPhalanxOutlineAgeGroup | 59.81 ±0.10 | 76.08 ±2.79 | 72.05 ±2.79 | 78.31 ±2.52 | 79.76 ±2.34 | 78.73 ±2.60 | 78.22 ±2.92 | 76.76 ±0.94 | 8 | 6 | 7 | 2 | 1 | 3 | 4 | 5 |
| DistalPhalanxOutlineCorrect | 73.83 ±0.10 | 73.32 ±2.76 | 59.92 ±2.52 | 79.91 ±0.37 | 80.43 ±0.52 | 65.62 ±2.76 | 78.08 ±2.70 | 81.32 ±2.14 | 5 | 6 | 8 | 3 | 2 | 7 | 4 | 1 |
| DistalPhalanxTW | 48.11 ±0.10 | 71.59 ±2.79 | 71.48 ±0.96 | 75.78 ±0.10 | 76.95 ±0.09 | 73.23 ±1.82 | 75.78 ±0.10 | 78.59 ±2.79 | 8 | 6 | 7 | 3 | 2 | 5 | 3 | 1 |
| DodgerLoopDay | 13.79 ±0.10 | 42.58 ±2.79 | 25.86 ±1.72 | 22.42 ±2.77 | 29.31 ±1.72 | 44.82 ±2.88 | 24.14 ±2.80 | 43.82 ±2.96 | 8 | 3 | 5 | 7 | 4 | 1 | 6 | 2 |
| DodgerLoopGame | 51.61 ±0.10 | 71.58 ±2.51 | 54.28 ±0.79 | 75.81 ±1.62 | 70.97 ±2.92 | 75.81 ±2.90 | 59.14 ±2.96 | 78.15 ±2.63 | 7 | 4 | 8 | 2 | 5 | 2 | 6 | 1 |
| DodgerLoopWeekend | 70.97 ±0.10 | 92.83 ±2.71 | 90.32 ±0.10 | 80.64 ±2.84 | 98.38 ±1.62 | 96.77 ±0.10 | 59.97 ±2.90 | 99.38 ±0.96 | 7 | 4 | 5 | 6 | 2 | 3 | 7 | 1 |
| ECG200 | 66.67 ±0.10 | 76.14 ±2.96 | 72.72 ±3.00 | 85.04 ±2.92 | 83.48 ±2.47 | 80.24 ±2.78 | 80.24 ±2.78 | 83.14 ±0.16 | 8 | 6 | 7 | 1 | 2 | 4 | 4 | 3 |
| ECG5000 | 56.93 ±0.10 | 91.38 ±2.77 | 90.83 ±0.10 | 92.30 ±0.32 | 93.04 ±0.10 | 94.68 ±0.24 | 94.68 ±0.24 | 95.68 ±0.36 | 8 | 6 | 7 | 5 | 4 | 2 | 2 | 1 |
| - | 37.15 | 63.60 | 46.11 | 63.85 | 69.02 | 70.28 | 63.27 | 76.24 | 7.33 | 4.47 | 6.80 | 4.13 | 3.13 | 3.17 | 4.27 | 1.53 |

# F    Runtime comparison

Compared with VAT ($\mathcal{O}(N)$), the computational complexity of f-VAT is $\mathcal{O}(N \log N)$ with a small constant factor, due to the FFT-based $\|\cdot\|_{H^{-s}}$ normalization. The empirical results in Table 11 show

that the extra computational cost remains in the same order as VAT. The proposed f-VAT achieves competitive performance without incurring significant computational costs, making it particularly suitable for semi-supervised time series classification with limited computational resources.

Table 11: Runtime comparison of f-VAT and VAT.

| Method | CricketX | UWave | InsectWing | NATOPS | SelfReg |
|---|---|---|---|---|---|
| VAT | 15.67 | 51.66 | 35.58 | 28.04 | 30.68 |
| f-VAT | 20.45 | 62.05 | 45.92 | 38.98 | 44.95 |
| $\Delta$ (%) | 30.50 | 20.11 | 29.06 | 39.02 | 46.51 |

