# OpenReview forum: "Functional Virtual Adversarial Training for Semi-Supervised Time Series Classification"
_NeurIPS.cc/2025/Conference — NeurIPS 2025 poster_

### Official Review · Reviewer_RR58 · 2025-07-01

**Clarity:** 2
**Significance:** 2
**Originality:** 3
**Rating:** 4
**Confidence:** 4

**Summary:**

The paper points out that the Virtual Adversarial Training falls short on time series as the added perturbations bounded by  Euclidean norm disrupt time series patterns. The paper proposes the Sobolev norm to replace the Euclidean norm to generate structured functional adversarial perturbations.

**Questions:**

Please refer to Weaknesses.

**Ethical Concerns:**

["NO or VERY MINOR ethics concerns only"]

**Final Justification:**

I appreciate the thorough experimental results provided by the authors, and I would increase my rating to 4.

**Limitations:**

The evaluation is limited and some statements are confusing.

**Paper Formatting Concerns:**

None.

**Quality:**

3

**Strengths And Weaknesses:**

### **Strengths**

This paper points out that the Virtual Adversarial Training falls short on time series: the perturbation vector constrained by Euclidean norm disrupts temporal patterns. This insight is very enlightening for designing effective adversarial methods tailored for time series.





### **Weakness 1**

Why is time series considered functional data, while images are not? Time series is a function of some interval $T$, and images can also be considered functions in a spatial context.

Despite "Functional" being a key term in the title, I still do not understand its specific meaning in this paper after reading the introduction. Additionally, it's unclear what unique designs are applied to time series (at least not sufficiently articulated in the introduction). From this perspective, I believe the writing of the paper needs improvement.

### **Weakness 2**

The clarity and motivations in Section 3 are confusing.

- **W 2.1**: How do Theorem 3.3 and Theorem 3.4 assist in the method design? The condition of Theorem 3.3  is that the model is linear, and the condition of  Theorem 3.4 is $\epsilon\rightarrow0^+$. **These two conditions are not satisfied in practical training**.
- **W 2.2**: Based on Line 163~170, the conclusions of Theorem 3.3 and Theorem 3.4 state that "if we want to impose a stronger norm on the sensitivity of the method, we should relax the norm of the perturbation." How is this related to the introduction of the Sobolev norm?
- **W 2.3**: Why does the Sobolev space penalize high-frequency oscillations? Is the benefit of the Sobolev space solely that it penalizes high-frequency noise? This can be achieved through common high-frequency filtering, which has already been widely applied in existing time series analysis methods.
- **W 2.4**: Compared with VAT:
  - The VAT only needs a hyperparameter $\epsilon$, but f-VAT needs **four hyperparameters** $\epsilon$,  $s$, $L$, and $\eta$. VAT only needs to compute once to obtain the optimal perturbation $r$, whereas f-VAT requires $L$ iterations and the hyperparameter $\eta$.
  - Do the perturbation vector $r$ and the model parameters $\theta$ share the same learning rate $\eta$ as shown in Algorithm 1?
  - In addition to the Sobolev norm order, the sensitive analysis of other hyperparameters should be included, along with a comparison of the training time between VAT and f-VAT.
- **W 2.5**: In Line 6 of Algorithm 1, why adopt gradient ascent and normalize  $r_i$? In the original paper of VAT, the optimal $r$ is obtained by calculating the first dominant eigenvector of the Hessian matrix $\nabla\nabla_r D(r, x, \hat{\theta})|_{r=0}$. Is the approach not suitable for f-VAT? For instance, calculate the optimal $r$ whose Sobolev norm is $\epsilon$ in the direction of the first dominant eigenvector?
- **W 2.6**: Could the authors provide the formula of $||\cdot|| _ {H ^ {-s}}$ as well, rather than just $||\cdot|| _ {H ^ {s}}$. It is confusing that Eq 11 only introduces $||\cdot|| _ {H ^ {s}}$, while Algorithm 1 actually constrains $r$ using $||\cdot|| _ {H ^ {-s}}$. Additionally, in Eq 11 and its context, $L$ and $L _ N$ seem to be confused.

### **Weakness 3**

This paper lacks a discussion of related works, particularly in the area of semi-supervised time series. Additionally, the experiments do not compare with the latest semi-supervised methods [1, 2]. More importantly, the paper only includes three datasets from UCR and three from UEA. However, TS-TFC [1] includes **106 datasets** from UCR. As far as I know, these time series datasets are relatively small-scale and highly heterogeneous, making the evaluation in the paper seem inadequate.

[1] Temporal-Frequency Co-training for Time Series Semi-supervised Learning. *AAAI*, 2023.

[2] Self-supervised Contrastive Representation Learning for Semi-supervised Time-Series Classification. TPAMI, 2022.



### **Weakness 4**

In Section 4.3, the authors claim that functional adversarial perturbations add subtle disturbances only in non-critical areas, while keeping them small in important regions. However, I did not observe this phenomenon. **Where are the critical regions versus the non-critical regions?** It appears that f-VAT applies small perturbations at each time point, while VAT applies larger perturbations at certain time points.

Therefore, a straightforward improvement to VAT could involve not only constraining the norm of the overall perturbation vector but also limiting the magnitude of perturbations at each time point, or constraining the variance of perturbations at each time point. Would this be viable？

---

> ### Author Rebuttal · Authors · 2025-07-31
>
> We would like to point out that the contribution of this paper is to propose the **theory-inspired f-VAT algorithm that performs well in practice**, which is supported by our experiments (and additional ones, see answers below) that f-VAT consistently outperforms other competitive baselines in all settings.
> While the theory considers simplified settings, it provides us with key insights into designing the f-VAT algorithm, particularly in constructing perturbation that aligns with the time series structure of the data.
> Therefore, we believe that our paper is novel and significant both empirically and theoretically.
>
> ### W1.1
> We agree that both time series and images can be viewed as functional data, but their intrinsic structures can be different. In this paper, we mainly focus on time series data, but our general f-VAT framework can be applied to broader settings. Exploring f-VAT for functional data with other structures, such as image data, can be one of our future directions.
> ### W1.2
> We use the term "Functional" to emphasize the function structure of the data, particularly for time series data. Focusing on the additional function structure of the data, our novel f-VAT method generates adversarial perturbations that align with the key temporal structure, i.e., the dynamic correlations within variables changing over time steps.  This is the key improvement of f-VAT over VAT.
>
> ### W2.1
> Our theory aims to provide insights into designing adversarial perturbations when the data have additional structure, particularly for time series data. We believe that our existing theory has already provided much insight for the design of f-VAT with the duality theory, and due to the highly complex nature of neural networks, further theoretical analysis would be beyond the scope of this paper. Moreover, strong empirical results on various time series datasets demonstrate the effectiveness of our method.
>
> Please see our response to the next point for more details on how the theory is used in the method design.
>
> ### W2.2
> In our theory, we first establish the general theory on duality between the perturbation norm and the model sensitivity norm (Theorem 3.3 and Theorem 3.4). However, we still need to choose a norm (or the space for the norm) in the concrete implementation. For time series data, it is natural to consider Sobolev spaces $H^s$ (of positive order) to capture the smoothness/regularity for model sensitivity. Then, with the duality theory, we transfer $H^s$ to $H^{-s}$ for the perturbation norm, where we use the properties of Sobolev spaces that the dual space of $H^s$ is $(H^s)^* = H^{-s}$. In fact, it is one of the key points in our paper to use the Sobolev norm $H^{-s}$ of negative order for the perturbation, which is critical for the success of the proposed f-VAT.
>
> We would like to clarify that our point is "The Sobolev norm allows us to properly control the model's sensitivity to high-frequency noises''. This can be seen from duality theory: the Sobolev norm $H^{-s}$ imposed on the perturbation effectively serves as a regularization term that penalizes the model sensitivity in terms of the Sobolev space $H^s$.
>
> ### W2.3
> The infinite-dimensional nature of the Sobolev space allows for a richer specification of perturbation constrains that align with the geometry of functional data like periodicity, rather than penalizing high-frequency filtering. Thus, the functional adversarial perturbation generated by f-VAT in Sobolev space can capture the temporal structure of time series data.
>
> ### W2.4
> - W2.4.1:
> We would like to clarify that f-VAT and VAT require the same set of hyperparameters: $\epsilon$ controls the perturbation radius; $L$ is the number of power-iteration steps (typically set to $L=1$); and $\eta$ is the step size for gradient ascent during the power-iteration. The only additional hyperparameter of f-VAT is the order $s$ of norm $\|\cdot\|_{H^s}$.
>
> - W2.4.2:
> Yes, both the perturbation vector $r$ and the model parameters share the same learning $\eta$, as clarifies in [Takeru et al., 2018].
>
> - W2.4.3:
> Compared to original VAT, f-VAT only introduces an additional hyperparameter $s$. We evaluate the performance on more datasets, which shows that setting $s=2$ achieves the best performance. Table shows that the extra computational cost remains in the same order as VAT. Due to the page limit, please refer to our replies to Reviewer #Jy1d Q1 & W2.
>
> ###  W2.5
> The original VAT also adopts gradient ascent and normalizes $r_i$ to approximate the first dominant eigenvector of the Hessian matrix $\nabla\nabla D(r,x,\hat{\theta})|_{r=0}$, while the proposed f-VAT adopts the same power-iteration procedure with $L=1$. The key difference is that the projection into the $\epsilon$ ball in the $L_2$ space is replaced by the $H^{-s}$ space.
>
> ###  W2.6
> We apologize for the typo in Eq.(11) and the confusion caused by the notation. In fact, as emphasized in the paragraph before Eq.(11), the formula $\|r\|_{H^\alpha}^2 \approx r_N^\top (I_N + L_N)^\alpha r_N$ applies to all $\alpha \in \mathbb{R}$, including negative orders, so we do not distinguish $s$ or $-s$. Here, the operator $L$ should always be replaced by $L_N$ which represents the discrete Laplacian operator. We have corrected the notation in the revised version of the paper, and we hope this resolves the confusion.
>
> ### W3
> We have add more detailed discussion of literature review in the revised version.
> We follow the settings [Haoyi et al., 2021] to compare f-VAT with other competitive baselines and report results on six UCR/UEA datasets. We **randomly sample 30 datasets from UCR/UEA datasets and add a state-of-the-art Class-Aware Temporal and Contextual Contrasting (TS-TCC) [Emadeldeen et al., 2023] within available resources**. Table shows that tVAT consistently outperforms other baselines in all settings. Please see our additional comments for more experiment details.
>
> Average accuracy (%) and average rank across the dataseets under different label ratios:
> | Method   | 10% |      | 20% |      | 40% |      |
> |:--------:|:----:|-----:|:----:|-----:|:----:|-----:|
> |          | AvgAcc | AvgRank | AvgAcc | AvgRank | AvgAcc | AvgRank |
> | SupL     | 35.31 | 6.67 | 36.92 | 7.00 | 37.15 | 7.33 |
> | PI       | 53.09 | 3.93 | 55.16 | 4.40 | 63.60 | 4.47 |
> | MTL      | 45.19 | 5.70 | 45.72 | 5.87 | 46.11 | 6.80 |
> | SemiTime | 56.53 | 3.57 | 58.93 | 3.77 | 69.02 | 3.17 |
> | TapNet   | 58.67 | 3.70 | 60.41 | 3.40 | 70.28 | 3.20 |
> | TS‑TCC   | 58.07 | 3.37 | 59.84 | 3.67 | 63.27 | 4.27 |
> | f-VAT     | 65.85 | 1.50 | 68.87 | 1.50 | 76.24 | 1.53 |
>
> Additionally, we construct empirical on several large-scale China Securities Index (CSI) datasets (i.e., CSI 50 and 500 futures) spanning from 2020 to 2023 for predicting directions (upward or downward) of futures prices [Zihao et al., 2019]. The dataset collects over $42\times 10^{3}$ records spanning from 2020 to 2022, and each time step contains bid/ask prices and corresponding volumes. Table presents the performance of various semi-supervised methods across different label ratios, and shows that f-VAT significantly outperforms other competitive baselines on futures datasets, especially on more volatile CSI 500 futures.
>
> The performance comparison on domestic futures datasets:
> | Futures | Ratio | supL | PI | MTL | SemiTime | TapNet | TS-TCC | f-VAT |
> |:------:|:----:|---------------:|-----------------------:|--------------:|-------------------:|-----------------:|-----------------:|--------------------:|
> |        | 10% | 40.05 $\pm$ 1.38 | 42.87 $\pm$ 1.45 | 53.94 $\pm$ 0.13 | 55.19 $\pm$ 0.77 | 54.62 $\pm$ 0.54 | 55.72 $\pm$ 1.16 | **58.64 $\pm$ 0.53** |
> | 50     | 20% | 45.08 $\pm$ 1.45 | 47.26 $\pm$ 0.73 | 54.97 $\pm$ 0.61 | 56.69 $\pm$ 0.50 | 56.93 $\pm$ 1.28 | 58.21 $\pm$ 0.44 | **62.09 $\pm$ 0.26** |
> |        | 40% | 50.69 $\pm$ 3.53 | 52.42 $\pm$ 1.21 | 56.97 $\pm$ 0.57 | 57.34 $\pm$ 0.20 | 59.75 $\pm$ 1.09 | 59.80 $\pm$ 1.32 | **64.78 $\pm$ 0.45** |
> |        | 10% | 34.23 $\pm$ 0.24 | 44.00 $\pm$ 0.06 | 39.53 $\pm$ 1.35 | 38.86 $\pm$ 2.04 | 40.53 $\pm$ 0.83 | 39.79 $\pm$ 1.54 | **43.77 $\pm$ 0.73** |
> | 500    | 20% | 35.38 $\pm$ 1.06 | 46.57 $\pm$ 0.48 | 45.26 $\pm$ 0.29 | 46.04 $\pm$ 0.12 | 44.58 $\pm$ 1.70 | 45.05 $\pm$ 1.74 | **52.14 $\pm$ 0.25** |
> |        | 40% | 43.85 $\pm$ 1.49 | 49.25 $\pm$ 0.55 | 47.66 $\pm$ 1.50 | 50.65 $\pm$ 1.05 | 51.50 $\pm$ 1.36 | 54.29 $\pm$ 0.87 | **58.66 $\pm$ 0.46** |
>
> ### W4.1
> The critical areas refer to the input feature essential for prediction. Figure 4 visualizes the essential "shapelets'' in yellow and the irrelevant regions in blue. Our proposed f-VAT can adaptively concentrate its perturbations on these non-critical regions, aligning with the original samples across all key time steps, while the original VAT may introduce large spikes in some critical regions and disrupt the underlying trend structure.
>
> ### W4.2
> The straightforward improvement to VAT may be unsuitable for time series. If we limit the magnitude of perturbations at each time step, the order of time steps can be arbitrarily permutable to destroy the temporal structure. On the other hand, our proposed f-VAT can capture the key trend structure, i.e., the implicit correlations within variables changing over time, by generating theory-inspired functional adversarial perturbations.
>
> - [Takeru et al., 2018] Takeru Miyato, etc., Virtual adversarial training: a regularization method for supervised and semi-supervised learning. TPAMI, 2018.
>
> - [Haoyi et al., 2021] Haoyi Fan, etc, Semi-supervised time series classification by temporal relation prediction. ICASSP, 2021.
>
> - [Emadeldeen et al., 2023] Emadeldeen Eldele, et. al., Self-supervised contrastive representation learning for semi-supervised time-series classification. TPAMI, 2023.
>
> - [Zihao et al., 2019] Zihao Zhang etc, Deeplob: Deep convolutional neural networks for limit order books, 2019.

---

> > ### Author Response · Authors · 2025-08-01
> > **Accuracy and rank of each method across 30 datasets with label ratio 0.2 (First part)**
> >
> > Due to character limitations, we split each label-ratio table into two parts and posted them in separate comments for your reference.
> >
> > | Dataset | SupCE | PI | MTL | meanTeacher | SemiTime | TapNet | TS-TCC | fVAT | SupCE_rank | PI_rank | MTL_rank | meanTeacher_rank | SemiTime_rank | TapNet_rank | CATCC_rank | f-VAT_rank |
> > | --- | --- | --- | --- | --- | --- | --- | --- | --- | --- | --- | --- | --- | --- | --- | --- | --- |
> > | ACSF1 | 10.00 $\pm$ 0.10 | 14.06 $\pm$ 1.56 | 10.93 $\pm$ 2.90 | 29.68 $\pm$ 2.80 | 21.10 $\pm$ 2.79 | 25.00 $\pm$ 2.82 | 24.22 $\pm$ 2.82 | 60.42 $\pm$ 2.63 | 8 | 6 | 7 | 2 | 5 | 3 | 4 | 1 |
> > | Adiac | 5.98 $\pm$ 2.82 | 7.45 $\pm$ 2.87 | 2.54 $\pm$ 0.04 | 14.08 $\pm$ 2.89 | 20.08 $\pm$ 1.43 | 34.39 $\pm$ 0.14 | 30.44 $\pm$ 2.90 | 38.85 $\pm$ 2.62 | 7 | 6 | 8 | 5 | 4 | 2 | 3 | 1 |
> > | AllGestureWiimoteX | 12.68 $\pm$ 1.21 | 52.23 $\pm$ 2.84 | 23.44 $\pm$ 0.22 | 50.00 $\pm$ 2.84 | 45.98 $\pm$ 0.45 | 40.62 $\pm$ 0.90 | 46.43 $\pm$ 2.83 | 55.36 $\pm$ 0.45 | 8 | 2 | 7 | 3 | 5 | 6 | 4 | 1 |
> > | AllGestureWiimoteY | 12.68 $\pm$ 1.21 | 43.08 $\pm$ 2.95 | 24.38 $\pm$ 2.05 | 14.74 $\pm$ 1.78 | 34.60 $\pm$ 2.23 | 44.64 $\pm$ 1.78 | 58.19 $\pm$ 1.28 | 61.50 $\pm$ 2.78 | 8 | 4 | 6 | 7 | 5 | 3 | 2 | 1 |
> > | AllGestureWiimoteZ | 13.89 $\pm$ 0.10 | 38.84 $\pm$ 2.54 | 20.31 $\pm$ 1.12 | 20.31 $\pm$ 2.75 | 42.18 $\pm$ 0.22 | 35.27 $\pm$ 2.68 | 40.85 $\pm$ 2.96 | 48.96 $\pm$ 0.67 | 8 | 4 | 6 | 6 | 2 | 5 | 3 | 1 |
> > | ArrowHead | 38.10 $\pm$ 0.10 | 65.16 $\pm$ 2.71 | 55.78 $\pm$ 0.78 | 25.00 $\pm$ 0.10 | 67.50 $\pm$ 1.88 | 75.16 $\pm$ 2.76 | 69.06 $\pm$ 2.57 | 83.44 $\pm$ 2.63 | 7 | 5 | 6 | 8 | 4 | 2 | 3 | 1 |
> > | BME | 45.83 $\pm$ 2.51 | 82.81 $\pm$ 2.55 | 49.22 $\pm$ 0.78 | 21.10 $\pm$ 2.93 | 65.62 $\pm$ 2.86 | 87.89 $\pm$ 2.79 | 88.54 $\pm$ 2.77 | 89.06 $\pm$ 2.79 | 7 | 4 | 6 | 8 | 5 | 3 | 2 | 1 |
> > | Beef | 20.00 $\pm$ 0.10 | 25.00 $\pm$ 2.61 | 20.00 $\pm$ 0.10 | 20.00 $\pm$ 0.10 | 35.00 $\pm$ 2.88 | 25.00 $\pm$ 2.68 | 23.33 $\pm$ 2.52 | 40.00 $\pm$ 2.95 | 6 | 3 | 6 | 6 | 2 | 3 | 5 | 1 |
> > | BeetleFly | 50.00 $\pm$ 0.10 | 50.00 $\pm$ 0.10 | 50.00 $\pm$ 0.10 | 50.00 $\pm$ 0.10 | 50.00 $\pm$ 0.10 | 37.50 $\pm$ 2.59 | 62.50 $\pm$ 2.79 | 63.50 $\pm$ 2.70 | 3 | 3 | 3 | 3 | 3 | 8 | 2 | 1 |
> > | BirdChicken | 50.00 $\pm$ 0.10 | 50.00 $\pm$ 0.10 | 50.00 $\pm$ 0.10 | 54.00 $\pm$ 0.10 | 50.00 $\pm$ 0.10 | 52.00 $\pm$ 2.53 | 50.00 $\pm$ 0.10 | 53.25 $\pm$ 0.10 | 4 | 4 | 4 | 1 | 4 | 3 | 4 | 2 |
> > | CBF | 57.80 $\pm$ 2.64 | 100.00 $\pm$ 0.00 | 97.55 $\pm$ 0.77 | 99.48 $\pm$ 0.52 | 98.96 $\pm$ 0.52 | 100.00 $\pm$ 0.00 | 97.54 $\pm$ 0.43 | 99.44 $\pm$ 1.95 | 8 | 1 | 6 | 3 | 5 | 1 | 7 | 4 |
> > | Car | 25.00 $\pm$ 0.10 | 25.00 $\pm$ 0.10 | 29.16 $\pm$ 2.85 | 25.00 $\pm$ 0.10 | 37.50 $\pm$ 2.79 | 52.08 $\pm$ 2.09 | 29.17 $\pm$ 2.70 | 58.34 $\pm$ 2.61 | 6 | 6 | 5 | 6 | 3 | 2 | 4 | 1 |
> > | Chinatown | 71.83 $\pm$ 0.10 | 87.43 $\pm$ 2.75 | 78.52 $\pm$ 2.60 | 97.92 $\pm$ 2.90 | 94.70 $\pm$ 0.10 | 96.35 $\pm$ 0.52 | 97.40 $\pm$ 0.85 | 94.96 $\pm$ 0.52 | 8 | 6 | 7 | 1 | 5 | 3 | 2 | 4 |
> > | ChlorineConcentration | 55.36 $\pm$ 0.10 | 65.07 $\pm$ 0.90 | 54.12 $\pm$ 0.34 | 77.22 $\pm$ 1.88 | 57.29 $\pm$ 0.10 | 66.90 $\pm$ 0.76 | 53.70 $\pm$ 2.69 | 84.80 $\pm$ 0.10 | 6 | 4 | 7 | 2 | 5 | 3 | 8 | 1 |
> > | CinCECGTorso | 18.49 $\pm$ 0.10 | 63.34 $\pm$ 2.51 | 34.00 $\pm$ 0.42 | 73.22 $\pm$ 2.67 | 74.31 $\pm$ 1.10 | 94.00 $\pm$ 1.04 | 63.69 $\pm$ 2.10 | 86.18 $\pm$ 3.00 | 8 | 6 | 7 | 4 | 3 | 1 | 5 | 2 |
> > | Coffee | 50.00 $\pm$ 0.10 | 50.00 $\pm$ 0.10 | 50.00 $\pm$ 0.10 | 50.00 $\pm$ 0.10 | 50.00 $\pm$ 0.10 | 80.00 $\pm$ 2.64 | 50.00 $\pm$ 0.10 | 58.00 $\pm$ 0.10 | 3 | 3 | 3 | 3 | 3 | 1 | 3 | 2 |

---

> > ### Comment · Reviewer_RR58 · 2025-08-04
> >
> > Thanks for the clarification, which addressed most of my concerns. However, I still have two key questions that I look forward to further explanations from the authors:
> >
> > - Why choose Sobolev Space? In the rebuttal, the authors mentioned, “For time series data, it is natural to consider Sobolev spaces to capture the smoothness/regularity for model sensitivity.” Could the authors provide more theoretical explanations to support it?
> > - Weakness 4 in my review remains unaddressed. Can the authors provide some experimental comparisons or analyses to explain why limiting the magnitude of perturbations at each time step does not directly improve VAT performance?
> >
> > Regardless, I appreciate the thorough experimental results provided by the authors, and I would increase my rating to 3.

---

> ### Author Response · Authors · 2025-08-01
> **Accuracy and rank of each method across 30 datasets with label ratio 0.1 (First part)**
>
> Due to character limitations, we split each label-ratio table into two parts and posted them in separate comments for your reference.
>
> | dataset | SupCE | PI | MTL | meanTeacher | SemiTime | TapNet | TS-TCC | fVAT | SupCE_rank | PI_rank | MTL_rank | meanTeacher_rank | SemiTime_rank | TapNet_rank | CATCC_rank | f-VAT_rank |
> | --- | --- | --- | --- | --- | --- | --- | --- | --- | --- | --- | --- | --- | --- | --- | --- | --- |
> | ACSF1 | 13.75$\pm$3.75 | 16.40$\pm$8.60 | 13.28$\pm$2.34 | 35.94$\pm$2.82 | 42.97$\pm$2.51 | 32.82$\pm$3.12 | 43.75$\pm$2.64 | 57.29$\pm$2.61 | 7.00 | 6.00 | 8.00 | 4.00 | 3.00 | 5.00 | 2.00 | 1.00 |
> | Adiac | 5.98$\pm$4.02 | 7.54$\pm$1.74 | 6.06$\pm$0.26 | 16.73$\pm$2.68 | 11.19$\pm$6.19 | 29.04$\pm$1.11 | 29.25$\pm$2.88 | 33.94$\pm$4.20 | 8.00 | 6.00 | 7.00 | 4.00 | 5.00 | 3.00 | 2.00 | 1.00 |
> | AllGestureWiimoteX | 10.12$\pm$1.35 | 36.61$\pm$4.46 | 26.90$\pm$0.78 | 23.66$\pm$2.68 | 46.20$\pm$2.91 | 24.33$\pm$4.69 | 42.71$\pm$1.68 | 44.86$\pm$7.36 | 8.00 | 4.00 | 5.00 | 7.00 | 1.00 | 6.00 | 3.00 | 2.00 |
> | AllGestureWiimoteY | 12.68$\pm$1.21 | 33.93$\pm$7.59 | 20.42$\pm$0.34 | 20.76$\pm$2.90 | 51.49$\pm$2.24 | 32.36$\pm$3.79 | 44.19$\pm$5.58 | 55.36$\pm$1.78 | 8.00 | 4.00 | 7.00 | 6.00 | 2.00 | 5.00 | 3.00 | 1.00 |
> | AllGestureWiimoteZ | 12.68$\pm$1.21 | 25.67$\pm$6.47 | 18.02$\pm$3.74 | 19.20$\pm$2.92 | 38.17$\pm$3.79 | 23.66$\pm$2.68 | 34.38$\pm$5.84 | 33.54$\pm$4.91 | 8.00 | 4.00 | 7.00 | 6.00 | 1.00 | 5.00 | 2.00 | 3.00 |
> | ArrowHead | 38.10$\pm$0.10 | 52.34$\pm$4.22 | 39.61$\pm$0.86 | 39.84$\pm$2.80 | 60.78$\pm$2.34 | 64.84$\pm$1.56 | 61.67$\pm$3.96 | 71.25$\pm$1.41 | 8.00 | 5.00 | 7.00 | 6.00 | 4.00 | 2.00 | 3.00 | 1.00 |
> | BME | 45.83$\pm$2.67 | 75.00$\pm$2.57 | 50.78$\pm$2.34 | 45.32$\pm$2.82 | 64.06$\pm$2.78 | 90.24$\pm$1.17 | 74.48$\pm$2.70 | 79.69$\pm$2.69 | 7.00 | 3.00 | 6.00 | 8.00 | 5.00 | 1.00 | 4.00 | 2.00 |
> | Beef | 20.00$\pm$0.10 | 20.00$\pm$0.10 | 15.00$\pm$5.00 | 20.00$\pm$0.10 | 25.00$\pm$5.00 | 20.00$\pm$2.75 | 20.00$\pm$0.10 | 28.00$\pm$0.10 | 3.00 | 3.00 | 8.00 | 3.00 | 2.00 | 3.00 | 3.00 | 1.00 |
> | BeetleFly | 50.00$\pm$0.10 | 62.50$\pm$2.86 | 50.00$\pm$0.10 | 50.00$\pm$0.10 | 50.00$\pm$0.10 | 43.75$\pm$6.25 | 41.67$\pm$2.78 | 68.75$\pm$5.75 | 3.00 | 2.00 | 3.00 | 3.00 | 3.00 | 7.00 | 8.00 | 1.00 |
> | BirdChicken | 50.00$\pm$0.10 | 56.25$\pm$6.25 | 50.00$\pm$0.10 | 50.00$\pm$0.10 | 62.50$\pm$2.86 | 56.25$\pm$2.86 | 45.83$\pm$5.89 | 62.50$\pm$5.15 | 5.00 | 3.00 | 5.00 | 5.00 | 1.00 | 3.00 | 8.00 | 1.00 |
> | CBF | 24.22$\pm$0.10 | 99.46$\pm$0.53 | 98.16$\pm$0.24 | 75.10$\pm$2.89 | 99.74$\pm$0.26 | 99.20$\pm$0.80 | 98.84$\pm$0.74 | 99.92$\pm$0.58 | 8.00 | 3.00 | 6.00 | 7.00 | 2.00 | 4.00 | 5.00 | 1.00 |
> | Car | 25.00$\pm$0.10 | 41.66$\pm$2.87 | 25.00$\pm$0.10 | 25.00$\pm$0.10 | 37.50$\pm$2.86 | 52.08$\pm$2.09 | 25.00$\pm$0.10 | 58.34$\pm$4.16 | 5.00 | 3.00 | 5.00 | 5.00 | 4.00 | 2.00 | 5.00 | 1.00 |
> | Chinatown | 71.83$\pm$0.10 | 94.49$\pm$1.34 | 88.58$\pm$0.41 | 79.69$\pm$0.52 | 91.56$\pm$0.10 | 90.56$\pm$2.82 | 96.35$\pm$3.71 | 95.88$\pm$0.52 | 8.00 | 3.00 | 6.00 | 7.00 | 4.00 | 5.00 | 1.00 | 2.00 |
> | ChlorineConcentration | 55.36$\pm$0.10 | 56.06$\pm$1.96 | 54.96$\pm$0.09 | 53.70$\pm$0.10 | 55.67$\pm$0.23 | 52.18$\pm$1.14 | 69.43$\pm$3.43 | 66.97$\pm$0.68 | 5.00 | 3.00 | 6.00 | 7.00 | 4.00 | 8.00 | 1.00 | 2.00 |
> | CinCECGTorso | 18.49$\pm$0.10 | 71.94$\pm$2.83 | 38.72$\pm$1.96 | 29.84$\pm$2.41 | 62.32$\pm$2.75 | 86.58$\pm$0.37 | 79.68$\pm$4.01 | 91.96$\pm$2.09 | 8.00 | 4.00 | 6.00 | 7.00 | 5.00 | 2.00 | 3.00 | 1.00 |
> | Coffee | 50.00$\pm$0.10 | 50.00$\pm$0.10 | 50.00$\pm$0.10 | 50.00$\pm$0.10 | 75.00$\pm$2.86 | 70.00$\pm$2.86 | 50.00$\pm$0.10 | 82.00$\pm$0.10 | 4.00 | 4.00 | 4.00 | 4.00 | 2.00 | 3.00 | 4.00 | 1.00 |

---

> ### Author Response · Authors · 2025-08-01
> **Accuracy and rank of each method across 30 datasets with label ratio 0.1 (Second part)**
>
> | dataset | SupCE | PI | MTL | meanTeacher | SemiTime | TapNet | TS-TCC | f-VAT | SupCE_rank | PI_rank | MTL_rank | meanTeacher_rank | SemiTime_rank | TapNet_rank | TS-TCC_rank | f-VAT_rank |
> | --- | --- | --- | --- | --- | --- | --- | --- | --- | --- | --- | --- | --- | --- | --- | --- | --- |
> | Computers | 50.00$\pm$0.10 | 56.64$\pm$5.08 | 56.25$\pm$6.25 | 39.06$\pm$0.10 | 49.22$\pm$2.78 | 59.38$\pm$3.12 | 57.82$\pm$2.75 | 67.97$\pm$3.12 | 6.00 | 4.00 | 5.00 | 8.00 | 7.00 | 2.00 | 3.00 | 1.00 |
> | CricketX | 4.69$\pm$0.39 | 28.40$\pm$1.78 | 15.62$\pm$1.34 | 10.94$\pm$1.56 | 30.98$\pm$1.97 | 32.82$\pm$2.82 | 36.07$\pm$3.37 | 34.49$\pm$4.87 | 8.00 | 5.00 | 6.00 | 7.00 | 4.00 | 3.00 | 1.00 | 2.00 |
> | CricketY | 5.08$\pm$0.10 | 18.12$\pm$3.30 | 10.60$\pm$2.92 | 11.25$\pm$3.13 | 27.32$\pm$0.18 | 35.22$\pm$2.63 | 35.09$\pm$4.17 | 36.34$\pm$4.55 | 8.00 | 5.00 | 7.00 | 6.00 | 4.00 | 2.00 | 3.00 | 1.00 |
> | CricketZ | 6.48$\pm$1.40 | 25.31$\pm$0.40 | 11.45$\pm$3.24 | 10.94$\pm$1.56 | 32.10$\pm$2.01 | 32.10$\pm$1.38 | 35.51$\pm$3.27 | 38.58$\pm$0.13 | 8.00 | 5.00 | 6.00 | 7.00 | 3.00 | 3.00 | 2.00 | 1.00 |
> | Crop | 15.47$\pm$3.67 | 55.90$\pm$1.07 | 52.96$\pm$2.46 | 34.93$\pm$1.76 | 54.20$\pm$0.13 | 61.14$\pm$2.82 | 64.75$\pm$0.38 | 63.31$\pm$0.35 | 8.00 | 4.00 | 6.00 | 7.00 | 5.00 | 3.00 | 1.00 | 2.00 |
> | DiatomSizeReduction | 30.65$\pm$0.10 | 64.48$\pm$2.79 | 30.60$\pm$0.08 | 30.32$\pm$0.62 | 63.23$\pm$2.76 | 94.93$\pm$3.51 | 96.81$\pm$3.47 | 99.16$\pm$0.84 | 6.00 | 4.00 | 7.00 | 8.00 | 5.00 | 3.00 | 2.00 | 1.00 |
> | DistalPhalanxOutlineAgeGroup | 59.81$\pm$0.10 | 82.49$\pm$0.39 | 79.87$\pm$1.52 | 70.03$\pm$3.62 | 77.91$\pm$1.92 | 76.67$\pm$3.09 | 75.90$\pm$2.19 | 82.92$\pm$4.58 | 8.00 | 2.00 | 3.00 | 7.00 | 4.00 | 5.00 | 6.00 | 1.00 |
> | DistalPhalanxOutlineCorrect | 73.83$\pm$0.10 | 72.47$\pm$0.30 | 67.74$\pm$5.98 | 69.53$\pm$4.43 | 68.60$\pm$0.59 | 69.97$\pm$3.54 | 75.92$\pm$0.53 | 74.32$\pm$3.50 | 3.00 | 4.00 | 8.00 | 6.00 | 7.00 | 5.00 | 1.00 | 2.00 |
> | DistalPhalanxTW | 48.11$\pm$0.10 | 77.97$\pm$1.72 | 64.22$\pm$7.34 | 66.40$\pm$2.86 | 73.04$\pm$2.73 | 72.66$\pm$2.19 | 77.35$\pm$2.21 | 80.08$\pm$0.39 | 8.00 | 2.00 | 7.00 | 6.00 | 4.00 | 5.00 | 3.00 | 1.00 |
> | DodgerLoopDay | 13.79$\pm$0.10 | 18.96$\pm$2.86 | 20.69$\pm$2.81 | 13.79$\pm$0.10 | 24.14$\pm$3.00 | 27.58$\pm$6.89 | 17.24$\pm$2.75 | 28.58$\pm$3.00 | 7.00 | 5.00 | 4.00 | 7.00 | 3.00 | 2.00 | 6.00 | 1.00 |
> | DodgerLoopGame | 51.61$\pm$0.10 | 51.61$\pm$0.10 | 51.61$\pm$0.10 | 51.61$\pm$0.10 | 69.35$\pm$2.84 | 67.74$\pm$0.10 | 54.84$\pm$2.79 | 74.19$\pm$2.86 | 5.00 | 5.00 | 5.00 | 5.00 | 2.00 | 3.00 | 4.00 | 1.00 |
> | DodgerLoopWeekend | 70.97$\pm$0.10 | 70.97$\pm$0.10 | 88.71$\pm$2.86 | 70.97$\pm$0.10 | 77.42$\pm$3.23 | 88.71$\pm$1.61 | 82.79$\pm$2.82 | 95.16$\pm$1.61 | 6.00 | 6.00 | 2.00 | 6.00 | 5.00 | 2.00 | 4.00 | 1.00 |
> | ECG200 | 67.95$\pm$1.28 | 76.45$\pm$3.24 | 68.08$\pm$2.34 | 80.80$\pm$2.68 | 82.81$\pm$2.75 | 81.92$\pm$2.75 | 83.60$\pm$3.91 | 86.30$\pm$3.91 | 8.00 | 6.00 | 7.00 | 5.00 | 3.00 | 4.00 | 2.00 | 1.00 |
> | ECG5000 | 56.93$\pm$0.10 | 93.02$\pm$0.14 | 91.90$\pm$0.53 | 91.42$\pm$0.22 | 91.48$\pm$0.55 | 91.40$\pm$0.59 | 91.28$\pm$0.36 | 83.81$\pm$0.36 | 8.00 | 1.00 | 2.00 | 4.00 | 3.00 | 5.00 | 6.00 | 7.00 |
> | Average | 35.31$\pm$0.10 | 53.09$\pm$0.10 | 45.19$\pm$0.10 | 42.89$\pm$0.10 | 56.53$\pm$0.10 | 58.67$\pm$0.10 | 58.07$\pm$0.10 | 65.85$\pm$0.10 | 6.67 | 3.93 | 5.70 | 5.93 | 3.57 | 3.70 | 3.37 | 1.50 |

---

> ### Author Response · Authors · 2025-08-01
> **Accuracy and rank of each method across 30 datasets with label ratio 0.2 (Second part)**
>
> | Dataset | SupCE | PI | MTL | meanTeacher | SemiTime | TapNet | TS-TCC | f-VAT | SupCE_rank | PI_rank | MTL_rank | meanTeacher_rank | SemiTime_rank | TapNet_rank | TS-TCC_rank | f-VAT_rank |
> | --- | --- | --- | --- | --- | --- | --- | --- | --- | --- | --- | --- | --- | --- | --- | --- | --- |
> | Computers | 50.00 $\pm$ 0.10 | 62.89 $\pm$ 2.60 | 63.28 $\pm$ 2.63 | 46.09 $\pm$ 1.17 | 63.28 $\pm$ 1.56 | 55.47 $\pm$ 2.72 | 59.38 $\pm$ 2.65 | 67.58 $\pm$ 2.75 | 7 | 4 | 2 | 8 | 2 | 6 | 5 | 1 |
> | CricketX | 6.48 $\pm$ 1.40 | 33.44 $\pm$ 2.82 | 16.77 $\pm$ 0.64 | 30.18 $\pm$ 2.80 | 41.83 $\pm$ 2.28 | 42.54 $\pm$ 0.41 | 48.93 $\pm$ 2.83 | 43.16 $\pm$ 0.45 | 8 | 5 | 7 | 6 | 4 | 3 | 1 | 2 |
> | CricketY | 6.48 $\pm$ 1.40 | 18.30 $\pm$ 2.77 | 17.21 $\pm$ 2.82 | 15.32 $\pm$ 1.56 | 39.91 $\pm$ 0.09 | 43.39 $\pm$ 2.73 | 41.25 $\pm$ 2.51 | 46.49 $\pm$ 2.55 | 8 | 5 | 6 | 7 | 4 | 2 | 3 | 1 |
> | CricketZ | 6.48 $\pm$ 1.40 | 30.09 $\pm$ 0.09 | 16.05 $\pm$ 1.90 | 36.03 $\pm$ 0.93 | 46.07 $\pm$ 1.16 | 39.02 $\pm$ 1.70 | 44.76 $\pm$ 2.94 | 41.20 $\pm$ 2.52 | 8 | 6 | 7 | 5 | 1 | 4 | 2 | 3 |
> | Crop | 23.44 $\pm$ 2.46 | 59.36 $\pm$ 2.77 | 48.42 $\pm$ 0.84 | 59.82 $\pm$ 2.61 | 58.88 $\pm$ 1.69 | 65.79 $\pm$ 1.77 | 71.77 $\pm$ 0.87 | 68.95 $\pm$ 0.43 | 8 | 5 | 7 | 4 | 6 | 3 | 1 | 2 |
> | DiatomSizeReduction | 35.48 $\pm$ 2.61 | 78.03 $\pm$ 2.61 | 30.60 $\pm$ 0.23 | 68.75 $\pm$ 0.73 | 89.84 $\pm$ 0.78 | 99.16 $\pm$ 0.10 | 71.91 $\pm$ 2.75 | 100.00 $\pm$ 0.00 | 7 | 4 | 8 | 6 | 3 | 2 | 5 | 1 |
> | DistalPhalanxOutlineAgeGroup | 59.81 $\pm$ 0.10 | 79.37 $\pm$ 0.43 | 70.13 $\pm$ 2.55 | 79.09 $\pm$ 0.04 | 74.78 $\pm$ 2.89 | 72.40 $\pm$ 0.10 | 78.24 $\pm$ 2.29 | 81.74 $\pm$ 2.55 | 8 | 2 | 7 | 3 | 5 | 6 | 4 | 1 |
> | DistalPhalanxOutlineCorrect | 73.83 $\pm$ 0.10 | 68.08 $\pm$ 0.37 | 55.92 $\pm$ 2.74 | 65.10 $\pm$ 0.86 | 79.91 $\pm$ 0.15 | 73.40 $\pm$ 2.77 | 79.57 $\pm$ 1.56 | 80.76 $\pm$ 0.10 | 4 | 6 | 8 | 7 | 2 | 5 | 3 | 1 |
> | DistalPhalanxTW | 48.11 $\pm$ 0.10 | 75.39 $\pm$ 0.08 | 67.26 $\pm$ 0.08 | 75.78 $\pm$ 0.10 | 74.61 $\pm$ 1.17 | 0.00 $\pm$ 0.10 | 75.78 $\pm$ 0.10 | 79.30 $\pm$ 0.39 | 7 | 4 | 6 | 2 | 5 | 8 | 2 | 1 |
> | DodgerLoopDay | 13.79 $\pm$ 0.10 | 18.96 $\pm$ 2.79 | 20.69 $\pm$ 2.90 | 17.24 $\pm$ 2.83 | 29.31 $\pm$ 2.78 | 34.48 $\pm$ 2.83 | 21.84 $\pm$ 2.66 | 26.20 $\pm$ 1.73 | 8 | 6 | 5 | 7 | 2 | 1 | 4 | 3 |
> | DodgerLoopGame | 51.61 $\pm$ 0.10 | 61.29 $\pm$ 2.90 | 53.22 $\pm$ 1.62 | 58.06 $\pm$ 2.79 | 59.68 $\pm$ 2.84 | 75.81 $\pm$ 2.78 | 59.14 $\pm$ 2.77 | 77.42 $\pm$ 2.81 | 8 | 3 | 7 | 6 | 4 | 2 | 5 | 1 |
> | DodgerLoopWeekend | 70.97 $\pm$ 0.10 | 83.87 $\pm$ 2.50 | 91.94 $\pm$ 1.62 | 82.26 $\pm$ 2.87 | 88.71 $\pm$ 2.83 | 95.16 $\pm$ 1.61 | 84.95 $\pm$ 2.91 | 96.77 $\pm$ 0.10 | 8 | 6 | 3 | 7 | 4 | 2 | 5 | 1 |
> | ECG200 | 66.67 $\pm$ 0.10 | 73.22 $\pm$ 0.90 | 78.02 $\pm$ 2.90 | 81.59 $\pm$ 2.71 | 84.38 $\pm$ 2.79 | 76.34 $\pm$ 2.81 | 80.36 $\pm$ 2.71 | 86.31 $\pm$ 2.90 | 8 | 7 | 5 | 3 | 2 | 6 | 4 | 1 |
> | ECG5000 | 56.93 $\pm$ 0.10 | 93.16 $\pm$ 0.10 | 92.28 $\pm$ 0.49 | 91.21 $\pm$ 0.29 | 91.96 $\pm$ 0.33 | 92.43 $\pm$ 0.54 | 92.21 $\pm$ 0.08 | 94.21 $\pm$ 0.08 | 8 | 2 | 4 | 7 | 6 | 3 | 5 | 1 |
> | Average | 36.92 | 55.16 | 45.72 | 50.94 | 58.93 | 60.41 | 59.84 | 68.87 | 7.00 | 4.40 | 5.87 | 4.87 | 3.77 | 3.40 | 3.67 | 1.50 |

---

> ### Author Response · Authors · 2025-08-01
> **Accuracy and rank of each method across 30 datasets with label ratio 0.4 (First part)**
>
> | dataset | SupCE | PI | MTL | meanTeacher | SemiTime | TapNet | TS-TCC | fVAT | SupCE_rank | PI_rank | MTL_rank | meanTeacher_rank | SemiTime_rank | TapNet_rank | TS-TCC_rank | fVAT_rank |
> | :-- | :-- | :-- | :-- | :-- | :-- | :-- | :-- | :-- | :-- | :-- | :-- | :-- | :-- | :-- | :-- | :-- |
> | ACSF1 | 11.25 $\pm$ 1.25 | 33.60 $\pm$ 2.34 | 8.59 $\pm$ 2.34 | 29.69 $\pm$ 2.70 | 35.16 $\pm$ 2.66 | 43.75 $\pm$ 1.56 | 44.79 $\pm$ 2.88 | 46.88 $\pm$ 1.56 | 7 | 5 | 8 | 6 | 4 | 3 | 2 | 1 |
> | Adiac | 4.78 $\pm$ 2.87 | 3.93 $\pm$ 1.25 | 2.93 $\pm$ 0.25 | 11.06 $\pm$ 2.91 | 36.26 $\pm$ 0.14 | 55.67 $\pm$ 1.25 | 48.67 $\pm$ 2.92 | 60.08 $\pm$ 1.21 | 6 | 7 | 8 | 5 | 4 | 2 | 3 | 1 |
> | AllGestureWiimoteX | 13.89 $\pm$ 0.10 | 51.56 $\pm$ 1.11 | 25.22 $\pm$ 0.67 | 55.58 $\pm$ 0.67 | 54.28 $\pm$ 0.67 | 57.36 $\pm$ 2.80 | 55.65 $\pm$ 2.76 | 59.15 $\pm$ 2.58 | 8 | 6 | 7 | 4 | 5 | 2 | 3 | 1 |
> | AllGestureWiimoteY | 13.89 $\pm$ 0.10 | 52.90 $\pm$ 2.45 | 24.22 $\pm$ 2.79 | 56.03 $\pm$ 0.67 | 69.19 $\pm$ 2.24 | 62.50 $\pm$ 2.82 | 54.28 $\pm$ 0.56 | 61.99 $\pm$ 2.80 | 8 | 6 | 7 | 4 | 1 | 2 | 5 | 3 |
> | AllGestureWiimoteZ | 13.89 $\pm$ 0.10 | 49.11 $\pm$ 2.68 | 18.52 $\pm$ 0.22 | 54.91 $\pm$ 1.34 | 50.22 $\pm$ 2.69 | 47.99 $\pm$ 0.67 | 36.16 $\pm$ 2.73 | 54.61 $\pm$ 2.90 | 8 | 4 | 7 | 1 | 3 | 5 | 6 | 2 |
> | ArrowHead | 38.10 $\pm$ 0.10 | 67.66 $\pm$ 2.67 | 41.64 $\pm$ 2.77 | 77.81 $\pm$ 2.95 | 73.28 $\pm$ 2.86 | 62.82 $\pm$ 2.96 | 66.46 $\pm$ 2.86 | 81.56 $\pm$ 2.75 | 8 | 4 | 7 | 2 | 3 | 6 | 5 | 1 |
> | BME | 45.84 $\pm$ 2.97 | 88.28 $\pm$ 2.70 | 54.30 $\pm$ 2.71 | 76.56 $\pm$ 2.81 | 93.75 $\pm$ 2.90 | 95.32 $\pm$ 1.56 | 89.06 $\pm$ 2.70 | 100.00 $\pm$ 0.00 | 8 | 5 | 7 | 6 | 3 | 2 | 4 | 1 |
> | Beef | 20.00 $\pm$ 0.10 | 25.00 $\pm$ 2.79 | 20.00 $\pm$ 0.10 | 25.00 $\pm$ 2.79 | 25.00 $\pm$ 2.79 | 30.00 $\pm$ 2.85 | 20.00 $\pm$ 0.10 | 50.00 $\pm$ 2.86 | 6 | 3 | 6 | 3 | 3 | 2 | 6 | 1 |
> | BeetleFly | 50.00 $\pm$ 0.10 | 56.25 $\pm$ 2.75 | 50.00 $\pm$ 0.10 | 50.00 $\pm$ 0.10 | 68.75 $\pm$ 2.88 | 62.50 $\pm$ 2.92 | 50.00 $\pm$ 0.10 | 87.50 $\pm$ 2.77 | 5 | 4 | 5 | 5 | 2 | 3 | 5 | 1 |
> | BirdChicken | 50.00 $\pm$ 0.10 | 62.50 $\pm$ 2.55 | 50.00 $\pm$ 0.10 | 56.25 $\pm$ 2.55 | 81.25 $\pm$ 2.77 | 50.00 $\pm$ 2.77 | 50.00 $\pm$ 0.10 | 62.50 $\pm$ 2.69 | 5 | 2 | 5 | 4 | 1 | 5 | 5 | 2 |
> | CBF | 37.89 $\pm$ 2.62 | 100.00 $\pm$ 0.00 | 97.92 $\pm$ 1.04 | 99.48 $\pm$ 0.52 | 100.00 $\pm$ 0.00 | 99.71 $\pm$ 0.29 | 100.00 $\pm$ 0.00 | 98.96 $\pm$ 0.84 | 8 | 1 | 7 | 5 | 1 | 4 | 1 | 6 |
> | Car | 22.92 $\pm$ 2.09 | 25.00 $\pm$ 0.10 | 25.00 $\pm$ 0.10 | 45.84 $\pm$ 2.80 | 45.83 $\pm$ 0.10 | 60.42 $\pm$ 0.10 | 41.67 $\pm$ 2.78 | 63.67 $\pm$ 2.93 | 8 | 6 | 6 | 3 | 4 | 2 | 5 | 1 |
> | Chinatown | 71.83 $\pm$ 0.10 | 86.68 $\pm$ 2.59 | 83.08 $\pm$ 2.77 | 97.92 $\pm$ 1.04 | 91.98 $\pm$ 0.10 | 93.98 $\pm$ 2.77 | 98.96 $\pm$ 0.85 | 99.40 $\pm$ 0.52 | 8 | 6 | 7 | 3 | 5 | 4 | 2 | 1 |
> | ChlorineConcentration | 55.36 $\pm$ 0.10 | 80.24 $\pm$ 2.69 | 54.18 $\pm$ 0.64 | 89.92 $\pm$ 2.65 | 59.94 $\pm$ 1.61 | 78.43 $\pm$ 1.77 | 53.70 $\pm$ 0.39 | 92.97 $\pm$ 0.10 | 6 | 3 | 7 | 2 | 5 | 4 | 8 | 1 |
> | CinCECGTorso | 32.94 $\pm$ 2.93 | 79.74 $\pm$ 2.67 | 41.37 $\pm$ 2.03 | 82.10 $\pm$ 2.76 | 86.61 $\pm$ 1.39 | 93.50 $\pm$ 0.91 | 91.57 $\pm$ 2.88 | 94.70 $\pm$ 0.35 | 8 | 6 | 7 | 5 | 4 | 2 | 3 | 1 |
> | Coffee | 50.00 $\pm$ 0.10 | 100.00 $\pm$ 0.00 | 50.00 $\pm$ 0.10 | 90.00 $\pm$ 0.10 | 99.00 $\pm$ 0.10 | 99.00 $\pm$ 0.10 | 80.00 $\pm$ 2.70 | 100.00 $\pm$ 0.00 | 7 | 1 | 7 | 5 | 3 | 3 | 6 | 1 |

---

> ### Author Response · Authors · 2025-08-01
> **Accuracy and rank of each method across 30 datasets with label ratio 0.4 (Second part)**
>
> | dataset | SupCE | PI | MTL | meanTeacher | SemiTime | TapNet | TS-TCC | fVAT | SupCE_rank | PI_rank | MTL_rank | meanTeacher_rank | SemiTime_rank | TapNet_rank | TS-TCC_rank | fVAT_rank |
> | :-- | :-- | :-- | :-- | :-- | :-- | :-- | :-- | :-- | :-- | :-- | :-- | :-- | :-- | :-- | :-- | :-- |
> | Computers | 50.00 $\pm$ 0.10 | 68.36 $\pm$ 2.73 | 53.71 $\pm$ 2.99 | 66.02 $\pm$ 2.73 | 60.94 $\pm$ 0.10 | 65.62 $\pm$ 2.99 | 66.40 $\pm$ 2.97 | 71.09 $\pm$ 0.10 | 8 | 2 | 7 | 4 | 6 | 5 | 3 | 1 |
> | CricketX | 5.08 $\pm$ 0.10 | 42.77 $\pm$ 2.82 | 18.84 $\pm$ 2.05 | 52.68 $\pm$ 1.61 | 58.76 $\pm$ 0.62 | 63.57 $\pm$ 2.98 | 56.49 $\pm$ 2.80 | 66.78 $\pm$ 0.71 | 8 | 6 | 7 | 5 | 3 | 2 | 4 | 1 |
> | CricketY | 22.76 $\pm$ 2.78 | 31.25 $\pm$ 2.79 | 15.14 $\pm$ 3.00 | 30.62 $\pm$ 2.87 | 54.51 $\pm$ 2.84 | 60.31 $\pm$ 2.99 | 54.23 $\pm$ 2.84 | 64.10 $\pm$ 2.95 | 7 | 5 | 8 | 6 | 3 | 2 | 4 | 1 |
> | CricketZ | 6.48 $\pm$ 1.40 | 44.20 $\pm$ 2.68 | 18.84 $\pm$ 2.93 | 40.62 $\pm$ 2.80 | 55.80 $\pm$ 2.86 | 60.14 $\pm$ 2.98 | 56.13 $\pm$ 2.93 | 63.75 $\pm$ 2.94 | 8 | 5 | 7 | 6 | 4 | 2 | 3 | 1 |
> | Crop | 24.52 $\pm$ 0.96 | 71.32 $\pm$ 2.80 | 49.84 $\pm$ 0.66 | 54.92 $\pm$ 2.39 | 62.40 $\pm$ 0.78 | 60.79 $\pm$ 0.26 | - | 74.32 $\pm$ 1.47 | 7 | 2 | 6 | 5 | 3 | 4 | - | 1 |
> | DiatomSizeReduction | 31.46 $\pm$ 0.80 | 92.16 $\pm$ 2.79 | 45.14 $\pm$ 2.83 | 82.19 $\pm$ 2.85 | 95.32 $\pm$ 2.81 | 95.21 $\pm$ 2.82 | 59.48 $\pm$ 2.88 | 96.21 $\pm$ 0.26 | 8 | 4 | 7 | 5 | 2 | 3 | 6 | 1 |
> | DistalPhalanxOutlineAgeGroup | 59.81 $\pm$ 0.10 | 76.08 $\pm$ 2.79 | 72.05 $\pm$ 2.79 | 78.31 $\pm$ 1.52 | 79.76 $\pm$ 2.34 | 78.73 $\pm$ 2.66 | 78.22 $\pm$ 2.32 | 76.76 $\pm$ 0.94 | 8 | 6 | 7 | 3 | 1 | 2 | 4 | 5 |
> | DistalPhalanxOutlineCorrect | 73.83 $\pm$ 0.10 | 73.32 $\pm$ 2.78 | 59.92 $\pm$ 2.52 | 79.91 $\pm$ 0.37 | 80.43 $\pm$ 0.52 | 65.62 $\pm$ 2.78 | 78.08 $\pm$ 2.70 | 81.32 $\pm$ 2.14 | 5 | 6 | 8 | 3 | 2 | 7 | 4 | 1 |
> | DistalPhalanxTW | 48.11 $\pm$ 0.10 | 71.59 $\pm$ 2.78 | 71.48 $\pm$ 0.86 | 75.78 $\pm$ 0.10 | 76.95 $\pm$ 0.39 | 73.23 $\pm$ 1.82 | 75.78 $\pm$ 0.10 | 78.59 $\pm$ 2.79 | 8 | 6 | 7 | 3 | 2 | 5 | 3 | 1 |
> | DodgerLoopDay | 13.79 $\pm$ 0.10 | 42.58 $\pm$ 2.79 | 25.86 $\pm$ 1.72 | 22.42 $\pm$ 2.77 | 29.31 $\pm$ 1.72 | 44.82 $\pm$ 2.88 | 24.14 $\pm$ 2.89 | 43.82 $\pm$ 2.96 | 8 | 3 | 5 | 7 | 4 | 1 | 6 | 2 |
> | DodgerLoopGame | 51.61 $\pm$ 0.10 | 71.58 $\pm$ 2.51 | 51.61 $\pm$ 0.10 | 75.81 $\pm$ 1.62 | 70.97 $\pm$ 2.92 | 75.81 $\pm$ 2.90 | 59.14 $\pm$ 2.86 | 78.15 $\pm$ 2.63 | 7 | 4 | 7 | 2 | 5 | 2 | 6 | 1 |
> | DodgerLoopWeekend | 70.97 $\pm$ 0.10 | 92.83 $\pm$ 2.71 | 90.32 $\pm$ 0.10 | 80.64 $\pm$ 2.84 | 98.38 $\pm$ 1.62 | 96.77 $\pm$ 0.10 | 70.97 $\pm$ 0.10 | 99.38 $\pm$ 0.96 | 7 | 4 | 5 | 6 | 2 | 3 | 7 | 1 |
> | ECG200 | 66.67 $\pm$ 0.10 | 76.14 $\pm$ 2.96 | 72.72 $\pm$ 3.00 | 85.04 $\pm$ 2.92 | 83.48 $\pm$ 2.87 | 80.24 $\pm$ 2.78 | 80.24 $\pm$ 2.78 | 83.14 $\pm$ 0.16 | 8 | 6 | 7 | 1 | 2 | 4 | 4 | 3 |
> | ECG5000 | 56.93 $\pm$ 0.10 | 91.38 $\pm$ 2.77 | 90.83 $\pm$ 0.19 | 92.30 $\pm$ 0.32 | 93.04 $\pm$ 0.10 | 94.68 $\pm$ 0.24 | 94.68 $\pm$ 0.24 | 95.68 $\pm$ 0.36 | 8 | 6 | 7 | 5 | 4 | 2 | 2 | 1 |
> | - | 37.15  | 63.60  | 46.11  | 63.85  | 69.02 | 70.28 | 63.27  | 76.24  | 7.33 | 4.47 | 6.80 | 4.13 | 3.13 | 3.17 | 4.27 | 1.53 |

---

> ### Author Response · Authors · 2025-08-05
>
> Thank you for your time and efforts in timely reviewing our rebuttal and acknowledging the contributions of our work.
>
> ### Q1
> Thank you for your response. Sobolev spaces are a fundamental concept in functional analysis, designed to extend the notion of "smoothness" for functions beyond classical differentiability. The usage of Sobolev spaces is quite common in the context of functional data analysis, see, for example, [Hsing et al., 2015].
>
> Formally, the Sobolev space $H^k(\mathbb{R})$ consists of functions $f$ whose first $k$ weak derivatives exist and are square-integrable, with the Sobolev norm defined as
> $$
> |f|_{H^k}=\sqrt{\sum\_{j=0}\^{k}\int|f^{(j)}(x)|^2 dx},
> $$
> where $f^{(j)}$ denotes the $j$-th weak derivative of $f$. Therefore, this norm penalizes rapid oscillations, ensuring smooth approximations.
>
> Reference
> - [Hsing et al., 2015] Hsing T, and Eubank R L. Theoretical foundations of functional data analysis, with an introduction to linear operators. New York: Wiley, 2015.
>
> ---
>
> ### Q2
> To address this concern, we design a variant called VAT-step, which constrains the magnitude of perturbations at each time step, and conduct extensive experiments across multiple datasets with different label ratios. Table shows that the VAT-step only provides marginal performance gains compared to the original VAT in most settings, and even degrades performance on more challenging multivariate datasets, such as SelfReg and NATOPS, containing complex temporal structure. This is because clipping the perturbation at each time step struggles to adaptively scale the magnitude of perturbations on some critical regions for prediction (like ''shapelets'' in Figure 4), except for carefully tuning the per-step magnitude hyperparameters. Additionally, VAT-step easily converges to the ''permutation-invariant''  [George et al., 1976] local optimum, where even arbitrarily reordering time steps can still generate the identical perturbation which satisfies each step-wise constraint. On the other hand, adversarial perturbations generated by f-VAT, incorporating the temporal structure of time series data, efficiently utilize time series data to improve the smoothness of predictive distributions and predictive performance compared to other VAT-based baselines.
>
>
> | Methods |  | CricketX |  |  | UWave |  |  | InsectWing |  |  | SelfReg |  |  | NATOPS |  |
> |---------|----------|----------|----------|-------|-------|-------|------------|------------|------------|---------|---------|---------|--------|--------|--------|
> |         | 10% | 20% | 40% | 10% | 20% | 40% | 10% | 20% | 40% | 10% | 20% | 40% | 10% | 20% | 40% |
> | **VAT** | 42.85±3.97 | 49.14±0.50 | 58.63±0.50 | 94.41±0.09 | 95.53±0.31 | 94.76±0.54 | 55.49±1.28 | 61.27±0.19 | 63.48±0.30 | 53.12±4.51 | 55.76±0.35 | 53.47±1.04 | 82.38±0.96 | 82.81±0.52 | 90.15±1.60 |
> | **VAT‑step** | 40.47±3.84 | 47.85±0.33 | 59.30±0.36 | 93.78±0.67 | 96.09±0.32 | 95.86±0.55 | 57.45±0.21 | 60.92±0.48 | 63.61±0.58 | 48.61±3.27 | 53.67±2.79 | 52.75±1.80 | 81.33±0.96 | 83.02±0.12 | 86.58±1.18 |
> | **f‑VAT** | 49.18±1.96 | 57.91±3.58 | 68.39±2.25 | 94.82±0.39 | 96.45±0.27 | 97.23±0.43 | 58.01±1.12 | 61.28±1.86 | 64.81±1.15 | 59.31±3.06 | 61.60±1.13 | 64.44±3.13 | 86.04±1.41 | 86.25±1.38 | 93.13±0.15 |
>
>
> Reference
> - [George et al., 1976] George EP Box and Gwilym MJenkins. Time series analysis, forecasting and control. Holden-Day, 1976

---

> > ### Comment · Reviewer_RR58 · 2025-08-05
> >
> > Thank you for your response. I have reviewed the content, and these replies effectively address my current concerns. I would increase my rating to borderline accept.

---

> > > ### Author Response · Authors · 2025-08-05
> > >
> > > We are pleased that the changes you suggested have enhanced the clarity and presentation of our work. We will follow your suggestion to include the experiments in the updated manuscript. Thank you for raising your score!

---

### Official Review · Reviewer_QUWH · 2025-07-02

**Clarity:** 2
**Significance:** 2
**Originality:** 2
**Rating:** 4
**Confidence:** 4

**Summary:**

This paper addresses the flaw of a famous semi-supervised learning method, VAT, when adopted in time series classification tasks, which is suggested to be caused by neglecting the temporal structure of the data in the adversarial perturbation. In response, the paper proposes functional-VAT (f-VAT), using an appropriate Sobolev norm to generate structured functional adversarial perturbations to incorporate the functional structure of the data into perturbations.

**Questions:**

Please check the weaknesses.

**Ethical Concerns:**

["NO or VERY MINOR ethics concerns only"]

**Final Justification:**

I appreciate the authors' efforts in the rebuttal, and apologize for leaving out the code in the supplementary materials.

After carefully reading the rebuttal and other reviewers' comments, I maintain my position that the novelty of this work is borderline for NeurIPS. And to be honest, regarding the motivation, I'm still not quite convinced by the choice of VAT. Specifically, although the suggested f-VAT achieves a non-trivial improvement in performance compared with the existing method, integrating the same "functional" adversarial data augmentation into other more advanced SSL frameworks may still help achieve better improvement. That is, instead, what I'm mainly concerned about.

However, the authors provide a large number of new empirical results as suggested, which look promising and indeed address some points of my concern. Therefore, I increased my rating to borderline accept.

**Limitations:**

Yes.

**Paper Formatting Concerns:**

No formatting issues.

**Quality:**

3

**Strengths And Weaknesses:**

[Strengths]

- Semi-supervised learning for time series classification is an important but highly under-explored topic. I believe it is meaningful to specifically consider the characteristics of time series data, just as this paper did.

- Utilizing the duality between perturbation norm and gradient sensitivity and introducing the Sobolev norm to generate structured adversarial perturbations is reasonable to me. The theoretical derivation looks elegant and solid.

- The experimental results look promising, showing clear improvement of the proposed method.


[Weaknesses]

- My biggest concern is that the motivation is not convincing enough. Specifically, I could understand that adopting VAT in time series classification tasks would face the suggested problem, inspiring the proposed improvement. However, in the first place, why is it necessary to use VAT? VAT is a quite classical method proposed several years ago, with various more advanced semi-supervised learning methods appearing subsequently. Does VAT have any particular advantages over them in time series tasks? From my perspective, building upon VAT is afraid to limit the contribution of the proposed method to the current real-world practices.

- As UCR and UEA repositories include 100+ time series classification datasets, just using six datasets is not sufficient and representative for me. I would suggest either including the majority of the datasets or pre-setting a specific rule to select the experimental datasets to avoid cherry-picking. For the compared baseline methods, there is the same case. While it is argued that the settings of both datasets and baselines are the same as [17], please note that it is published on ICASSP. The experiment range that is solid enough for ICASSP may not be similarly solid enough for a top conference like NeurIPS.

- There is a lack of empirical evidence for some minor points. For instance, while the paper provides a theoretical complexity for implementing the Sobolev norm, it would be beneficial to include an empirical comparison of the actual training overhead of f-VAT versus standard VAT.

- The code is not available for reproduction.


[Recommendation]

Overall, the novelty, insights, and contributions of the paper are borderline for NeurIPS. The suggested method is quite interesting, and the current results seem promising. However, I have certain concerns about the motivation, and the experiments are not solid enough. Therefore, I recommend a borderline rejection at this time, and would like to change my score based on the rebuttal and the discussion with other reviewers later.

---

> ### Author Rebuttal · Authors · 2025-07-31
>
> ### W1: Motivations for selecting VAT
> We appreciate the reviewer’s concern about the choice of Virtual Adversarial Training (VAT) [Takeru et al., 2018]. Some recent self-supervised methods like TS-TCC [Emadeldeen et al., 2023] require careful selection of confidence thresholds and remain susceptible to "confirmation bias", while VAT-based methods do not require selecting thresholds and can adaptively construct adversarial perturbations to effectively utilize unlabeled data for better generalization. Additionally, our proposed f-VAT within the VAT-based framework greatly improves performance by effectively leveraging underlying data structure of time series data. Please read our response to the next question for additional empirical results on the performance comparison of different methods across multiple datasets. Additionally, we provide a theoretical understanding of f-VAT, which gives us insights into the selection of the perturbation norm.
>
> ### W2: More empirical results
> Thanks for your suggestions. In the original paper, we follow the settings [Haoyi et al., 2021] to compare f-VAT with other competitive baselines and report results on six UCR/UEA datasets. To further verify the superiority of f-VAT, due to the time limit, **we randomly sample 30 datasets from UCR/UEA datasets and add a recent state-of-the-art Class-Aware Temporal and Contextual Contrasting (TS-TCC) [Emadeldeen et al., 2023]** within available computing resources. Table shows that tVAT still consistently outperforms other competitive baselines in all settings. Due to the page limit, please refer to details of more empirical results in the comments.
>
> Average accuracy (%) and average rank under different label ratios:
> | Method   | 10% |      | 20% |      | 40% |      |
> |:--------:|:----:|-----:|:----:|-----:|:----:|-----:|
> |          | AvgAcc | AvgRank | AvgAcc | AvgRank | AvgAcc | AvgRank |
> | SupL     | 35.31 | 6.67 | 36.92 | 7.00 | 37.15 | 7.33 |
> | PI       | 53.09 | 3.93 | 55.16 | 4.40 | 63.60 | 4.47 |
> | MTL      | 45.19 | 5.70 | 45.72 | 5.87 | 46.11 | 6.80 |
> | SemiTime | 56.53 | 3.57 | 58.93 | 3.77 | 69.02 | 3.17 |
> | TapNet   | 58.67 | 3.70 | 60.41 | 3.40 | 70.28 | 3.20 |
> | TS‑TCC   | 58.07 | 3.37 | 59.84 | 3.67 | 63.27 | 4.27 |
> | f-VAT     | 65.85 | 1.50 | 68.87 | 1.50 | 76.24 | 1.53 |
>
> Additionally, we construct more empirical results on several large-scale China Securities Index (CSI) datasets (i.e., CSI 50 and 500 futures) spanning from 2020 to 2023 for predicting directions (upward or downward) of futures prices [Zihao et al., 2019] [Qingyi et al., 2024]. The dataset collects over $42\times 10^{3}$ records spanning from 2020 to 2022, and each time step contains bid/ask prices and corresponding volumes. Table presents the performance of various semi-supervised methods across different label ratios, and shows that f-VAT significantly outperforms other competitive baselines on futures datasets, especially on more volatile CSI 500 futures. This is because f-VAT's adversarial perturbations incorporating key temporal structure facilitate deep models to effectively use unlabeled samples to yield smoother predictive distribution with better generalization.
>
> The performance comparison on domestic futures datasets:
> | Futures | Ratio | supL | PI | MTL | SemiTime | TapNet | TS-TCC | f-VAT |
> |:------:|:----:|---------------:|-----------------------:|--------------:|-------------------:|-----------------:|-----------------:|--------------------:|
> |        | 10% | 40.05 $\pm$ 1.38 | 42.87 $\pm$ 1.45 | 53.94 $\pm$ 0.13 | 55.19 $\pm$ 0.77 | 54.62 $\pm$ 0.54 | 55.72 $\pm$ 1.16 | **58.64 $\pm$ 0.53** |
> | 50     | 20% | 45.08 $\pm$ 1.45 | 47.26 $\pm$ 0.73 | 54.97 $\pm$ 0.61 | 56.69 $\pm$ 0.50 | 56.93 $\pm$ 1.28 | 58.21 $\pm$ 0.44 | **62.09 $\pm$ 0.26** |
> |        | 40% | 50.69 $\pm$ 3.53 | 52.42 $\pm$ 1.21 | 56.97 $\pm$ 0.57 | 57.34 $\pm$ 0.20 | 59.75 $\pm$ 1.09 | 59.80 $\pm$ 1.32 | **64.78 $\pm$ 0.45** |
> |        | 10% | 34.23 $\pm$ 0.24 | 44.00 $\pm$ 0.06 | 39.53 $\pm$ 1.35 | 38.86 $\pm$ 2.04 | 40.53 $\pm$ 0.83 | 39.79 $\pm$ 1.54 | **43.77 $\pm$ 0.73** |
> | 500    | 20% | 35.38 $\pm$ 1.06 | 46.57 $\pm$ 0.48 | 45.26 $\pm$ 0.29 | 46.04 $\pm$ 0.12 | 44.58 $\pm$ 1.70 | 45.05 $\pm$ 1.74 | **52.14 $\pm$ 0.25** |
> |        | 40% | 43.85 $\pm$ 1.49 | 49.25 $\pm$ 0.55 | 47.66 $\pm$ 1.50 | 50.65 $\pm$ 1.05 | 51.50 $\pm$ 1.36 | 54.29 $\pm$ 0.87 | **58.66 $\pm$ 0.46** |
>
> ### W3: Runtime comparison
> Thanks for your suggestions. Compared with VAT ($\mathcal{O}(N)$), the computational complexity of f-VAT is $\mathcal{O}(N\log(N))$ with a small constant factor, involving the FFT-based $\|\cdot\|_{H^{-s}}$ normalization. The empirical results show that the extra computational cost remains in the same order as VAT.
>
> Runtime comparison f-VAT and VAT:
> | Method  | CricketX | UWave | InsectWing | NATOPS | SelfReg |
> |-------|---------|-------------|-----------|-------|---------------|
> | VAT     | 15.67 | 51.66 | 35.58 | 28.04 | 30.68 |
> | f-VAT   | 20.45 | 62.05 | 45.92 | 38.98 | 44.95 |
> | Δ (%)   | 30.50 | 20.11 | 29.06 | 39.02 | 46.51 |
>
> ### W4: The available code for reproduction
> We provide the training logs and the source code in the supplementary materials for reference. Could you please provide more details while trying to reproduce empirical results, such as error messages or missing packages?
>
> - [Takeru et al., 2018] Takeru Miyato, Shin-ichi Maeda, Masanori Koyama, and Shin Ishii. Virtual adversarial training: a regularization method for supervised and semi-supervised learning. IEEE Transactions on Pattern Analysis and Machine Intelligence, 2018.
> - [Emadeldeen et al., 2023] Emadeldeen Eldele, Mohamed Ragab, Zhenghua Chen, Min Wu, Chee-Keong Kwoh, Xiaoli Li, and Cuntai Guan. Self-supervised contrastive representation learning for semi-supervised time-series classification. IEEE Transactions on Pattern Analysis and Machine Intelligence, , 2023.
> - [Haoyi et al., 2021] Haoyi Fan, Fengbin Zhang, Ruidong Wang, Xunhua Huang, and Zuoyong Li. Semi-supervised time series classification by temporal relation prediction. In ICASSP, IEEE, 2021.
> - [Zihao et al., 2019] Zihao Zhang, Stefan Zohren, and Stephen Roberts. Deeplob: Deep convolutional neural networks for limit order books. IEEE Transactions on Signal Processing, 2019.
> - [Qingyi et al., 2024] Qingyi Pan, Suyu Sun, Pei Yang, and Jingyi Zhang. Futuresnet: Capturing patterns of price fluctuations in domestic futures trading. Electronics, 13(22):4482, 2024

---

> ### Author Response · Authors · 2025-08-01
> **Accuracy and rank of each method across 30 datasets with label ratio 0.1 (First part)**
>
> Due to character limitations, we split each label-ratio table into two parts and posted them in separate comments for your reference.
> | dataset | SupCE | PI | MTL | meanTeacher | SemiTime | TapNet | TS-TCC | fVAT | SupCE_rank | PI_rank | MTL_rank | meanTeacher_rank | SemiTime_rank | TapNet_rank | TS-TCC_rank | f-VAT_rank |
> | --- | --- | --- | --- | --- | --- | --- | --- | --- | --- | --- | --- | --- | --- | --- | --- | --- |
> | ACSF1 | 13.75$\pm$3.75 | 16.40$\pm$8.60 | 13.28$\pm$2.34 | 35.94$\pm$2.82 | 42.97$\pm$2.51 | 32.82$\pm$3.12 | 43.75$\pm$2.64 | 57.29$\pm$2.61 | 7.00 | 6.00 | 8.00 | 4.00 | 3.00 | 5.00 | 2.00 | 1.00 |
> | Adiac | 5.98$\pm$4.02 | 7.54$\pm$1.74 | 6.06$\pm$0.26 | 16.73$\pm$2.68 | 11.19$\pm$6.19 | 29.04$\pm$1.11 | 29.25$\pm$2.88 | 33.94$\pm$4.20 | 8.00 | 6.00 | 7.00 | 4.00 | 5.00 | 3.00 | 2.00 | 1.00 |
> | AllGestureWiimoteX | 10.12$\pm$1.35 | 36.61$\pm$4.46 | 26.90$\pm$0.78 | 23.66$\pm$2.68 | 46.20$\pm$2.91 | 24.33$\pm$4.69 | 42.71$\pm$1.68 | 44.86$\pm$7.36 | 8.00 | 4.00 | 5.00 | 7.00 | 1.00 | 6.00 | 3.00 | 2.00 |
> | AllGestureWiimoteY | 12.68$\pm$1.21 | 33.93$\pm$7.59 | 20.42$\pm$0.34 | 20.76$\pm$2.90 | 51.49$\pm$2.24 | 32.36$\pm$3.79 | 44.19$\pm$5.58 | 55.36$\pm$1.78 | 8.00 | 4.00 | 7.00 | 6.00 | 2.00 | 5.00 | 3.00 | 1.00 |
> | AllGestureWiimoteZ | 12.68$\pm$1.21 | 25.67$\pm$6.47 | 18.02$\pm$3.74 | 19.20$\pm$2.92 | 38.17$\pm$3.79 | 23.66$\pm$2.68 | 34.38$\pm$5.84 | 33.54$\pm$4.91 | 8.00 | 4.00 | 7.00 | 6.00 | 1.00 | 5.00 | 2.00 | 3.00 |
> | ArrowHead | 38.10$\pm$0.10 | 52.34$\pm$4.22 | 39.61$\pm$0.86 | 39.84$\pm$2.80 | 60.78$\pm$2.34 | 64.84$\pm$1.56 | 61.67$\pm$3.96 | 71.25$\pm$1.41 | 8.00 | 5.00 | 7.00 | 6.00 | 4.00 | 2.00 | 3.00 | 1.00 |
> | BME | 45.83$\pm$2.67 | 75.00$\pm$2.57 | 50.78$\pm$2.34 | 45.32$\pm$2.82 | 64.06$\pm$2.78 | 90.24$\pm$1.17 | 74.48$\pm$2.70 | 79.69$\pm$2.69 | 7.00 | 3.00 | 6.00 | 8.00 | 5.00 | 1.00 | 4.00 | 2.00 |
> | Beef | 20.00$\pm$0.10 | 20.00$\pm$0.10 | 15.00$\pm$5.00 | 20.00$\pm$0.10 | 25.00$\pm$5.00 | 20.00$\pm$2.75 | 20.00$\pm$0.10 | 28.00$\pm$0.10 | 3.00 | 3.00 | 8.00 | 3.00 | 2.00 | 3.00 | 3.00 | 1.00 |
> | BeetleFly | 50.00$\pm$0.10 | 62.50$\pm$2.86 | 50.00$\pm$0.10 | 50.00$\pm$0.10 | 50.00$\pm$0.10 | 43.75$\pm$6.25 | 41.67$\pm$2.78 | 68.75$\pm$5.75 | 3.00 | 2.00 | 3.00 | 3.00 | 3.00 | 7.00 | 8.00 | 1.00 |
> | BirdChicken | 50.00$\pm$0.10 | 56.25$\pm$6.25 | 50.00$\pm$0.10 | 50.00$\pm$0.10 | 62.50$\pm$2.86 | 56.25$\pm$2.86 | 45.83$\pm$5.89 | 62.50$\pm$5.15 | 5.00 | 3.00 | 5.00 | 5.00 | 1.00 | 3.00 | 8.00 | 1.00 |
> | CBF | 24.22$\pm$0.10 | 99.46$\pm$0.53 | 98.16$\pm$0.24 | 75.10$\pm$2.89 | 99.74$\pm$0.26 | 99.20$\pm$0.80 | 98.84$\pm$0.74 | 99.92$\pm$0.58 | 8.00 | 3.00 | 6.00 | 7.00 | 2.00 | 4.00 | 5.00 | 1.00 |
> | Car | 25.00$\pm$0.10 | 41.66$\pm$2.87 | 25.00$\pm$0.10 | 25.00$\pm$0.10 | 37.50$\pm$2.86 | 52.08$\pm$2.09 | 25.00$\pm$0.10 | 58.34$\pm$4.16 | 5.00 | 3.00 | 5.00 | 5.00 | 4.00 | 2.00 | 5.00 | 1.00 |
> | Chinatown | 71.83$\pm$0.10 | 94.49$\pm$1.34 | 88.58$\pm$0.41 | 79.69$\pm$0.52 | 91.56$\pm$0.10 | 90.56$\pm$2.82 | 96.35$\pm$3.71 | 95.88$\pm$0.52 | 8.00 | 3.00 | 6.00 | 7.00 | 4.00 | 5.00 | 1.00 | 2.00 |
> | ChlorineConcentration | 55.36$\pm$0.10 | 56.06$\pm$1.96 | 54.96$\pm$0.09 | 53.70$\pm$0.10 | 55.67$\pm$0.23 | 52.18$\pm$1.14 | 69.43$\pm$3.43 | 66.97$\pm$0.68 | 5.00 | 3.00 | 6.00 | 7.00 | 4.00 | 8.00 | 1.00 | 2.00 |
> | CinCECGTorso | 18.49$\pm$0.10 | 71.94$\pm$2.83 | 38.72$\pm$1.96 | 29.84$\pm$2.41 | 62.32$\pm$2.75 | 86.58$\pm$0.37 | 79.68$\pm$4.01 | 91.96$\pm$2.09 | 8.00 | 4.00 | 6.00 | 7.00 | 5.00 | 2.00 | 3.00 | 1.00 |
> | Coffee | 50.00$\pm$0.10 | 50.00$\pm$0.10 | 50.00$\pm$0.10 | 50.00$\pm$0.10 | 75.00$\pm$2.86 | 70.00$\pm$2.86 | 50.00$\pm$0.10 | 82.00$\pm$0.10 | 4.00 | 4.00 | 4.00 | 4.00 | 2.00 | 3.00 | 4.00 | 1.00 |

---

> ### Author Response · Authors · 2025-08-01
> **Accuracy and rank of each method across 30 datasets with label ratio 0.1 (Second part)**
>
> | dataset | SupCE | PI | MTL | meanTeacher | SemiTime | TapNet | TS-TCC | f-VAT | SupCE_rank | PI_rank | MTL_rank | meanTeacher_rank | SemiTime_rank | TapNet_rank | TS-TCC_rank | f-VAT_rank |
> | --- | --- | --- | --- | --- | --- | --- | --- | --- | --- | --- | --- | --- | --- | --- | --- | --- |
> | Computers | 50.00$\pm$0.10 | 56.64$\pm$5.08 | 56.25$\pm$6.25 | 39.06$\pm$0.10 | 49.22$\pm$2.78 | 59.38$\pm$3.12 | 57.82$\pm$2.75 | 67.97$\pm$3.12 | 6.00 | 4.00 | 5.00 | 8.00 | 7.00 | 2.00 | 3.00 | 1.00 |
> | CricketX | 4.69$\pm$0.39 | 28.40$\pm$1.78 | 15.62$\pm$1.34 | 10.94$\pm$1.56 | 30.98$\pm$1.97 | 32.82$\pm$2.82 | 36.07$\pm$3.37 | 34.49$\pm$4.87 | 8.00 | 5.00 | 6.00 | 7.00 | 4.00 | 3.00 | 1.00 | 2.00 |
> | CricketY | 5.08$\pm$0.10 | 18.12$\pm$3.30 | 10.60$\pm$2.92 | 11.25$\pm$3.13 | 27.32$\pm$0.18 | 35.22$\pm$2.63 | 35.09$\pm$4.17 | 36.34$\pm$4.55 | 8.00 | 5.00 | 7.00 | 6.00 | 4.00 | 2.00 | 3.00 | 1.00 |
> | CricketZ | 6.48$\pm$1.40 | 25.31$\pm$0.40 | 11.45$\pm$3.24 | 10.94$\pm$1.56 | 32.10$\pm$2.01 | 32.10$\pm$1.38 | 35.51$\pm$3.27 | 38.58$\pm$0.13 | 8.00 | 5.00 | 6.00 | 7.00 | 3.00 | 3.00 | 2.00 | 1.00 |
> | Crop | 15.47$\pm$3.67 | 55.90$\pm$1.07 | 52.96$\pm$2.46 | 34.93$\pm$1.76 | 54.20$\pm$0.13 | 61.14$\pm$2.82 | 64.75$\pm$0.38 | 63.31$\pm$0.35 | 8.00 | 4.00 | 6.00 | 7.00 | 5.00 | 3.00 | 1.00 | 2.00 |
> | DiatomSizeReduction | 30.65$\pm$0.10 | 64.48$\pm$2.79 | 30.60$\pm$0.08 | 30.32$\pm$0.62 | 63.23$\pm$2.76 | 94.93$\pm$3.51 | 96.81$\pm$3.47 | 99.16$\pm$0.84 | 6.00 | 4.00 | 7.00 | 8.00 | 5.00 | 3.00 | 2.00 | 1.00 |
> | DistalPhalanxOutlineAgeGroup | 59.81$\pm$0.10 | 82.49$\pm$0.39 | 79.87$\pm$1.52 | 70.03$\pm$3.62 | 77.91$\pm$1.92 | 76.67$\pm$3.09 | 75.90$\pm$2.19 | 82.92$\pm$4.58 | 8.00 | 2.00 | 3.00 | 7.00 | 4.00 | 5.00 | 6.00 | 1.00 |
> | DistalPhalanxOutlineCorrect | 73.83$\pm$0.10 | 72.47$\pm$0.30 | 67.74$\pm$5.98 | 69.53$\pm$4.43 | 68.60$\pm$0.59 | 69.97$\pm$3.54 | 75.92$\pm$0.53 | 74.32$\pm$3.50 | 3.00 | 4.00 | 8.00 | 6.00 | 7.00 | 5.00 | 1.00 | 2.00 |
> | DistalPhalanxTW | 48.11$\pm$0.10 | 77.97$\pm$1.72 | 64.22$\pm$7.34 | 66.40$\pm$2.86 | 73.04$\pm$2.73 | 72.66$\pm$2.19 | 77.35$\pm$2.21 | 80.08$\pm$0.39 | 8.00 | 2.00 | 7.00 | 6.00 | 4.00 | 5.00 | 3.00 | 1.00 |
> | DodgerLoopDay | 13.79$\pm$0.10 | 18.96$\pm$2.86 | 20.69$\pm$2.81 | 13.79$\pm$0.10 | 24.14$\pm$3.00 | 27.58$\pm$6.89 | 17.24$\pm$2.75 | 28.58$\pm$3.00 | 7.00 | 5.00 | 4.00 | 7.00 | 3.00 | 2.00 | 6.00 | 1.00 |
> | DodgerLoopGame | 51.61$\pm$0.10 | 51.61$\pm$0.10 | 51.61$\pm$0.10 | 51.61$\pm$0.10 | 69.35$\pm$2.84 | 67.74$\pm$0.10 | 54.84$\pm$2.79 | 74.19$\pm$2.86 | 5.00 | 5.00 | 5.00 | 5.00 | 2.00 | 3.00 | 4.00 | 1.00 |
> | DodgerLoopWeekend | 70.97$\pm$0.10 | 70.97$\pm$0.10 | 88.71$\pm$2.86 | 70.97$\pm$0.10 | 77.42$\pm$3.23 | 88.71$\pm$1.61 | 82.79$\pm$2.82 | 95.16$\pm$1.61 | 6.00 | 6.00 | 2.00 | 6.00 | 5.00 | 2.00 | 4.00 | 1.00 |
> | ECG200 | 67.95$\pm$1.28 | 76.45$\pm$3.24 | 68.08$\pm$2.34 | 80.80$\pm$2.68 | 82.81$\pm$2.75 | 81.92$\pm$2.75 | 83.60$\pm$3.91 | 86.30$\pm$3.91 | 8.00 | 6.00 | 7.00 | 5.00 | 3.00 | 4.00 | 2.00 | 1.00 |
> | ECG5000 | 56.93$\pm$0.10 | 93.02$\pm$0.14 | 91.90$\pm$0.53 | 91.42$\pm$0.22 | 91.48$\pm$0.55 | 91.40$\pm$0.59 | 91.28$\pm$0.36 | 83.81$\pm$0.36 | 8.00 | 1.00 | 2.00 | 4.00 | 3.00 | 5.00 | 6.00 | 7.00 |
> | Average | 35.31$\pm$0.10 | 53.09$\pm$0.10 | 45.19$\pm$0.10 | 42.89$\pm$0.10 | 56.53$\pm$0.10 | 58.67$\pm$0.10 | 58.07$\pm$0.10 | 65.85$\pm$0.10 | 6.67 | 3.93 | 5.70 | 5.93 | 3.57 | 3.70 | 3.37 | 1.50 |

---

> ### Author Response · Authors · 2025-08-01
> **Accuracy and rank of each method across 30 datasets with label ratio 0.2 (First part)**
>
> Due to character limitations, we split each label-ratio table into two parts and posted them in separate comments for your reference.
>
> | Dataset | SupCE | PI | MTL | meanTeacher | SemiTime | TapNet | TS-TCC | fVAT | SupCE_rank | PI_rank | MTL_rank | meanTeacher_rank | SemiTime_rank | TapNet_rank | TS-TCC_rank | fVAT_rank |
> | --- | --- | --- | --- | --- | --- | --- | --- | --- | --- | --- | --- | --- | --- | --- | --- | --- |
> | ACSF1 | 10.00 $\pm$ 0.10 | 14.06 $\pm$ 1.56 | 10.93 $\pm$ 2.90 | 29.68 $\pm$ 2.80 | 21.10 $\pm$ 2.79 | 25.00 $\pm$ 2.82 | 24.22 $\pm$ 2.82 | 60.42 $\pm$ 2.63 | 8 | 6 | 7 | 2 | 5 | 3 | 4 | 1 |
> | Adiac | 5.98 $\pm$ 2.82 | 7.45 $\pm$ 2.87 | 2.54 $\pm$ 0.04 | 14.08 $\pm$ 2.89 | 20.08 $\pm$ 1.43 | 34.39 $\pm$ 0.14 | 30.44 $\pm$ 2.90 | 38.85 $\pm$ 2.62 | 7 | 6 | 8 | 5 | 4 | 2 | 3 | 1 |
> | AllGestureWiimoteX | 12.68 $\pm$ 1.21 | 52.23 $\pm$ 2.84 | 23.44 $\pm$ 0.22 | 50.00 $\pm$ 2.84 | 45.98 $\pm$ 0.45 | 40.62 $\pm$ 0.90 | 46.43 $\pm$ 2.83 | 55.36 $\pm$ 0.45 | 8 | 2 | 7 | 3 | 5 | 6 | 4 | 1 |
> | AllGestureWiimoteY | 12.68 $\pm$ 1.21 | 43.08 $\pm$ 2.95 | 24.38 $\pm$ 2.05 | 14.74 $\pm$ 1.78 | 34.60 $\pm$ 2.23 | 44.64 $\pm$ 1.78 | 58.19 $\pm$ 1.28 | 61.50 $\pm$ 2.78 | 8 | 4 | 6 | 7 | 5 | 3 | 2 | 1 |
> | AllGestureWiimoteZ | 13.89 $\pm$ 0.10 | 38.84 $\pm$ 2.54 | 20.31 $\pm$ 1.12 | 20.31 $\pm$ 2.75 | 42.18 $\pm$ 0.22 | 35.27 $\pm$ 2.68 | 40.85 $\pm$ 2.96 | 48.96 $\pm$ 0.67 | 8 | 4 | 6 | 6 | 2 | 5 | 3 | 1 |
> | ArrowHead | 38.10 $\pm$ 0.10 | 65.16 $\pm$ 2.71 | 55.78 $\pm$ 0.78 | 25.00 $\pm$ 0.10 | 67.50 $\pm$ 1.88 | 75.16 $\pm$ 2.76 | 69.06 $\pm$ 2.57 | 83.44 $\pm$ 2.63 | 7 | 5 | 6 | 8 | 4 | 2 | 3 | 1 |
> | BME | 45.83 $\pm$ 2.51 | 82.81 $\pm$ 2.55 | 49.22 $\pm$ 0.78 | 21.10 $\pm$ 2.93 | 65.62 $\pm$ 2.86 | 87.89 $\pm$ 2.79 | 88.54 $\pm$ 2.77 | 89.06 $\pm$ 2.79 | 7 | 4 | 6 | 8 | 5 | 3 | 2 | 1 |
> | Beef | 20.00 $\pm$ 0.10 | 25.00 $\pm$ 2.61 | 20.00 $\pm$ 0.10 | 20.00 $\pm$ 0.10 | 35.00 $\pm$ 2.88 | 25.00 $\pm$ 2.68 | 23.33 $\pm$ 2.52 | 40.00 $\pm$ 2.95 | 6 | 3 | 6 | 6 | 2 | 3 | 5 | 1 |
> | BeetleFly | 50.00 $\pm$ 0.10 | 50.00 $\pm$ 0.10 | 50.00 $\pm$ 0.10 | 50.00 $\pm$ 0.10 | 50.00 $\pm$ 0.10 | 37.50 $\pm$ 2.59 | 62.50 $\pm$ 2.79 | 63.50 $\pm$ 2.70 | 3 | 3 | 3 | 3 | 3 | 8 | 2 | 1 |
> | BirdChicken | 50.00 $\pm$ 0.10 | 50.00 $\pm$ 0.10 | 50.00 $\pm$ 0.10 | 54.00 $\pm$ 0.10 | 50.00 $\pm$ 0.10 | 52.00 $\pm$ 2.53 | 50.00 $\pm$ 0.10 | 53.25 $\pm$ 0.10 | 4 | 4 | 4 | 1 | 4 | 3 | 4 | 2 |
> | CBF | 57.80 $\pm$ 2.64 | 100.00 $\pm$ 0.00 | 97.55 $\pm$ 0.77 | 99.48 $\pm$ 0.52 | 98.96 $\pm$ 0.52 | 100.00 $\pm$ 0.00 | 97.54 $\pm$ 0.43 | 99.44 $\pm$ 1.95 | 8 | 1 | 6 | 3 | 5 | 1 | 7 | 4 |
> | Car | 25.00 $\pm$ 0.10 | 25.00 $\pm$ 0.10 | 29.16 $\pm$ 2.85 | 25.00 $\pm$ 0.10 | 37.50 $\pm$ 2.79 | 52.08 $\pm$ 2.09 | 29.17 $\pm$ 2.70 | 58.34 $\pm$ 2.61 | 6 | 6 | 5 | 6 | 3 | 2 | 4 | 1 |
> | Chinatown | 71.83 $\pm$ 0.10 | 87.43 $\pm$ 2.75 | 78.52 $\pm$ 2.60 | 97.92 $\pm$ 2.90 | 94.70 $\pm$ 0.10 | 96.35 $\pm$ 0.52 | 97.40 $\pm$ 0.85 | 94.96 $\pm$ 0.52 | 8 | 6 | 7 | 1 | 5 | 3 | 2 | 4 |
> | ChlorineConcentration | 55.36 $\pm$ 0.10 | 65.07 $\pm$ 0.90 | 54.12 $\pm$ 0.34 | 77.22 $\pm$ 1.88 | 57.29 $\pm$ 0.10 | 66.90 $\pm$ 0.76 | 53.70 $\pm$ 2.69 | 84.80 $\pm$ 0.10 | 6 | 4 | 7 | 2 | 5 | 3 | 8 | 1 |
> | CinCECGTorso | 18.49 $\pm$ 0.10 | 63.34 $\pm$ 2.51 | 34.00 $\pm$ 0.42 | 73.22 $\pm$ 2.67 | 74.31 $\pm$ 1.10 | 94.00 $\pm$ 1.04 | 63.69 $\pm$ 2.10 | 86.18 $\pm$ 3.00 | 8 | 6 | 7 | 4 | 3 | 1 | 5 | 2 |
> | Coffee | 50.00 $\pm$ 0.10 | 50.00 $\pm$ 0.10 | 50.00 $\pm$ 0.10 | 50.00 $\pm$ 0.10 | 50.00 $\pm$ 0.10 | 80.00 $\pm$ 2.64 | 50.00 $\pm$ 0.10 | 58.00 $\pm$ 0.10 | 3 | 3 | 3 | 3 | 3 | 1 | 3 | 2 |

---

> ### Author Response · Authors · 2025-08-01
> **Accuracy and rank of each method across 30 datasets with label ratio 0.2 (Second part)**
>
> | Dataset | SupCE | PI | MTL | meanTeacher | SemiTime | TapNet | TS-TCC | f-VAT | SupCE_rank | PI_rank | MTL_rank | meanTeacher_rank | SemiTime_rank | TapNet_rank | TS-TCC_rank | f-VAT_rank |
> | --- | --- | --- | --- | --- | --- | --- | --- | --- | --- | --- | --- | --- | --- | --- | --- | --- |
> | Computers | 50.00 $\pm$ 0.10 | 62.89 $\pm$ 2.60 | 63.28 $\pm$ 2.63 | 46.09 $\pm$ 1.17 | 63.28 $\pm$ 1.56 | 55.47 $\pm$ 2.72 | 59.38 $\pm$ 2.65 | 67.58 $\pm$ 2.75 | 7 | 4 | 2 | 8 | 2 | 6 | 5 | 1 |
> | CricketX | 6.48 $\pm$ 1.40 | 33.44 $\pm$ 2.82 | 16.77 $\pm$ 0.64 | 30.18 $\pm$ 2.80 | 41.83 $\pm$ 2.28 | 42.54 $\pm$ 0.41 | 48.93 $\pm$ 2.83 | 43.16 $\pm$ 0.45 | 8 | 5 | 7 | 6 | 4 | 3 | 1 | 2 |
> | CricketY | 6.48 $\pm$ 1.40 | 18.30 $\pm$ 2.77 | 17.21 $\pm$ 2.82 | 15.32 $\pm$ 1.56 | 39.91 $\pm$ 0.09 | 43.39 $\pm$ 2.73 | 41.25 $\pm$ 2.51 | 46.49 $\pm$ 2.55 | 8 | 5 | 6 | 7 | 4 | 2 | 3 | 1 |
> | CricketZ | 6.48 $\pm$ 1.40 | 30.09 $\pm$ 0.09 | 16.05 $\pm$ 1.90 | 36.03 $\pm$ 0.93 | 46.07 $\pm$ 1.16 | 39.02 $\pm$ 1.70 | 44.76 $\pm$ 2.94 | 41.20 $\pm$ 2.52 | 8 | 6 | 7 | 5 | 1 | 4 | 2 | 3 |
> | Crop | 23.44 $\pm$ 2.46 | 59.36 $\pm$ 2.77 | 48.42 $\pm$ 0.84 | 59.82 $\pm$ 2.61 | 58.88 $\pm$ 1.69 | 65.79 $\pm$ 1.77 | 71.77 $\pm$ 0.87 | 68.95 $\pm$ 0.43 | 8 | 5 | 7 | 4 | 6 | 3 | 1 | 2 |
> | DiatomSizeReduction | 35.48 $\pm$ 2.61 | 78.03 $\pm$ 2.61 | 30.60 $\pm$ 0.23 | 68.75 $\pm$ 0.73 | 89.84 $\pm$ 0.78 | 99.16 $\pm$ 0.10 | 71.91 $\pm$ 2.75 | 100.00 $\pm$ 0.00 | 7 | 4 | 8 | 6 | 3 | 2 | 5 | 1 |
> | DistalPhalanxOutlineAgeGroup | 59.81 $\pm$ 0.10 | 79.37 $\pm$ 0.43 | 70.13 $\pm$ 2.55 | 79.09 $\pm$ 0.04 | 74.78 $\pm$ 2.89 | 72.40 $\pm$ 0.10 | 78.24 $\pm$ 2.29 | 81.74 $\pm$ 2.55 | 8 | 2 | 7 | 3 | 5 | 6 | 4 | 1 |
> | DistalPhalanxOutlineCorrect | 73.83 $\pm$ 0.10 | 68.08 $\pm$ 0.37 | 55.92 $\pm$ 2.74 | 65.10 $\pm$ 0.86 | 79.91 $\pm$ 0.15 | 73.40 $\pm$ 2.77 | 79.57 $\pm$ 1.56 | 80.76 $\pm$ 0.10 | 4 | 6 | 8 | 7 | 2 | 5 | 3 | 1 |
> | DistalPhalanxTW | 48.11 $\pm$ 0.10 | 75.39 $\pm$ 0.08 | 67.26 $\pm$ 0.08 | 75.78 $\pm$ 0.10 | 74.61 $\pm$ 1.17 | 0.00 $\pm$ 0.10 | 75.78 $\pm$ 0.10 | 79.30 $\pm$ 0.39 | 7 | 4 | 6 | 2 | 5 | 8 | 2 | 1 |
> | DodgerLoopDay | 13.79 $\pm$ 0.10 | 18.96 $\pm$ 2.79 | 20.69 $\pm$ 2.90 | 17.24 $\pm$ 2.83 | 29.31 $\pm$ 2.78 | 34.48 $\pm$ 2.83 | 21.84 $\pm$ 2.66 | 26.20 $\pm$ 1.73 | 8 | 6 | 5 | 7 | 2 | 1 | 4 | 3 |
> | DodgerLoopGame | 51.61 $\pm$ 0.10 | 61.29 $\pm$ 2.90 | 53.22 $\pm$ 1.62 | 58.06 $\pm$ 2.79 | 59.68 $\pm$ 2.84 | 75.81 $\pm$ 2.78 | 59.14 $\pm$ 2.77 | 77.42 $\pm$ 2.81 | 8 | 3 | 7 | 6 | 4 | 2 | 5 | 1 |
> | DodgerLoopWeekend | 70.97 $\pm$ 0.10 | 83.87 $\pm$ 2.50 | 91.94 $\pm$ 1.62 | 82.26 $\pm$ 2.87 | 88.71 $\pm$ 2.83 | 95.16 $\pm$ 1.61 | 84.95 $\pm$ 2.91 | 96.77 $\pm$ 0.10 | 8 | 6 | 3 | 7 | 4 | 2 | 5 | 1 |
> | ECG200 | 66.67 $\pm$ 0.10 | 73.22 $\pm$ 0.90 | 78.02 $\pm$ 2.90 | 81.59 $\pm$ 2.71 | 84.38 $\pm$ 2.79 | 76.34 $\pm$ 2.81 | 80.36 $\pm$ 2.71 | 86.31 $\pm$ 2.90 | 8 | 7 | 5 | 3 | 2 | 6 | 4 | 1 |
> | ECG5000 | 56.93 $\pm$ 0.10 | 93.16 $\pm$ 0.10 | 92.28 $\pm$ 0.49 | 91.21 $\pm$ 0.29 | 91.96 $\pm$ 0.33 | 92.43 $\pm$ 0.54 | 92.21 $\pm$ 0.08 | 94.21 $\pm$ 0.08 | 8 | 2 | 4 | 7 | 6 | 3 | 5 | 1 |
> | Average | 36.92 | 55.16 | 45.72 | 50.94 | 58.93 | 60.41 | 59.84 | 68.87 | 7.00 | 4.40 | 5.87 | 4.87 | 3.77 | 3.40 | 3.67 | 1.50 |

---

> ### Author Response · Authors · 2025-08-01
> **Accuracy and rank of each method across 30 datasets with label ratio 0.4 (First part)**
>
> | dataset | SupCE | PI | MTL | meanTeacher | SemiTime | TapNet | TS-TCC | fVAT | SupCE_rank | PI_rank | MTL_rank | meanTeacher_rank | SemiTime_rank | TapNet_rank | TS-TCC_rank | fVAT_rank |
> | :-- | :-- | :-- | :-- | :-- | :-- | :-- | :-- | :-- | :-- | :-- | :-- | :-- | :-- | :-- | :-- | :-- |
> | ACSF1 | 11.25 $\pm$ 1.25 | 33.60 $\pm$ 2.34 | 8.59 $\pm$ 2.34 | 29.69 $\pm$ 2.70 | 35.16 $\pm$ 2.66 | 43.75 $\pm$ 1.56 | 44.79 $\pm$ 2.88 | 46.88 $\pm$ 1.56 | 7 | 5 | 8 | 6 | 4 | 3 | 2 | 1 |
> | Adiac | 4.78 $\pm$ 2.87 | 3.93 $\pm$ 1.25 | 2.93 $\pm$ 0.25 | 11.06 $\pm$ 2.91 | 36.26 $\pm$ 0.14 | 55.67 $\pm$ 1.25 | 48.67 $\pm$ 2.92 | 60.08 $\pm$ 1.21 | 6 | 7 | 8 | 5 | 4 | 2 | 3 | 1 |
> | AllGestureWiimoteX | 13.89 $\pm$ 0.10 | 51.56 $\pm$ 1.11 | 25.22 $\pm$ 0.67 | 55.58 $\pm$ 0.67 | 54.28 $\pm$ 0.67 | 57.36 $\pm$ 2.80 | 55.65 $\pm$ 2.76 | 59.15 $\pm$ 2.58 | 8 | 6 | 7 | 4 | 5 | 2 | 3 | 1 |
> | AllGestureWiimoteY | 13.89 $\pm$ 0.10 | 52.90 $\pm$ 2.45 | 24.22 $\pm$ 2.79 | 56.03 $\pm$ 0.67 | 69.19 $\pm$ 2.24 | 62.50 $\pm$ 2.82 | 54.28 $\pm$ 0.56 | 61.99 $\pm$ 2.80 | 8 | 6 | 7 | 4 | 1 | 2 | 5 | 3 |
> | AllGestureWiimoteZ | 13.89 $\pm$ 0.10 | 49.11 $\pm$ 2.68 | 18.52 $\pm$ 0.22 | 54.91 $\pm$ 1.34 | 50.22 $\pm$ 2.69 | 47.99 $\pm$ 0.67 | 36.16 $\pm$ 2.73 | 54.61 $\pm$ 2.90 | 8 | 4 | 7 | 1 | 3 | 5 | 6 | 2 |
> | ArrowHead | 38.10 $\pm$ 0.10 | 67.66 $\pm$ 2.67 | 41.64 $\pm$ 2.77 | 77.81 $\pm$ 2.95 | 73.28 $\pm$ 2.86 | 62.82 $\pm$ 2.96 | 66.46 $\pm$ 2.86 | 81.56 $\pm$ 2.75 | 8 | 4 | 7 | 2 | 3 | 6 | 5 | 1 |
> | BME | 45.84 $\pm$ 2.97 | 88.28 $\pm$ 2.70 | 54.30 $\pm$ 2.71 | 76.56 $\pm$ 2.81 | 93.75 $\pm$ 2.90 | 95.32 $\pm$ 1.56 | 89.06 $\pm$ 2.70 | 100.00 $\pm$ 0.00 | 8 | 5 | 7 | 6 | 3 | 2 | 4 | 1 |
> | Beef | 20.00 $\pm$ 0.10 | 25.00 $\pm$ 2.79 | 20.00 $\pm$ 0.10 | 25.00 $\pm$ 2.79 | 25.00 $\pm$ 2.79 | 30.00 $\pm$ 2.85 | 20.00 $\pm$ 0.10 | 50.00 $\pm$ 2.86 | 6 | 3 | 6 | 3 | 3 | 2 | 6 | 1 |
> | BeetleFly | 50.00 $\pm$ 0.10 | 56.25 $\pm$ 2.75 | 50.00 $\pm$ 0.10 | 50.00 $\pm$ 0.10 | 68.75 $\pm$ 2.88 | 62.50 $\pm$ 2.92 | 50.00 $\pm$ 0.10 | 87.50 $\pm$ 2.77 | 5 | 4 | 5 | 5 | 2 | 3 | 5 | 1 |
> | BirdChicken | 50.00 $\pm$ 0.10 | 62.50 $\pm$ 2.55 | 50.00 $\pm$ 0.10 | 56.25 $\pm$ 2.55 | 81.25 $\pm$ 2.77 | 50.00 $\pm$ 2.77 | 50.00 $\pm$ 0.10 | 62.50 $\pm$ 2.69 | 5 | 2 | 5 | 4 | 1 | 5 | 5 | 2 |
> | CBF | 37.89 $\pm$ 2.62 | 100.00 $\pm$ 0.00 | 97.92 $\pm$ 1.04 | 99.48 $\pm$ 0.52 | 100.00 $\pm$ 0.00 | 99.71 $\pm$ 0.29 | 100.00 $\pm$ 0.00 | 98.96 $\pm$ 0.84 | 8 | 1 | 7 | 5 | 1 | 4 | 1 | 6 |
> | Car | 22.92 $\pm$ 2.09 | 25.00 $\pm$ 0.10 | 25.00 $\pm$ 0.10 | 45.84 $\pm$ 2.80 | 45.83 $\pm$ 0.10 | 60.42 $\pm$ 0.10 | 41.67 $\pm$ 2.78 | 63.67 $\pm$ 2.93 | 8 | 6 | 6 | 3 | 4 | 2 | 5 | 1 |
> | Chinatown | 71.83 $\pm$ 0.10 | 86.68 $\pm$ 2.59 | 83.08 $\pm$ 2.77 | 97.92 $\pm$ 1.04 | 91.98 $\pm$ 0.10 | 93.98 $\pm$ 2.77 | 98.96 $\pm$ 0.85 | 99.40 $\pm$ 0.52 | 8 | 6 | 7 | 3 | 5 | 4 | 2 | 1 |
> | ChlorineConcentration | 55.36 $\pm$ 0.10 | 80.24 $\pm$ 2.69 | 54.18 $\pm$ 0.64 | 89.92 $\pm$ 2.65 | 59.94 $\pm$ 1.61 | 78.43 $\pm$ 1.77 | 53.70 $\pm$ 0.39 | 92.97 $\pm$ 0.10 | 6 | 3 | 7 | 2 | 5 | 4 | 8 | 1 |
> | CinCECGTorso | 32.94 $\pm$ 2.93 | 79.74 $\pm$ 2.67 | 41.37 $\pm$ 2.03 | 82.10 $\pm$ 2.76 | 86.61 $\pm$ 1.39 | 93.50 $\pm$ 0.91 | 91.57 $\pm$ 2.88 | 94.70 $\pm$ 0.35 | 8 | 6 | 7 | 5 | 4 | 2 | 3 | 1 |
> | Coffee | 50.00 $\pm$ 0.10 | 100.00 $\pm$ 0.00 | 50.00 $\pm$ 0.10 | 90.00 $\pm$ 0.10 | 99.00 $\pm$ 0.10 | 99.00 $\pm$ 0.10 | 80.00 $\pm$ 2.70 | 100.00 $\pm$ 0.00 | 7 | 1 | 7 | 5 | 3 | 3 | 6 | 1 |

---

> ### Author Response · Authors · 2025-08-01
> **Accuracy and rank of each method across 30 datasets with label ratio 0.4 (Second part)**
>
> | dataset | SupCE | PI | MTL | meanTeacher | SemiTime | TapNet | TS-TCC | fVAT | SupCE_rank | PI_rank | MTL_rank | meanTeacher_rank | SemiTime_rank | TapNet_rank | TS-TCC_rank | fVAT_rank |
> | :-- | :-- | :-- | :-- | :-- | :-- | :-- | :-- | :-- | :-- | :-- | :-- | :-- | :-- | :-- | :-- | :-- |
> | Computers | 50.00 $\pm$ 0.10 | 68.36 $\pm$ 2.73 | 53.71 $\pm$ 2.99 | 66.02 $\pm$ 2.73 | 60.94 $\pm$ 0.10 | 65.62 $\pm$ 2.99 | 66.40 $\pm$ 2.97 | 71.09 $\pm$ 0.10 | 8 | 2 | 7 | 4 | 6 | 5 | 3 | 1 |
> | CricketX | 5.08 $\pm$ 0.10 | 42.77 $\pm$ 2.82 | 18.84 $\pm$ 2.05 | 52.68 $\pm$ 1.61 | 58.76 $\pm$ 0.62 | 63.57 $\pm$ 2.98 | 56.49 $\pm$ 2.80 | 66.78 $\pm$ 0.71 | 8 | 6 | 7 | 5 | 3 | 2 | 4 | 1 |
> | CricketY | 22.76 $\pm$ 2.78 | 31.25 $\pm$ 2.79 | 15.14 $\pm$ 3.00 | 30.62 $\pm$ 2.87 | 54.51 $\pm$ 2.84 | 60.31 $\pm$ 2.99 | 54.23 $\pm$ 2.84 | 64.10 $\pm$ 2.95 | 7 | 5 | 8 | 6 | 3 | 2 | 4 | 1 |
> | CricketZ | 6.48 $\pm$ 1.40 | 44.20 $\pm$ 2.68 | 18.84 $\pm$ 2.93 | 40.62 $\pm$ 2.80 | 55.80 $\pm$ 2.86 | 60.14 $\pm$ 2.98 | 56.13 $\pm$ 2.93 | 63.75 $\pm$ 2.94 | 8 | 5 | 7 | 6 | 4 | 2 | 3 | 1 |
> | Crop | 24.52 $\pm$ 0.96 | 71.32 $\pm$ 2.80 | 49.84 $\pm$ 0.66 | 54.92 $\pm$ 2.39 | 62.40 $\pm$ 0.78 | 60.79 $\pm$ 0.26 | - | 74.32 $\pm$ 1.47 | 7 | 2 | 6 | 5 | 3 | 4 | - | 1 |
> | DiatomSizeReduction | 31.46 $\pm$ 0.80 | 92.16 $\pm$ 2.79 | 45.14 $\pm$ 2.83 | 82.19 $\pm$ 2.85 | 95.32 $\pm$ 2.81 | 95.21 $\pm$ 2.82 | 59.48 $\pm$ 2.88 | 96.21 $\pm$ 0.26 | 8 | 4 | 7 | 5 | 2 | 3 | 6 | 1 |
> | DistalPhalanxOutlineAgeGroup | 59.81 $\pm$ 0.10 | 76.08 $\pm$ 2.79 | 72.05 $\pm$ 2.79 | 78.31 $\pm$ 1.52 | 79.76 $\pm$ 2.34 | 78.73 $\pm$ 2.66 | 78.22 $\pm$ 2.32 | 76.76 $\pm$ 0.94 | 8 | 6 | 7 | 3 | 1 | 2 | 4 | 5 |
> | DistalPhalanxOutlineCorrect | 73.83 $\pm$ 0.10 | 73.32 $\pm$ 2.78 | 59.92 $\pm$ 2.52 | 79.91 $\pm$ 0.37 | 80.43 $\pm$ 0.52 | 65.62 $\pm$ 2.78 | 78.08 $\pm$ 2.70 | 81.32 $\pm$ 2.14 | 5 | 6 | 8 | 3 | 2 | 7 | 4 | 1 |
> | DistalPhalanxTW | 48.11 $\pm$ 0.10 | 71.59 $\pm$ 2.78 | 71.48 $\pm$ 0.86 | 75.78 $\pm$ 0.10 | 76.95 $\pm$ 0.39 | 73.23 $\pm$ 1.82 | 75.78 $\pm$ 0.10 | 78.59 $\pm$ 2.79 | 8 | 6 | 7 | 3 | 2 | 5 | 3 | 1 |
> | DodgerLoopDay | 13.79 $\pm$ 0.10 | 42.58 $\pm$ 2.79 | 25.86 $\pm$ 1.72 | 22.42 $\pm$ 2.77 | 29.31 $\pm$ 1.72 | 44.82 $\pm$ 2.88 | 24.14 $\pm$ 2.89 | 43.82 $\pm$ 2.96 | 8 | 3 | 5 | 7 | 4 | 1 | 6 | 2 |
> | DodgerLoopGame | 51.61 $\pm$ 0.10 | 71.58 $\pm$ 2.51 | 51.61 $\pm$ 0.10 | 75.81 $\pm$ 1.62 | 70.97 $\pm$ 2.92 | 75.81 $\pm$ 2.90 | 59.14 $\pm$ 2.86 | 78.15 $\pm$ 2.63 | 7 | 4 | 7 | 2 | 5 | 2 | 6 | 1 |
> | DodgerLoopWeekend | 70.97 $\pm$ 0.10 | 92.83 $\pm$ 2.71 | 90.32 $\pm$ 0.10 | 80.64 $\pm$ 2.84 | 98.38 $\pm$ 1.62 | 96.77 $\pm$ 0.10 | 70.97 $\pm$ 0.10 | 99.38 $\pm$ 0.96 | 7 | 4 | 5 | 6 | 2 | 3 | 7 | 1 |
> | ECG200 | 66.67 $\pm$ 0.10 | 76.14 $\pm$ 2.96 | 72.72 $\pm$ 3.00 | 85.04 $\pm$ 2.92 | 83.48 $\pm$ 2.87 | 80.24 $\pm$ 2.78 | 80.24 $\pm$ 2.78 | 83.14 $\pm$ 0.16 | 8 | 6 | 7 | 1 | 2 | 4 | 4 | 3 |
> | ECG5000 | 56.93 $\pm$ 0.10 | 91.38 $\pm$ 2.77 | 90.83 $\pm$ 0.19 | 92.30 $\pm$ 0.32 | 93.04 $\pm$ 0.10 | 94.68 $\pm$ 0.24 | 94.68 $\pm$ 0.24 | 95.68 $\pm$ 0.36 | 8 | 6 | 7 | 5 | 4 | 2 | 2 | 1 |
> | - | 37.15  | 63.60  | 46.11  | 63.85 $\pm$ - | 69.02 | 70.28 | 63.27  | 76.24  | 7.33 | 4.47 | 6.80 | 4.13 | 3.13 | 3.17 | 4.27 | 1.53 |

---

> > ### Comment · Reviewer_QUWH · 2025-08-04
> >
> > I appreciate the authors' efforts in the rebuttal, and apologize for leaving out the code in the supplementary materials. After carefully reading the rebuttal and other reviewers' comments, I maintain my position that the novelty of this work is borderline for NeurIPS. And to be honest, regarding the motivation, I'm still not quite convinced by the choice of VAT. Specifically, although the suggested f-VAT achieves a non-trivial improvement in performance compared with the existing method, integrating the same "functional" adversarial data augmentation into other more advanced SSL frameworks may still help achieve better improvement. That is, instead, what I'm mainly concerned about. However, the authors provide a large number of new empirical results as suggested, which look promising and indeed address some points of my concern. Therefore, I would increase my rating to borderline accept.

---

> > > ### Author Response · Authors · 2025-08-04
> > >
> > > Thank you for your efforts in timely reviewing our rebuttal and recognizing the contributions of our work.  Additionally, we would like to emphasize that we theoretically establish the duality between the perturbation and the smoothness (non-)linear functional models, which inspires our proposed method, is yet another key contribution of our paper. We would appreciate it if you can recognize our theory contribution.

---

### Official Review · Reviewer_jNPY · 2025-07-04

**Clarity:** 3
**Significance:** 3
**Originality:** 3
**Rating:** 5
**Confidence:** 4

**Summary:**

This paper proposes a framework of functional Virtual Adversarial Training (f-VAT) for semi-supervised time series classification. The theoretical analysis and foundation are established for the proposed  algorithm. The experimental results demonatre its effectiveness.

**Questions:**

Accoding to Eq.(11), the H^s norm can be only approximated, how about the approximation error here? Is there the other method to compute  the norm? How to select the proper s for a practical problem or task?

**Ethical Concerns:**

["NO or VERY MINOR ethics concerns only"]

**Final Justification:**

The authors addressed my concerns and questions and I still kept my previous scores.

**Limitations:**

Yes

**Paper Formatting Concerns:**

No.

**Quality:**

3

**Strengths And Weaknesses:**

Strengths:
    This paper tries to utlize the Virtual Adversarial Training (VAT) method on leveraging the unlabeled data of semi-supervised time series classification for smoothing the predictive distributions and proposes the framework of functional Virtual Adversarial Training  (f-VAT) to incorporate the functional structure of the data into perturbations. It theoretically establishes a duality between the perturbation norm and the functional model sensitivity, and then use an appropriate Sobolev norm to generate structured functional adversarial perturbations for semi-supervised time series classification. The proposed algorithm is reasonable  and novel for  semi-supervised time series classification. The experiments  strongly support the theoretical  analysis.

Weeknesses: The description of theoretical results is not so logic or explainable. It is a big jump to make the derivation from the linear model to the continously differentiiable case. The authors should give a good explain for this generalization in the paper.

---

> ### Author Rebuttal · Authors · 2025-07-31
>
> ### W1: Description of theoretical results
> We apologize for not making the generalization clear in the original version. The case of **linear model (as also functional linear model in the infinite-dimensional case) provides main insights into the duality between the adversarial perturbation norm and the model sensitivity norm**, while the generalization to the continuously differentiable case serves mainly as additional theoretical supports for more complex models, though, due to the highly complex nature of neural networks, further theoretical analysis would be beyond the scope of this paper.
> Nevertheless, our existing theory has already provided much insight for the design of f-VAT with the duality theory--we should choose the proper Sobolev $H^{-s}$ norm (but not $H^s$) for adversarial perturbations, which is key for the success of our method.
>
> We hope that this explanation clarifies your doubts, and we will make it more clear in the revised version.
>
> ### Q1: Approximation error
> Thank you for your insightful question. The discrete approximation of the $H^s$ norm is known as the spectral method [Canuto et al., 2006], which is the most common way to approximate the Sobolev norm with discrete observations. The approximation error is determined by multiple factors, including the order $s$, the distribution of the samples, and the smoothness of the function. Roughly speaking, assuming that $u \in H^r$ for some $r > s \in \mathbb{R}$, the approximation error of the $H^s$ norm with $N$ equidistant samples would be $\mathcal{O}(N^{r-s})$.
>
> There are indeed other methods to compute the Sobolev norm, such as finite element methods and wavelet methods. However, they are more complicated and less efficient. Since the approximation error is not our main focus, we think the spectral method is sufficient for our purpose.
>
> References:
> - [Canuto et al., 2006] Claudio Canuto, M. Youssuff Hussaini, Alfio Quarteroni, and Thomas A. Zang. Spectral Methods: Fundamentals in Single Domains. Scientific Computation. Springer Berlin Heidelberg,2006.
>
> ### Q2: Selection of $s$
> Thanks for your suggestions. We evaluate the performance on more datasets in Table, where $s=0$ reduces to the original VAT[Takeru et al., 2018]. Table shows that setting $s=2$ achieves the best performance in almost all settings. For high volatile time series data, setting high order $s=3$ generates adversarial perturbations that capture low-frequency temporal structure, while for relatively stable time series data, the small order $s=1$ allows the perturbation to flexibly explore the input space. In practice, it suffices to evaluate $s\in{1,2,3,4}$ based on the validation set and report the average performance over several runs.
>
> The performance of f-VAT with different Sobolev norm order $s$:
> | $s$ | CricketX               | UWave           | InsectWing            | NATOPS               | SelfReg        |
> |:---:|-----------------------:|-----------------------:|----------------------:|---------------------:|----------------------:|
> | 0 | 58.63 $\pm$ 0.50 | 94.76 $\pm$ 0.54 | 63.48 $\pm$ 0.30 | 90.15 $\pm$ 1.60 | 53.47 $\pm$ 1.04 |
> | 1 | 59.91 $\pm$ 2.32 | 96.54 $\pm$ 0.67 | 66.70 $\pm$ 0.50 | 89.58 $\pm$ 0.12 | 56.16 $\pm$ 1.73 |
> | 2 | 61.66 $\pm$ 2.33 | 97.16 $\pm$ 0.28 | 67.08 $\pm$ 0.86 | 93.13 $\pm$ 0.15 | 58.86 $\pm$ 0.35 |
> | 3 | 60.44 $\pm$ 0.23 | 96.82 $\pm$ 0.16 | 64.10 $\pm$ 0.86 | 90.10 $\pm$ 0.65 | 51.39 $\pm$ 1.21 |
> | 4 | 58.22 $\pm$ 3.39 | 96.71 $\pm$ 0.61 | 66.11 $\pm$ 0.82 | 87.51 $\pm$ 0.52 | 50.93 $\pm$ 0.12 |
>
> Reference
> - [Takeru et al., 2018] Takeru Miyato, Shin-ichi Maeda, Masanori Koyama, and Shin Ishii. Virtual adversarial training: a regularization method for supervised and semi-supervised learning. IEEE Transactions on Pattern Analysis and Machine Intelligence, 2018.

---

### Official Review · Reviewer_Jy1d · 2025-07-12

**Clarity:** 3
**Significance:** 3
**Originality:** 3
**Rating:** 5
**Confidence:** 3

**Summary:**

This paper proposes functional Virtual Adversarial Training (f-VAT), a novel method for semi-supervised time series classification. The key insight is that standard Virtual Adversarial Training (VAT), when applied to time series, uses Euclidean norm-bounded perturbations, which ignores the temporal structure of the data and leads to unrealistic "jagged" adversarial examples. To address this, f-VAT incorporates the structure of time series by constraining the adversarial perturbations using a Sobolev norm. Empirically, the authors demonstrate that f-VAT significantly outperforms standard VAT and other state-of-the-art methods on several semi-supervised and fully-supervised time series classification benchmarks.

**Questions:**

- Regarding the choice of the Sobolev order `s`, do the authors have any practical guidance on selecting `s` for a new task without performing an exhaustive search?
- Could you elaborate on how one might design other norms for different types of time series? For instance, for data with known seasonality or periodic patterns, could a norm be designed in the Fourier domain to specifically encourage or penalize perturbations in certain frequency bands?

**Ethical Concerns:**

["NO or VERY MINOR ethics concerns only"]

**Final Justification:**

The author response provides the justification for s and the empirical runtime data, which address my initial concern. As a result, I'm increasing my score.

**Limitations:**

The authors have adequately addressed the limitations of their work in Section 5

**Quality:**

4

**Strengths And Weaknesses:**

##### Strengths
- The paper introduces a novel and well-motivated approach, where the authors use functional analysis tools (Sobolev norms) to generate structured adversarial perturbations for time series data
- The work is technically strong, where it provides a theoretical foundation for the proposed framework, establishing a duality between the perturbation norm and the model's sensitivity.
- The empirical evaluation is comprehensive, where the proposed method is tested on six datasets, consistently showing significant performance improvements over multiple baselines, including standard VAT. The results hold across different ratios of labeled data and in a fully-supervised setting.
- The paper is well-written. The qualitative analyses, including visualizations of perturbations and loss landscapes, provide good intuition and support the paper's claims.
##### Weaknesses
- The paper's empirical validation of the Sobolev order `s` is based on a single dataset (NATOPS), where `s=2` was found to be optimal. It is not entirely clear if this choice is universally optimal or if `s` is a sensitive hyperparameter that would require tuning for each new dataset. A brief discussion on the generalization of this choice would strengthen the paper.
- While the paper mentions the efficacy of the norm computation (O(N log N)), I could not find an explicit comparison of the total training time between f-VAT and standard VAT. A discussion on the practical computational overhead would be a valuable addition.

---

> ### Author Rebuttal · Authors · 2025-07-31
>
> ### W1 & Q1: Selection of $s$
> Thanks for your suggestions. We evaluate the performance on more datasets in Table, where $s=0$ reduces to the original VAT[Takeru et al., 2018]. Table shows that setting $s=2$ achieves the best performance in almost all settings. For high volatile time series data, setting high order $s=3$ generates adversarial perturbations that preserve low-frequency temporal information, while for relatively stable time series data, the small order $s=1$ allows the perturbation to flexibly explore the input space. In practice, it suffices to evaluate $s\in[1,2,3,4]$ based on the validation set and report the average performance over several runs.
>
> The performance of f-VAT with different Sobolev norm order $s$:
> | $s$ | CricketX               | UWave          | InsectWing            | NATOPS               | SelfReg        |
> |:---:|-----------------------:|-----------------------:|----------------------:|---------------------:|----------------------:|
> | 0 | 58.63 $\pm$ 0.50 | 94.76 $\pm$ 0.54 | 63.48 $\pm$ 0.30 | 90.15 $\pm$ 1.60 | 53.47 $\pm$ 1.04 |
> | 1 | 59.91 $\pm$ 2.32 | 96.54 $\pm$ 0.67 | 66.70 $\pm$ 0.50 | 89.58 $\pm$ 0.12 | 56.16 $\pm$ 1.73 |
> | 2 | 61.66 $\pm$ 2.33 | 97.16 $\pm$ 0.28 | 67.08 $\pm$ 0.86 | 93.13 $\pm$ 0.15 | 58.86 $\pm$ 0.35 |
> | 3 | 60.44 $\pm$ 0.23 | 96.82 $\pm$ 0.16 | 64.10 $\pm$ 0.86 | 90.10 $\pm$ 0.65 | 51.39 $\pm$ 1.21 |
> | 4 | 58.22 $\pm$ 3.39 | 96.71 $\pm$ 0.61 | 66.11 $\pm$ 0.82 | 87.51 $\pm$ 0.52 | 50.93 $\pm$ 0.12 |
>
> Reference
>
> - [Takeru et al., 2018] Takeru Miyato, Shin-ichi Maeda, Masanori Koyama, and Shin Ishii. Virtual adversarial training: a regularization method for supervised and semi-supervised learning. IEEE Transactions on Pattern Analysis and Machine Intelligence, 2018.
>
> ### W2: Runtime comparison
> Thanks for your suggestions. Compared with VAT ($\mathcal{O}(N)$), the computational complexity of f-VAT is $\mathcal{O}(N\log(N))$ with a small constant factor, involving the FFT-based $\|\cdot\|_{H^{-s}}$ normalization. The empirical results show that the extra computational cost remains in the same order as VAT.
>
> Runtime comparison f-VAT and VAT:
> | Method  | CricketX | UWave | InsectWing | NATOPS | SelfReg |
> |-------|---------|-------------|-----------|-------|---------------|
> | VAT     | 15.67 | 51.66 | 35.58 | 28.04 | 30.68 |
> | f-VAT   | 20.45 | 62.05 | 45.92 | 38.98 | 44.95 |
> | Δ (%)   | 30.50 | 20.11 | 29.06 | 39.02 | 46.51 |
>
> ### Q2: Norms for different types of time series
> Thanks for your insightful comments. Indeed, we should design other norms to align with different types of time series. For example, for data with known seasonality or periodic patterns, we can design a norm that additionally penalizes the non-periodicity of the perturbation to align with the periodic patterns, which could be done using the Fourier domain decomposition. We believe that investigating additional structure of the time series would further enhance the performance of the f-VAT method, and it can be one of our future directions.

---

> > ### Comment · Reviewer_Jy1d · 2025-08-05
> > **response**
> >
> > Thank you for providing the justification for s and the empirical runtime data, which address my initial concern. As a result, I'm increasing my score.

---

> ### Author Response · Authors · 2025-08-06
>
> Thank you as well! We will follow your suggestion to provide a more detailed justification for $s$ and include additional runtime analyses in our revised manuscript. We sincerely appreciate your positive assessment of our paper.

---

### Comment · Area_Chair_qfJV · 2025-08-04

Dear Reviewers,

Thank you for your time and effort in reviewing submissions for NeurIPS 2025. As we approach the final stage of the review process, we kindly remind you to submit your responses to the author rebuttals by **August 6**.

Your engagement in this discussion phase is crucial to ensuring a fair and thorough evaluation of each submission. Please:

- Carefully consider the authors’ rebuttal and any additional evidence they provide.

- Update your review (if applicable) to reflect your revised perspective.

**The authors provided a thorough experimental analysis in their response to `Reviewer QUWH` and `Reviewer RR58`, Could you please have a look at the rebuttal and reply to the authors as soon as possible?**


Your AC

---

### Note · Authors · 2025-08-13

We appreciate the AC and all the reviewers for their great efforts and constructive comments. Below, we summarize the key points of our rebuttal and discussions.

- Theory contributions: We clarify that the duality between perturbation norms and model sensitivity for (non-)linear functional models is a key contribution of our paper. The theory provides us with key insights into constructing perturbation that aligns with the temporal strucutre of time series data in various function spaces that perform well in practice.

- More empirical results: We add empirical results on 30 randomly sampled UCR/UEA  datasets and include a recent TS‑TCC baseline. Our proposed f-VAT achieves the best average accuracy and rank across various semi-supervised settings. We also conduct experiments on large-scale China Securities Index datasets CSI 50/500, where f-VAT still outperforms other baselines, especially on more volatile CSI 500 futures. Additionally, we construct additional ablation studies for the selection of Sobolev order s and empirical runtime comparison to give practical guidance on each new dataset.

We believe the manuscript addresses the main concerns and is significant both empirically and theoretically for semi-supervised time series classification.

---

### Decision · Program_Chairs · 2025-09-17

**Decision:**

Accept (poster)

**Comment:**

This paper proposes functional Virtual Adversarial Training (f-VAT), a new framework for semi-supervised time series classification. Unlike standard VAT, which relies on Euclidean norm perturbations that fail to capture temporal structures, f-VAT introduces structural functional perturbations. By establishing a duality between perturbation norms and model sensitivity, the authors design structured adversarial perturbations that preserve low-frequency information while suppressing high-frequency noise. Extensive experiments on multiple datasets demonstrate consistent and significant performance improvements over baselines.

The strengths of this paper include:

1.	The work introduces a novel and well-motivated approach by applying Sobolev norms to construct structured adversarial perturbations that are more appropriate for time series data.

2.	The paper is theoretically solid, providing a clear foundation and duality between perturbation norm and model sensitivity.

3.	The empirical evaluation is comprehensive, demonstrating improvements across multiple semi-supervised datasets and settings.

4.	The paper is also clearly written, with qualitative visualizations that support the claims.

The authors further clarified their theory contributions during the rebuttal period. They have also provided additional experiments on more datasets, runtime comparisons, and clarifications on the selection of the Sobolev order, which resolved most of the reviewers’ concerns. Reviewers acknowledged that their initial issues were addressed.

I recommend acceptance. This is a technically strong paper with clear theoretical contributions and thorough empirical validation. That said, the revision should